# QUANTIFYING DIFFERENCES IN REWARD FUNCTIONS

**Adam Gleave**[1,2]   **Michael Dennis**[1]   **Shane Legg**[2]   **Stuart Russell**[1]   **Jan Leike**[3]*

[1]UC Berkeley   [2]DeepMind   [3]OpenAI
`gleave@berkeley.edu`

## ABSTRACT

For many tasks, the reward function is inaccessible to introspection or too complex to be specified procedurally, and must instead be learned from user data. Prior work has evaluated learned reward functions by evaluating policies optimized for the learned reward. However, this method cannot distinguish between the learned reward function failing to reflect user preferences and the policy optimization process failing to optimize the learned reward. Moreover, this method can only tell us about behavior in the evaluation environment, but the reward may incentivize very different behavior in even a slightly different deployment environment. To address these problems, we introduce the *Equivalent-Policy Invariant Comparison (EPIC)* distance to quantify the difference between two reward functions directly, without a policy optimization step. We prove EPIC is invariant on an equivalence class of reward functions that always induce the same optimal policy. Furthermore, we find EPIC can be efficiently approximated and is more robust than baselines to the choice of coverage distribution. Finally, we show that EPIC distance bounds the regret of optimal policies even under different transition dynamics, and we confirm empirically that it predicts policy training success. Our source code is available at `https://github.com/HumanCompatibleAI/evaluating-rewards`.

## 1   INTRODUCTION

Reinforcement learning (RL) has reached or surpassed human performance in many domains with clearly defined reward functions, such as games [20; 15; 23] and narrowly scoped robotic manipulation tasks [16]. Unfortunately, the reward functions for most real-world tasks are difficult or impossible to specify procedurally. Even a task as simple as peg insertion from pixels has a non-trivial reward function that must usually be learned [22, IV.A]. Tasks involving human interaction can have far more complex reward functions that users may not even be able to introspect on. These challenges have inspired work on learning a reward function, whether from demonstrations [13; 17; 26; 8; 3], preferences [1; 25; 6; 18; 27] or both [10; 4].

Prior work has usually evaluated the learned reward function $\hat{R}$ using the "rollout method": training a policy $\pi_{\hat{R}}$ to optimize $\hat{R}$ and then examining rollouts from $\pi_{\hat{R}}$. Unfortunately, using RL to compute $\pi_{\hat{R}}$ is often computationally expensive. Furthermore, the method produces *false negatives* when the reward $\hat{R}$ matches user preferences but the RL algorithm fails to optimize with respect to $\hat{R}$.

The rollout method also produces *false positives*. Of the many reward functions that induce the desired rollout in a given environment, only a small subset align with the user's preferences. For example, suppose the agent can reach states $\{A, B, C\}$. If the user prefers $A > B > C$, but the agent instead learns $A > C > B$, the agent will still go to the correct state $A$. However, if the initial state distribution or transition dynamics change, misaligned rewards may induce undesirable policies. For example, if $A$ is no longer reachable at deployment, the previously reliable agent would misbehave by going to the least-favoured state $C$.

We propose instead to evaluate learned rewards via their distance from other reward functions, and summarize our desiderata for reward function distances in Table 1. For benchmarks, it is usually possible to directly compare a learned reward $\hat{R}$ to the true reward function $R$. Alternatively, benchmark creators can train a "proxy" reward function from a large human data set. This proxy can then be used as a stand-in for the true reward $R$ when evaluating algorithms trained on a different or smaller data set.

---

*Work partially conducted while at DeepMind.

Table 1: Summary of the desiderata satisfied by each reward function distance. **Key** – the distance is: a *pseudometric* (section 3); *invariant* to potential shaping [14] and positive rescaling (section 3); a computationally *efficient* approximation achieving low error (section 6.1); *robust* to the choice of coverage distribution (section 6.2); and *predictive* of the similarity of the trained policies (section 6.3).

| Distance | Pseudometric | Invariant | Efficient | Robust | Predictive |
|---|---|---|---|---|---|
| EPIC | ✓ | ✓ | ✓ | ✓ | ✓ |
| NPEC | ✗ | ✓ | ✗ | ✗ | ✓ |
| ERC | ✓ | ✗ | ✓ | ✗ | ✓ |

Comparison with a ground-truth reward function is rarely possible outside of benchmarks. However, even in this challenging case, comparisons can at least be used to cluster reward models trained using different techniques or data. Larger clusters are more likely to be correct, since multiple methods arrived at a similar result. Moreover, our regret bound (Theorem 4.9) suggests we could use interpretability methods [12] on one model and get some guarantees for models in the same cluster.

We introduce the *Equivalent-Policy Invariant Comparison (EPIC)* distance that meets all the criteria in Table 1. We believe EPIC is the first method to quantitatively evaluate reward functions without training a policy. EPIC (section 4) canonicalizes the reward functions' potential-based shaping [14], then takes the correlation between the canonical rewards over a *coverage distribution* $\mathcal{D}$ of transitions. We also introduce baselines *NPEC* and *ERC* (section 5) which partially satisfy the criteria.

EPIC works best when $\mathcal{D}$ has support on all realistic transitions. We achieve this in our experiments by using uninformative priors, such as rollouts of a policy taking random actions. Moreover, we find EPIC is robust to the exact choice of distribution $\mathcal{D}$, producing similar results across a range of distributions, whereas ERC and especially NPEC are highly sensitive to the choice of $\mathcal{D}$ (section 6.2).

Moreover, low EPIC distance between a learned reward $\hat{R}$ and the true reward $R$ predicts low regret. That is, the policies $\pi_{\hat{R}}$ and $\pi_R$ optimized for $\hat{R}$ and $R$ obtain similar returns under $R$. Theorem 4.9 bounds the regret even in unseen environments; by contrast, the rollout method can only determine regret in the evaluation environment. We also confirm this result empirically (section 6.3).

## 2 RELATED WORK

There exist a variety of methods to learn reward functions. Inverse reinforcement learning (IRL) [13] is a common approach that works by inferring a reward function from demonstrations. The IRL problem is inherently underconstrained: many different reward functions lead to the same demonstrations. Bayesian IRL [17] handles this ambiguity by inferring a posterior over reward functions. By contrast, Maximum Entropy IRL [26] selects the highest entropy reward function consistent with the demonstrations; this method has scaled to high-dimensional environments [7; 8].

An alternative approach is to learn from *preference comparisons* between two trajectories [1; 25; 6; 18]. T-REX [4] is a hybrid approach, learning from a *ranked* set of demonstrations. More directly, Cabi et al. [5] learn from "sketches" of cumulative reward over an episode.

To the best of our knowledge, there is no prior work that focuses on evaluating reward functions directly. The most closely related work is Ng et al. [14], identifying reward transformations guaranteed to not change the optimal policy. However, a variety of ad-hoc methods have been developed to evaluate reward functions. The rollout method – evaluating rollouts of a policy trained on the learned reward – is evident in the earliest work on IRL [13]. Fu et al. [8] refined the rollout method by testing on a transfer environment, inspiring our experiment in section 6.3. Recent work has compared reward functions by scatterplotting returns [10; 4], inspiring our ERC baseline (section 5.1).

## 3 BACKGROUND

This section introduces material needed for the distances defined in subsequent sections. We start by introducing the *Markov Decision Process (MDP)* formalism, then describe when reward functions induce the same optimal policies in an MDP, and finally define the notion of a distance *metric*.

**Definition 3.1.** *A* Markov Decision Process (MDP) $M = (\mathcal{S}, \mathcal{A}, \gamma, d_0, \mathcal{T}, R)$ *consists of a set of states* $\mathcal{S}$ *and a set of actions* $\mathcal{A}$; *a discount factor* $\gamma \in [0, 1]$; *an initial state distribution* $d_0(s)$; *a transition distribution* $\mathcal{T}(s' \mid s, a)$ *specifying the probability of transitioning to* $s'$ *from* $s$ *after taking action* $a$; *and a reward function* $R(s, a, s')$ *specifying the reward upon taking action* $a$ *in state* $s$ *and transitioning to state* $s'$.

A trajectory $\tau = (s_0, a_0, s_1, a_1, \cdots)$ consists of a sequence of states $s_i \in \mathcal{S}$ and actions $a_i \in \mathcal{A}$. The *return* on a trajectory is defined as the sum of discounted rewards, $g(\tau; R) = \sum_{t=0}^{|\tau|} \gamma^t R(s_t, a_t, s_{t+1})$, where the length of the trajectory $|\tau|$ may be infinite.

In the following, we assume a discounted ($\gamma < 1$) infinite-horizon MDP. The results can be generalized to undiscounted ($\gamma = 1$) MDPs subject to regularity conditions needed for convergence.

A *stochastic policy* $\pi(a \mid s)$ assigns probabilities to taking action $a \in \mathcal{A}$ in state $s \in \mathcal{S}$. The objective of an MDP is to find a policy $\pi$ that maximizes the expected return $G(\pi) = \mathbb{E}_{\tau(\pi)}[g(\tau; R)]$, where $\tau(\pi)$ is a trajectory generated by sampling the initial state $s_0$ from $d_0$, each action $a_t$ from the policy $\pi(a_t \mid s_t)$ and successor states $s_{t+1}$ from the transition distribution $\mathcal{T}(s_{t+1} \mid s_t, a_t)$. An MDP $M$ has a set of optimal policies $\pi^*(M)$ that maximize the expected return, $\pi^*(M) = \arg\max_\pi G(\pi)$.

In this paper, we consider the case where we only have access to an MDP\R, $M^- = (\mathcal{S}, \mathcal{A}, \gamma, d_0, \mathcal{T})$. The unknown reward function $R$ must be learned from human data. Typically, only the state space $\mathcal{S}$, action space $\mathcal{A}$ and discount factor $\gamma$ are known exactly, with the initial state distribution $d_0$ and transition dynamics $\mathcal{T}$ only observable from interacting with the environment $M^-$. Below, we describe an equivalence class whose members are guaranteed to have the same optimal policy set in *any* MDP\R $M^-$ with fixed $\mathcal{S}$, $\mathcal{A}$ and $\gamma$ (allowing the unknown $\mathcal{T}$ and $d_0$ to take arbitrary values).

**Definition 3.2.** *Let* $\gamma \in [0, 1]$ *be the discount factor, and* $\Phi : \mathcal{S} \to \mathbb{R}$ *a real-valued function. Then* $R(s, a, s') = \gamma\Phi(s') - \Phi(s)$ *is a* potential shaping *reward, with* potential $\Phi$ *[14].*

**Definition 3.3** (Reward Equivalence). *We define two bounded reward functions* $R_A$ *and* $R_B$ *to be equivalent,* $R_A \equiv R_B$, *for a fixed* $(\mathcal{S}, \mathcal{A}, \gamma)$ *if and only if there exists a constant* $\lambda > 0$ *and a bounded potential function* $\Phi : \mathcal{S} \to \mathbb{R}$ *such that for all* $s, s' \in \mathcal{S}$ *and* $a \in \mathcal{A}$:

$$R_B(s, a, s') = \lambda R_A(s, a, s') + \gamma\Phi(s') - \Phi(s). \tag{1}$$

**Proposition 3.4.** *The binary relation* $\equiv$ *is an equivalence relation. Let* $R_A, R_B, R_C : \mathcal{S} \times \mathcal{A} \times \mathcal{S} \to \mathbb{R}$ *be bounded reward functions. Then* $\equiv$ *is reflexive,* $R_A \equiv R_A$; *symmetric,* $R_A \equiv R_B$ *implies* $R_B \equiv R_A$; *and transitive,* $(R_A \equiv R_B) \wedge (R_B \equiv R_C)$ *implies* $R_A \equiv R_C$.

*Proof.* See section A.3.1 in supplementary material. □

The expected return of potential shaping $\gamma\Phi(s') - \Phi(s)$ on a trajectory segment $(s_0, \cdots, s_T)$ is $\gamma^T\Phi(s_T) - \Phi(s_0)$. The first term $\gamma^T\Phi(s_T) \to 0$ as $T \to \infty$, while the second term $\Phi(s_0)$ only depends on the initial state, and so potential shaping does not change the set of optimal policies. Moreover, any additive transformation that is not potential shaping will, for some reward $R$ and transition distribution $\mathcal{T}$, produce a set of optimal policies that is disjoint from the original [14].

The set of optimal policies is invariant to constant shifts $c \in \mathbb{R}$ in the reward, however this can already be obtained by shifting $\Phi$ by $\frac{c}{\gamma-1}$.* Scaling a reward function by a positive factor $\lambda > 0$ scales the expected return of all trajectories by $\lambda$, also leaving the set of optimal policies unchanged.

If $R_A \equiv R_B$ for some fixed $(\mathcal{S}, \mathcal{A}, \gamma)$, then for any MDP\R $M^- = (\mathcal{S}, \mathcal{A}, \gamma, d_0, \mathcal{T})$ we have $\pi^*((M^-, R_A)) = \pi^*((M^-, R_B))$, where $(M^-, R)$ denotes the MDP specified by $M^-$ with reward function $R$. In other words, $R_A$ and $R_B$ induce the same optimal policies for all initial state distributions $d_0$ and transition dynamics $\mathcal{T}$.

**Definition 3.5.** *Let* $X$ *be a set and* $d : X \times X \to [0, \infty)$ *a function.* $d$ *is a* premetric *if* $d(x, x) = 0$ *for all* $x \in X$. $d$ *is a* pseudometric *if, furthermore, it is symmetric,* $d(x, y) = d(y, x)$ *for all* $x, y \in X$; *and satisfies the triangle inequality,* $d(x, z) \leq d(x, y) + d(y, z)$ *for all* $x, y, z \in X$. $d$ *is a* metric *if, furthermore, for all* $x, y \in X$, $d(x, y) = 0 \implies x = y$.

We wish for $d(R_A, R_B) = 0$ whenever the rewards are equivalent, $R_A \equiv R_B$, even if they are not identical, $R_A \neq R_B$. This is forbidden in a metric but permitted in a pseudometric, while retaining

---

*Note constant shifts in the reward of an undiscounted MDP would cause the value function to diverge. Fortunately, the shaping $\gamma\Phi(s') - \Phi(s)$ is unchanged by constant shifts to $\Phi$ when $\gamma = 1$.

other guarantees such as symmetry and triangle inequality that a metric provides. Accordingly, a pseudometric is usually the best choice for a distance $d$ over reward functions.

## 4 COMPARING REWARD FUNCTIONS WITH EPIC

In this section we introduce the *Equivalent-Policy Invariant Comparison (EPIC)* pseudometric. This novel distance canonicalizes the reward functions' potential-based shaping, then compares the canonical representatives using Pearson correlation, which is invariant to scale. Together, this construction makes EPIC invariant on reward equivalence classes. See section A.3.2 for proofs.

We define the *canonically shaped reward* $C_{\mathcal{D}_\mathcal{S},\mathcal{D}_\mathcal{A}}(R)$ as an expectation over some arbitrary distributions $\mathcal{D}_\mathcal{S}$ and $\mathcal{D}_\mathcal{A}$ over states $\mathcal{S}$ and actions $\mathcal{A}$ respectively. This construction means that $C_{\mathcal{D}_\mathcal{S},\mathcal{D}_\mathcal{A}}(R)$ does not depend on the MDP's initial state distribution $d_0$ or transition dynamics $\mathcal{T}$. In particular, we may evaluate $R$ on transitions that are impossible in the training environment, since these may become possible in a deployment environment with a different $d_0$ or $\mathcal{T}$.

**Definition 4.1** (Canonically Shaped Reward). *Let $R : \mathcal{S} \times \mathcal{A} \times \mathcal{S} \to \mathbb{R}$ be a reward function. Given distributions $\mathcal{D}_\mathcal{S} \in \Delta(\mathcal{S})$ and $\mathcal{D}_\mathcal{A} \in \Delta(\mathcal{A})$ over states and actions, let $S$ and $S'$ be random variables independently sampled from $\mathcal{D}_\mathcal{S}$ and $A$ sampled from $\mathcal{D}_\mathcal{A}$. We define the* canonically shaped $R$ *to be:*

$$C_{\mathcal{D}_\mathcal{S},\mathcal{D}_\mathcal{A}}(R)(s,a,s') = R(s,a,s') + \mathbb{E}\left[\gamma R(s',A,S') - R(s,A,S') - \gamma R(S,A,S')\right]. \quad (2)$$

Informally, if $R'$ is shaped by potential $\Phi$, then increasing $\Phi(s)$ decreases $R'(s,a,s')$ but increases $\mathbb{E}\left[-R'(s,A,S')\right]$, canceling. Similarly, increasing $\Phi(s')$ increases $R'(s,a,s')$ but decreases $\mathbb{E}\left[\gamma R'(s',A,S')\right]$. Finally, $\mathbb{E}[\gamma R(S,A,S')]$ centers the reward, canceling constant shift.

**Proposition 4.2** (The Canonically Shaped Reward is Invariant to Shaping). *Let $R : \mathcal{S} \times \mathcal{A} \times \mathcal{S} \to \mathbb{R}$ be a reward function and $\Phi : \mathcal{S} \to \mathbb{R}$ a potential function. Let $\gamma \in [0,1]$ be a discount rate, and $\mathcal{D}_\mathcal{S} \in \Delta(\mathcal{S})$ and $\mathcal{D}_\mathcal{A} \in \Delta(\mathcal{A})$ be distributions over states and actions. Let $R'$ denote $R$ shaped by $\Phi$: $R'(s,a,s') = R(s,a,s') + \gamma\Phi(s') - \Phi(s)$. Then the canonically shaped $R'$ and $R$ are equal: $C_{\mathcal{D}_\mathcal{S},\mathcal{D}_\mathcal{A}}(R') = C_{\mathcal{D}_\mathcal{S},\mathcal{D}_\mathcal{A}}(R)$.*

Proposition 4.2 holds for arbitrary distributions $\mathcal{D}_\mathcal{S}$ and $\mathcal{D}_\mathcal{A}$. However, in the following Proposition we show that the potential shaping applied by the canonicalization $C_{\mathcal{D}_\mathcal{S},\mathcal{D}_\mathcal{A}}(R)$ is more influenced by perturbations to $R$ of transitions $(s,a,s')$ with high joint probability. This suggests choosing $\mathcal{D}_\mathcal{S}$ and $\mathcal{D}_\mathcal{A}$ to have broad support, making $C_{\mathcal{D}_\mathcal{S},\mathcal{D}_\mathcal{A}}(R)$ more robust to perturbations of any given transition.

**Proposition 4.3.** *Let $\mathcal{S}$ and $\mathcal{A}$ be finite, with $|\mathcal{S}| \geq 2$. Let $\mathcal{D}_\mathcal{S} \in \Delta(\mathcal{S})$ and $\mathcal{D}_\mathcal{A} \in \Delta(\mathcal{A})$. Let $R, \nu : \mathcal{S} \times \mathcal{A} \times \mathcal{S} \to \mathbb{R}$ be reward functions, with $\nu(s,a,s') = \lambda\mathbb{I}[(s,a,s') = (x,u,x')]$, $\lambda \in \mathbb{R}$, $x,x' \in \mathcal{S}$ and $u \in \mathcal{A}$. Let $\Phi_{\mathcal{D}_\mathcal{S},\mathcal{D}_\mathcal{A}}(R)(s,a,s') = C_{\mathcal{D}_\mathcal{S},\mathcal{D}_\mathcal{A}}(R)(s,a,s') - R(s,a,s')$. Then:*

$$\left\|\Phi_{\mathcal{D}_\mathcal{S},\mathcal{D}_\mathcal{A}}(R+\nu) - \Phi_{\mathcal{D}_\mathcal{S},\mathcal{D}_\mathcal{A}}(R)\right\|_\infty = \lambda\left(1 + \gamma\mathcal{D}_\mathcal{S}(x)\right)\mathcal{D}_\mathcal{A}(u)\mathcal{D}_\mathcal{S}(x'). \quad (3)$$

We have canonicalized potential shaping; next, we compare the rewards in a scale-invariant manner.

**Definition 4.4.** *The* Pearson distance *between random variables $X$ and $Y$ is defined by the expression $D_\rho(X,Y) = \sqrt{1 - \rho(X,Y)}/\sqrt{2}$, where $\rho(X,Y)$ is the Pearson correlation between $X$ and $Y$.*

**Lemma 4.5.** *The Pearson distance $D_\rho$ is a pseudometric. Moreover, let $a,b \in (0,\infty)$, $c,d \in \mathbb{R}$ and $X,Y$ be random variables. Then it follows that $0 \leq D_\rho(aX+c, bY+d) = D_\rho(X,Y) \leq 1$.*

We can now define EPIC in terms of the Pearson distance between canonically shaped rewards.

**Definition 4.6** (Equivalent-Policy Invariant Comparison (EPIC) pseudometric). *Let $\mathcal{D}$ be some coverage distribution over transitions $s \xrightarrow{a} s'$. Let $S, A, S'$ be random variables jointly sampled from $\mathcal{D}$. Let $\mathcal{D}_\mathcal{S}$ and $\mathcal{D}_\mathcal{A}$ be some distributions over states $\mathcal{S}$ and $\mathcal{A}$ respectively. The* Equivalent-Policy Invariant Comparison (EPIC) *distance between reward functions $R_A$ and $R_B$ is:*

$$D_{\text{EPIC}}(R_A, R_B) = D_\rho\left(C_{\mathcal{D}_\mathcal{S},\mathcal{D}_\mathcal{A}}(R_A)(S,A,S'), C_{\mathcal{D}_\mathcal{S},\mathcal{D}_\mathcal{A}}(R_B)(S,A,S')\right). \quad (4)$$

**Theorem 4.7.** *The Equivalent-Policy Invariant Comparison distance is a pseudometric.*

Since EPIC is a pseudometric, it satisfies the triangle inequality. To see why this is useful, consider an environment with an expensive to evaluate ground-truth reward $R$. Directly comparing many learned

rewards $\hat{R}$ to $R$ might be prohibitively expensive. We can instead pay a one-off cost: query $R$ a finite number of times and infer a proxy reward $R_P$ with $D_{\text{EPIC}}(R, R_P) \leq \epsilon$. The triangle inequality allows us to evaluate $\hat{R}$ via comparison to $R_P$, since $D_{\text{EPIC}}(\hat{R}, R) \leq D_{\text{EPIC}}(\hat{R}, R_P) + \epsilon$. This is particularly useful for benchmarks, which can be expensive to build but should be cheap to use.

**Theorem 4.8.** *Let* $R_A$, $R'_A$, $R_B, R'_B : \mathcal{S} \times \mathcal{A} \times \mathcal{S} \to \mathbb{R}$ *be reward functions such that* $R'_A \equiv R_A$ *and* $R'_B \equiv R_B$. *Then* $0 \leq D_{\text{EPIC}}(R'_A, R'_B) = D_{\text{EPIC}}(R_A, R_B) \leq 1$.

The following is our main theoretical result, showing that $D_{\text{EPIC}}(R_A, R_B)$ distance gives an upper bound on the difference in returns under *either* $R_A$ or $R_B$ between optimal policies $\pi^*_{R_A}$ and $\pi^*_{R_B}$. In other words, EPIC bounds the regret under $R_A$ of using $\pi^*_{R_B}$ instead of $\pi^*_{R_A}$. Moreover, by symmetry $D_{\text{EPIC}}(R_A, R_B)$ also bounds the regret under $R_B$ of using $\pi^*_{R_A}$ instead of $\pi^*_{R_B}$.

**Theorem 4.9.** *Let* $M$ *be a* $\gamma$-*discounted MDP\R with finite state and action spaces* $\mathcal{S}$ *and* $\mathcal{A}$. *Let* $R_A, R_B : \mathcal{S} \times \mathcal{A} \times \mathcal{S} \to \mathbb{R}$ *be rewards, and* $\pi^*_A, \pi^*_B$ *be respective optimal policies. Let* $\mathcal{D}_\pi(t, s_t, a_t, s_{t+1})$ *denote the distribution over transitions* $\mathcal{S} \times \mathcal{A} \times \mathcal{S}$ *induced by policy* $\pi$ *at time* $t$, *and* $\mathcal{D}(s, a, s')$ *be the coverage distribution used to compute* $D_{\text{EPIC}}$. *Suppose there exists* $K > 0$ *such that* $K\mathcal{D}(s_t, a_t, s_{t+1}) \geq \mathcal{D}_\pi(t, s_t, a_t, s_{t+1})$ *for all times* $t \in \mathbb{N}$, *triples* $(s_t, a_t, s_{t+1}) \in \mathcal{S} \times \mathcal{A} \times \mathcal{S}$ *and policies* $\pi \in \{\pi^*_A, \pi^*_B\}$. *Then the regret under* $R_A$ *from executing* $\pi^*_B$ *instead of* $\pi^*_A$ *is at most*

$$G_{R_A}(\pi^*_A) - G_{R_A}(\pi^*_B) \leq 16K\|R_A\|_2 \, (1 - \gamma)^{-1} \, D_{\text{EPIC}}(R_A, R_B),$$

*where* $G_R(\pi)$ *is the return of policy* $\pi$ *under reward* $R$.

We generalize the regret bound to continuous spaces in theorem A.16 via a Lipschitz assumption, with Wasserstein distance replacing $K$. Importantly, the returns of $\pi^*_A$ and $\pi^*_B$ **converge** as $D_{\text{EPIC}}(R_A, R_B) \to 0$ in both cases, no matter which reward function you evaluate on.

The key assumption is that the coverage distribution $\mathcal{D}$ has adequate support for transitions occurring in rollouts of $\pi^*_A$ and $\pi^*_B$. The bound is tightest when $\mathcal{D}$ is similar to $\mathcal{D}_{\pi^*_A}$ and $\mathcal{D}_{\pi^*_B}$. However, computing $\pi^*_A$ and $\pi^*_B$ is often intractable. The MDP $M$ may be unknown, such as when making predictions about an unseen deployment environment. Even when $M$ is known, RL is computationally expensive and may fail to converge in non-trivial environments.

In finite cases, a uniform $\mathcal{D}$ satisfies the requirements with $K \leq |\mathcal{S}|^2|\mathcal{A}|$. In general, it is best to choose $\mathcal{D}$ to have broad coverage over plausible transitions. Broad coverage ensures adequate support for $\mathcal{D}_{\pi^*_A}$ and $\mathcal{D}_{\pi^*_B}$. But excluding transitions that are unlikely or impossible to occur leads to tighter regret bounds due to a smaller $K$ (finite case) or Wasserstein distance (continuous case).

While EPIC upper bounds policy regret, it does not lower bound it. In fact, no reward distance can lower bound regret in arbitrary environments. For example, suppose the deployment environment transitions to a randomly chosen state independent of the action taken. In this case, all policies obtain the same expected return, so the policy regret is always zero, regardless of the reward functions.

To demonstrate EPIC's properties, we compare the gridworld reward functions from Figure 1, reporting the distances between all reward pairs in Figure A.2. `Dense` is a rescaled and shaped version of `Sparse`, despite looking dissimilar at first glance, so $D_{\text{EPIC}}(\text{Sparse}, \text{Dense}) = 0$. By contrast, $D_{\text{EPIC}}(\text{Path}, \text{Cliff}) = 0.27$. In *deterministic* gridworlds, `Path` and `Cliff` have the same optimal policy, so the rollout method could wrongly conclude they are equivalent. But in fact the rewards are fundamentally different: when there is a significant risk of "slipping" in the wrong direction the optimal policy for `Cliff` walks along the top instead of the middle row, incurring a $-1$ penalty to avoid the risk of falling into the $-4$ "cliff".

For this example, we used state and action distributions $\mathcal{D}_\mathcal{S}$ and $\mathcal{D}_\mathcal{A}$ uniform over $\mathcal{S}$ and $\mathcal{A}$, and coverage distribution $\mathcal{D}$ uniform over state-action pairs $(s, a)$, with $s'$ deterministically computed. It is important these distributions have adequate support. As an extreme example, if $\mathcal{D}_\mathcal{S}$ and $\mathcal{D}$ have no support for a particular state then the reward of that state has no effect on the distance. We can compute EPIC exactly in a tabular setting, but in general use a sample-based approximation (section A.1.1).

# 5 BASELINE APPROACHES FOR COMPARING REWARD FUNCTIONS

Given the lack of established methods, we develop two alternatives as baselines: Episode Return Correlation (ERC) and Nearest Point in Equivalence Class (NPEC).

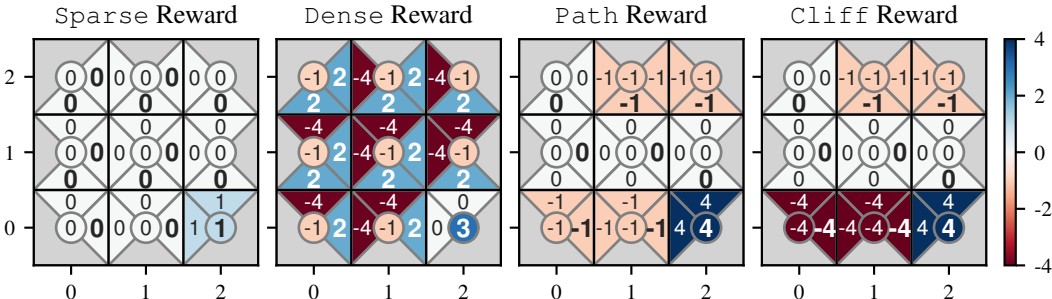

Figure 1: Heatmaps of four reward functions for a $3 \times 3$ gridworld. `Sparse` and `Dense` look different but are actually equivalent with $D_{\text{EPIC}}(\texttt{Sparse}, \texttt{Dense}) = 0$. By contrast, the optimal policies for `Path` and `Cliff` are the same if the gridworld is deterministic but different if it is "slippery". EPIC recognizes this difference with $D_{\text{EPIC}}(\texttt{Path}, \texttt{Cliff}) = 0.27$. **Key**: Reward $R(s, s')$ for moving from $s$ to $s'$ is given by the **triangular wedge** in cell $s$ that is adjacent to cell $s'$. $R(s, s)$ is given by the **central circle** in cell $s$. Optimal action(s) (deterministic, infinite horizon, discount $\gamma = 0.99$) have **bold** labels. See Figure A.2 for the distances between all reward pairs.

## 5.1 EPISODE RETURN CORRELATION (ERC)

The goal of an MDP is to maximize expected episode return, so it is natural to compare reward functions by the returns they induce. If the return of a reward function $R_A$ is a positive affine transformation of another reward $R_B$, then $R_A$ and $R_B$ have the same set of optimal policies. This suggests using Pearson distance, which is invariant to positive affine transformations.

**Definition 5.1** (Episode Return Correlation (ERC) pseudometric). *Let $\mathcal{D}$ be some distribution over trajectories. Let $E$ be a random variable sampled from $\mathcal{D}$. The* Episode Return Correlation *distance between reward functions $R_A$ and $R_B$ is the Pearson distance between their episode returns on $\mathcal{D}$, $D_{\text{ERC}}(R_A, R_B) = D_\rho(g(E; R_A), g(E; R_B))$.*

Prior work has produced scatter plots of the return of $R_A$ against $R_B$ over episodes [4, Figure 3] and fixed-length segments [10, section D]. ERC is the Pearson distance of such plots, so is a natural baseline. We approximate ERC by the correlation of episode returns on a finite collection of rollouts.

ERC is invariant to shaping when the initial state $s_0$ and terminal state $s_T$ are fixed. Let $R$ be a reward function and $\Phi$ a potential function, and define the shaped reward $R'(s, a, s') = R(s, a, s') + \gamma\Phi(s') - \Phi(s)$. The return under the shaped reward on a trajectory $\tau = (s_0, a_0, \cdots, s_T)$ is $g(\tau; R') = g(\tau; R) + \gamma^T\Phi(s_T) - \Phi(s_0)$. Since $s_0$ and $s_T$ are fixed, $\gamma^T\Phi(s_T) - \Phi(s_0)$ is constant. It follows that ERC is invariant to shaping, as Pearson distance is invariant to constant shifts. In fact, for infinite-horizon discounted MDPs only $s_0$ needs to be fixed, since $\gamma^T\Phi(s_T) \to 0$ as $T \to \infty$.

However, if the initial state $s_0$ is stochastic, then the ERC distance can take on arbitrary values under shaping. Let $R_A$ and $R_B$ be two arbitrary reward functions. Suppose that there are at least two distinct initial states, $s_X$ and $s_Y$, with non-zero measure in $\mathcal{D}$. Choose potential $\Phi(s) = 0$ everywhere except $\Phi(s_X) = \Phi(s_Y) = c$, and let $R'_A$ and $R'_B$ denote $R_A$ and $R_B$ shaped by $\Phi$. As $c \to \infty$, the correlation $\rho(g(E; R'_A), g(E; R'_B)) \to 1$. This is since the relative difference tends to zero, even though $g(E; R'_A)$ and $g(E; R'_B)$ continue to have the same absolute difference as $c$ varies. Consequently, the ERC pseudometric $D_{\text{ERC}}(R'_A, R'_B) \to 0$ as $c \to \infty$. By an analogous argument, setting $\Phi(s_X) = c$ and $\Phi(s_Y) = -c$ gives $D_{\text{ERC}}(R'_A, R'_B) \to 1$ as $c \to \infty$.

## 5.2 NEAREST POINT IN EQUIVALENCE CLASS (NPEC)

NPEC takes the minimum $L^p$ distance between equivalence classes. See section A.3.3 for proofs.

**Definition 5.2** ($L^p$ distance). *Let $\mathcal{D}$ be a coverage distribution over transitions $s \xrightarrow{a} s'$ and let $p \geq 1$ be a power. The $L^p$ distance between reward functions $R_A$ and $R_B$ is the $L^p$ norm of their difference: $D_{L^p, \mathcal{D}}(R_A, R_B) = \left( \mathbb{E}_{s, a, s' \sim \mathcal{D}} \left[ |R_A(s, a, s') - R_B(s, a, s')|^p \right] \right)^{1/p}.$*

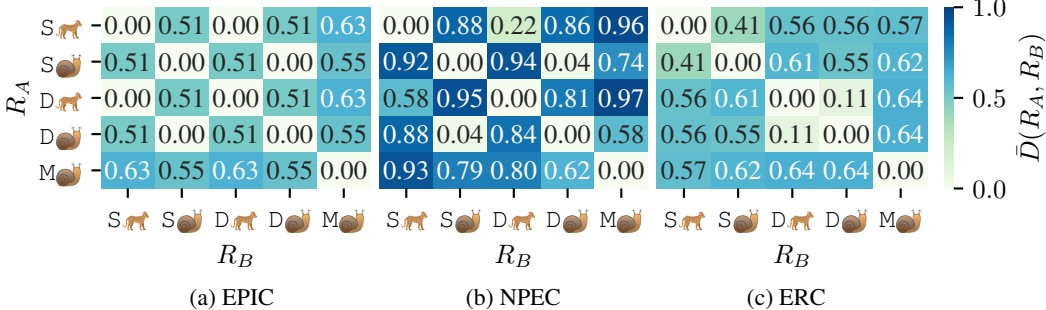

Figure 2: Approximate distances between hand-designed reward functions in `PointMass`, where the agent moves on a line trying to reach the origin. EPIC correctly assigns $0$ distance between equivalent rewards such as $(\text{D}, \text{S})$ while $D_{\text{NPEC}}(\text{D}, \text{S}) = 0.58$ and $D_{\text{ERC}}(\text{D}, \text{S}) = 0.56$. The coverage distribution $\mathcal{D}$ is sampled from rollouts of a policy $\pi_{\text{uni}}$ taking actions uniformly at random. **Key**: The agent has position $x \in \mathbb{R}$, velocity $\dot{x} \in \mathbb{R}$ and can accelerate $\ddot{x} \in \mathbb{R}$, producing future position $x' \in \mathbb{R}$. quadratic penalty on control $\ddot{x}^2$, no control penalty. S is $\text{Sparse}(x) = \mathbb{1}[|x| < 0.05]$, D is shaped $\text{Dense}(x, x') = \text{Sparse}(x) + |x'| - |x|$, while M is $\text{Magnitude}(x) = -|x|$.

The $L^p$ distance is affected by potential shaping and positive rescaling that do not change the optimal policy. A natural solution is to take the distance from the *nearest point* in the equivalence class: $D^U_{\text{NPEC}}(R_A, R_B) = \inf_{R'_A \equiv R_A} D_{L^p, \mathcal{D}}(R'_A, R_B)$. Unfortunately, $D^U_{\text{NPEC}}$ is sensitive to $R_B$'s scale.

It is tempting to instead take the infimum over both arguments of $D_{L^p, \mathcal{D}}$. However, $\inf_{R'_A \equiv R_A, R'_B \equiv R_B} D_{L^p, \mathcal{D}}(R'_A, R_B) = 0$ since all equivalence classes come arbitrarily close to the origin in $L^p$ space. Instead, we fix this by normalizing $D^U_{\text{NPEC}}$.

**Definition 5.3.** NPEC *is defined by* $D_{\text{NPEC}}(R_A, R_B) = D^U_{\text{NPEC}}(R_A, R_B) / D^U_{\text{NPEC}}(\text{Zero}, R_B)$ *when* $D^U_{\text{NPEC}}(\text{Zero}, R_B) \neq 0$, *and is otherwise given by* $D_{\text{NPEC}}(R_A, R_B) = 0$.

If $D^U_{\text{NPEC}}(\text{Zero}, R_B) = 0$ then $D^U_{\text{NPEC}}(R_A, R_B) = 0$ since $R_A$ can be scaled arbitrarily close to Zero. Since all policies are optimal for $R \equiv \text{Zero}$, we choose $D_{\text{NPEC}}(R_A, R_B) = 0$ in this case.

**Theorem 5.4.** $D_{\text{NPEC}}$ *is a premetric on the space of bounded reward functions. Moreover, let* $R_A, R_A', R_B, R_B' : \mathcal{S} \times \mathcal{A} \times \mathcal{S} \to \mathbb{R}$ *be bounded reward functions such that* $R_A \equiv R_A'$ *and* $R_B \equiv R_B'$. *Then* $0 \leq D_{\text{NPEC}}(R_A', R_B') = D_{\text{NPEC}}(R_A, R_B) \leq 1$.

Note that $D_{\text{NPEC}}$ may not be symmetric and so is not, in general, a pseudometric: see proposition A.3. The infimum in $D^U_{\text{NPEC}}$ can be computed exactly in a tabular setting, but in general we must approximate it using gradient descent. This gives an upper bound for $D^U_{\text{NPEC}}$, but the quotient of upper bounds $D_{\text{NPEC}}$ may be too low or too high. See section A.1.2 for details of the approximation.

## 6 EXPERIMENTS

We evaluate EPIC and the baselines ERC and NPEC in a variety of continuous control tasks. In section 6.1, we compute the distance between hand-designed reward functions, finding EPIC to be the most reliable. NPEC has substantial approximation error, and ERC sometimes erroneously assigns high distance to equivalent rewards. Next, in section 6.2 we show EPIC is robust to the exact choice of coverage distribution $\mathcal{D}$, whereas ERC and especially NPEC are highly sensitive to the choice of $\mathcal{D}$. Finally, in section 6.3 we find that the distance of learned reward functions to a ground-truth reward predicts the return obtained by policy training, even in an unseen test environment.

### 6.1 COMPARING HAND-DESIGNED REWARD FUNCTIONS

We compare procedurally specified reward functions in four tasks, finding that EPIC is more reliable than the baselines NPEC and ERC, and more computationally efficient than NPEC. Figure 2 presents results in the proof-of-concept `PointMass` task. The results for `Gridworld`, `HalfCheetah` and `Hopper`, in section A.2.4, are qualitatively similar.

Table 2: Low reward distance from the ground-truth (GT) in `PointMaze-Train` predicts high policy return even in unseen task `PointMaze-Test`. EPIC distance is robust to the choice of coverage distribution $\mathcal{D}$, with similar values across columns, while ERC and especially NPEC are sensitive to $\mathcal{D}$. **Center**: approximate distances ($1000\times$ scale) of reward functions from GT. The coverage distribution $\mathcal{D}$ is computed from rollouts in `PointMaze-Train` of: a uniform random policy $\pi_{\mathbf{uni}}$, an expert $\pi^*$ and a **Mix**ture of these policies. $\mathcal{D}_{\mathcal{S}}$ and $\mathcal{D}_{\mathcal{A}}$ are computed by marginalizing $\mathcal{D}$. **Right**: mean GT return over 9 seeds of RL training on the reward in `PointMaze-{Train,Test}`, and returns for AIRL's *generator* policy. **Confidence Intervals**: see Table A.7.

| Reward Function | $1000 \times D_{\mathbf{EPIC}}$ | | | $1000 \times D_{\mathbf{NPEC}}$ | | | $1000 \times D_{\mathbf{ERC}}$ | | | Episode Return | | |
| | $\pi_{\mathbf{uni}}$ | $\pi^*$ | Mix | $\pi_{\mathbf{uni}}$ | $\pi^*$ | Mix | $\pi_{\mathbf{uni}}$ | $\pi^*$ | Mix | Gen. | Train | Test |
|---|---|---|---|---|---|---|---|---|---|---|---|---|
| GT | 0.06 | 0.05 | 0.04 | 0.04 | 3.17 | 0.01 | 0.00 | 0.00 | 0.00 | — | $-5.19$ | $-6.59$ |
| Regress | 35.8 | 33.7 | 26.1 | 1.42 | 38.9 | 0.35 | 9.99 | 90.7 | 2.43 | — | $-5.47$ | $-6.30$ |
| Pref | 68.7 | 100 | 56.8 | 8.51 | 1333 | 9.74 | 24.9 | 360 | 19.6 | — | $-5.57$ | $-5.04$ |
| AIRL SO | 572 | 520 | 404 | 817 | 2706 | 488 | 549 | 523 | 240 | $-5.43$ | $-27.3$ | $-22.7$ |
| AIRL SA | 776 | 930 | 894 | 1067 | 2040 | 1039 | 803 | 722 | 964 | $-5.05$ | $-30.7$ | $-29.0$ |
| Mirage | 17.0 | 0.05 | 397 | 0.68 | 6.30 | 597 | 35.3 | <0.01 | 166 | — | $-30.4$ | $-29.1$ |

In `PointMass` the agent can accelerate $\ddot{x}$ left or right on a line. The reward functions include (🐴) or exclude (🐎) a quadratic penalty $\ddot{x}^2$. The sparse reward (S) gives a reward of 1 in the region $\pm0.05$ from the origin. The dense reward (D) is a shaped version of the sparse reward. The magnitude reward (M) is the negative distance of the agent from the origin.

We find that EPIC correctly identifies the equivalent reward pairs (S🐴-D🐴 and S🐎-D🐎) with estimated distance $< 1 \times 10^{-3}$. By contrast, NPEC has substantial approximation error: $D_{\mathrm{NPEC}}(\text{D}🐎, \text{S}🐎) = 0.58$. Similarly, $D_{\mathrm{ERC}}(\text{D}🐎, \text{S}🐎) = 0.56$ due to ERC's erroneous handling of stochastic initial states. Moreover, NPEC is computationally inefficient: Figure 2(b) took 31 hours to compute. By contrast, the figures for EPIC and ERC were generated in less than two hours, and a lower precision approximation of EPIC finishes in just 17 seconds (see section A.2.6).

## 6.2 SENSITIVITY OF REWARD DISTANCE TO COVERAGE DISTRIBUTION

Reward distances should be robust to the choice of coverage distribution $\mathcal{D}$. In Table 2 (center), we report distances from the ground-truth reward (GT) to reward functions (rows) across coverage distributions $\mathcal{D} \in \{\pi_{\mathrm{uni}}, \pi^*, \text{Mix}\}$ (columns). We find EPIC is fairly robust to the choice of $\mathcal{D}$ with a similar ratio between rows in each column $\mathcal{D}$. By contrast, ERC and especially NPEC are substantially more sensitive to the choice of $\mathcal{D}$.

We evaluate in the `PointMaze` MuJoCo task from Fu et al. [8], where a point mass agent must navigate around a wall to reach a goal. The coverage distributions $\mathcal{D}$ are induced by rollouts from three different policies: $\pi_{\mathrm{uni}}$ takes actions uniformly at random, producing broad support over transitions; $\pi^*$ is an expert policy, yielding a distribution concentrated around the goal; and Mix is a mixture of the two. In EPIC, $\mathcal{D}_{\mathcal{S}}$ and $\mathcal{D}_{\mathcal{A}}$ are marginalized from $\mathcal{D}$ and so also vary with $\mathcal{D}$.

We evaluate four reward learning algorithms: Regression onto reward labels [*target* method from 6, section 3.3], Preference comparisons on trajectories [6], and adversarial IRL with a state-only (AIRL SO) and state-action (AIRL SA) reward model [8]. All models are trained using synthetic data from an oracle with access to the ground-truth; see section A.2.2 for details.

We find EPIC is robust to varying $\mathcal{D}$ when comparing the learned reward models: the distance varies by less than $2\times$, and the ranking between the reward models is the same across coverage distributions. By contrast, NPEC is highly sensitive to $\mathcal{D}$: the ratio of AIRL SO (817) to Pref (8.51) is $96:1$ under $\pi_{\mathrm{uni}}$ but only $2:1$ (2706 : 1333) under $\pi^*$. ERC lies somewhere in the middle: the ratio is $22:1$ (549 : 24.9) under $\pi_{\mathrm{uni}}$ and $3:2$ (523 : 360) under $\pi^*$.

We evaluate the effect of pathological choices of coverage distribution $\mathcal{D}$ in Table A.8. For example, **Ind** independently samples states and next states, giving physically impossible transitions, while **Jail** constrains rollouts to a tiny region excluding the goal. We find that the ranking of EPIC changes in only one distribution, whilst the ranking of NPEC changes in two cases and ERC changes in all cases.

However, we do find that EPIC is sensitive to $\mathcal{D}$ on `Mirage`, a reward function we explicitly designed to break these methods. `Mirage` assigns a larger reward when close to a "mirage" state than when at the true goal, but is identical to `GT` at all other points. The "mirage" state is rarely visited by random exploration $\pi_{\text{uni}}$ as it is far away and on the opposite side of the wall from the agent. The expert policy $\pi^*$ is even less likely to visit it, as it is not on or close to the optimal path to the goal. As a result, the EPIC distance from `Mirage` to `GT` (Table 2, bottom row) is small under $\pi_{\text{uni}}$ and $\pi^*$.

In general, any black-box method for assessing reward models – including the rollout method – only has predictive power on transitions visited during testing. Fortunately, we can achieve a broad support over states with `Mix`: it often navigates around the wall due to $\pi^*$, but strays from the goal thanks to $\pi_{\text{uni}}$. As a result, EPIC under `Mix` correctly infers that `Mirage` is far from the ground-truth `GT`.

These empirical results support our theoretically inspired recommendation from section 4: "in general, it is best to choose $\mathcal{D}$ to have broad coverage over plausible transitions." Distributions such as $\pi^*$ are too narrow, assigning coverage only on a direct path from the initial state to the goal. Very broad distributions such as **Ind** waste probability mass on impossible transitions like teleporting. Distributions like `Mix` strike the right balance between these extremes.

### 6.3 PREDICTING POLICY PERFORMANCE FROM REWARD DISTANCE

We find that low distance from the ground-truth reward `GT` (Table 2, center) predicts high `GT` return (Table 2, right) of policies optimized for that reward. Moreover, the distance is predictive of return not just in `PointMaze-Train` where the reward functions were trained and evaluated in, but also in the unseen variant `PointMaze-Test`. This is despite the two variants differing in the position of the wall, such that policies for `PointMaze-Train` run directly into the wall in `PointMaze-Test`.

Both `Regress` and `Pref` achieve very low distances at convergence, producing near-expert policy performance. The `AIRL SO` and `AIRL SA` models have reward distances an order of magnitude higher and poor policy performance. Yet intriguingly, the *generator* policies for `AIRL SO` and `AIRL SA` – trained simultaneously with the reward – perform reasonably in `PointMaze-Train`. This suggests the learned rewards are reasonable on the subset of transitions taken by the generator policy, yet fail to transfer to the different transitions taken by a policy being trained from scratch.

Figure A.6 shows reward distance and policy regret during reward model training. The lines all closely track each other, showing that the distance to `GT` is highly correlated with policy regret for intermediate reward checkpoints as well as at convergence. `Regress` and `Pref` converge quickly to low distance and low regret, while `AIRL SO` and `AIRL SA` are slower and more unstable.

## 7 CONCLUSION

Our novel EPIC distance compares reward functions directly, without training a policy. We have proved it is a pseudometric, is bounded and invariant to equivalent rewards, and bounds the regret of optimal policies (Theorems 4.7, 4.8 and 4.9). Empirically, we find EPIC correctly infers zero distance between equivalent reward functions that the NPEC and ERC baselines wrongly consider dissimilar. Furthermore, we find the distance of learned reward functions to the ground-truth reward predicts the return of policies optimized for the learned reward, even in unseen environments.

Standardized metrics are an important driver of progress in machine learning. Unfortunately, traditional policy-based metrics do not provide any guarantees as to the fidelity of the learned reward function. We believe the EPIC distance will be a highly informative addition to the evaluation toolbox, and would encourage researchers to report EPIC distance in addition to policy-based metrics. Our implementation of EPIC and our baselines, including a tutorial and documentation, are available at `https://github.com/HumanCompatibleAI/evaluating-rewards`.

### ACKNOWLEDGEMENTS

Thanks to Sam Toyer, Rohin Shah, Eric Langlois, Siddharth Reddy and Stuart Armstrong for helpful discussions; to Miljan Martic for code-review; and to David Krueger, Matthew Rahtz, Rachel Freedman, Cody Wild, Alyssa Dayan, Adria Garriga, Jon Uesato, Zac Kenton and Alden Hung for feedback on drafts. This work was supported by Open Philanthropy and the Leverhulme Trust.

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

# A    SUPPLEMENTARY MATERIAL

## A.1    APPROXIMATION PROCEDURES

### A.1.1    SAMPLE-BASED APPROXIMATION FOR EPIC DISTANCE

We approximate EPIC distance (definition 4.6) by estimating Pearson distance on a set of samples, canonicalizing the reward on-demand. Specifically, we sample a batch $B_V$ of $N_V$ samples from the coverage distribution $\mathcal{D}$, and a batch $B_M$ of $N_M$ samples from the joint state and action distributions $\mathcal{D}_\mathcal{S} \times \mathcal{D}_\mathcal{A}$. For each $(s, a, s') \in B_V$, we approximate the canonically shaped rewards (definition 4.1) by taking the mean over $B_M$:

$$C_{\mathcal{D}_\mathcal{S}, \mathcal{D}_\mathcal{A}}(R)(s, a, s') = R(s, a, s') + \mathbb{E}\left[\gamma R(s', A, S') - R(s, A, S') - \gamma R(S, A, S')\right] \quad (5)$$

$$\approx R(s, a, s') + \frac{\gamma}{N_M} \sum_{(x, u) \in B_M} R(s', u, x) \quad (6)$$

$$- \frac{1}{N_M} \sum_{(x, u) \in B_M} R(s, u, x) - c. \quad (7)$$

We drop the constant $c$ from the approximation since it does not affect the Pearson distance; it can also be estimated in $O(N_M^2)$ time by $c = \frac{\gamma}{N_M^2} \sum_{(x, \cdot) \in B_M} \sum_{(x', u) \in B_M} R(x, u, x')$. Finally, we compute the Pearson distance between the approximate canonically shaped rewards on the batch of samples $B_V$, yielding an $O(N_V N_M)$ time algorithm.

### A.1.2    OPTIMIZATION-BASED APPROXIMATION FOR NPEC DISTANCE

$D_{\text{NPEC}}(R_A, R_B)$ (section 5.2) is defined as the infimum of $L^p$ distance over an infinite set of equivalent reward functions $R \equiv R_A$. We approximate this using gradient descent on the reward model

$$R_{\nu, c, w}(s, a, s') = \exp(\nu) R_A(s, a, s') + c + \gamma \Phi_w(s') - \Phi_w(s), \quad (8)$$

where $\nu, c \in \mathbb{R}$ are scalar weights and $w$ is a vector of weights parameterizing a deep neural network $\Phi_w$. The constant $c \in \mathbb{R}$ is unnecessary if $\Phi_w$ has a bias term, but its inclusion simplifies the optimization problem.

We optimize $\nu, c, w$ to minimize the mean of the cost

$$J(\nu, c, w)(s, a, s') = \|R_{\nu, c, w}(s, a, s'), R_B(s, a, s')\|^p \quad (9)$$

on samples $(s, a, s')$ from a coverage distribution $\mathcal{D}$. Note

$$\mathbb{E}_{(S, A, S') \sim \mathcal{D}}\left[J(\nu, c, w)(S, A, S')\right]^{1/p} = D_{L^p, \mathcal{D}}(R_{\nu, c, w}, R_B) \quad (10)$$

upper bounds the true NPEC distance since $R_{\nu, c, w} \equiv R_A$.

We found empirically that $\nu$ and $c$ need to be initialized close to their optimal values for gradient descent to reliably converge. To resolve this problem, we initialize the affine parameters to $\nu \leftarrow \log \lambda$ and $c$ found by:

$$\underset{\lambda \geq 0, c \in \mathbb{R}}{\arg \min} \; \mathbb{E}_{s, a, s' \sim \mathcal{D}} \left(\lambda R_A(s, a, s') + c - R_B(s, a, s')\right)^2. \quad (11)$$

We use the active set method of Lawson & Hanson [11] to solve this constrained least-squares problem. These initial affine parameters minimize the $L^p$ distance $D_{L^p, \mathcal{D}}(R_{\nu, c, 0}(s, a, s'), R_B(s, a, s'))$ when $p = 2$ with the potential fixed at $\Phi_0(s) = 0$.

### A.1.3    CONFIDENCE INTERVALS

We report confidence intervals to help measure the degree of error introduced by the approximations. Since approximate distances may not be normally distributed, we use bootstrapping to produce a distribution-free confidence interval. For EPIC, NPEC and Episode Return (sometimes reported as regret rather than return), we compute independent approximate distances or returns over different

Table A.1: Summary of hyperparameters and distributions used in experiments. The uniform random coverage distribution $\mathcal{D}_{\mathrm{unif}}$ samples states and actions uniformly at random, and samples the next state from the transition dynamics. Random policy $\pi_{\mathrm{uni}}$ takes uniform random actions. The synthetic expert policy $\pi^*$ was trained with PPO on the ground-truth reward. **Mix**ture samples actions from either $\pi_{\mathrm{uni}}$ or $\pi^*$, switching between them at each time step with probability 0.05. Warmstart Size is the size of the dataset used to compute initialization parameters described in section A.1.2.

| Parameter | Value | In experiment |
|---|---|---|
| | Random transitions $\mathcal{D}_{\mathrm{unif}}$ | `GridWorld` |
| Coverage Distribution $\mathcal{D}$ | Rollouts from $\pi_{\mathrm{uni}}$ | `PointMass, HalfCheetah, Hopper` |
| | $\pi_{\mathrm{uni}}, \pi^*$ and **Mix**ture | `PointMaze` |
| Bootstrap Samples | 10 000 | All |
| Discount $\gamma$ | 0.99 | All |
| **EPIC** | | |
| State Distribution $\mathcal{D}_{\mathcal{S}}$ | Marginalized from $\mathcal{D}$ | All |
| Action Distribution $\mathcal{D}_{\mathcal{A}}$ | Marginalized from $\mathcal{D}$ | All |
| Seeds | 30 | All |
| Samples $N_V$ | 32 768 | All |
| Mean Samples $N_M$ | 32 768 | All |
| **NPEC** | | |
| Seeds | 3 | All |
| Total Time Steps | $1 \times 10^6$ | All |
| Optimizer | Adam | All |
| Learning Rate | $1 \times 10^{-2}$ | All |
| Batch Size | 4096 | All |
| Warmstart Size | 16 386 | All |
| Loss $\ell$ | $\ell(x,y) = (x-y)^2$ | All |
| **ERC** | | |
| Episodes | 131 072 | All |
| **Episode Return** | | |
| Seeds | 9 | All |
| See Table A.2 for the policy training hyperparameters | | |

seeds, and then compute a bootstrapped confidence interval for each seed. We use 30 seeds for EPIC, but only 9 seeds for computing Episode Return and 3 seeds for NPEC due to their greater computational requirements. In ERC, computing the distance is very fast, so we instead apply bootstrapping to the collected *episodes*, computing the ERC distance for each bootstrapped episode sample.

## A.2 EXPERIMENTS

### A.2.1 HYPERPARAMETERS FOR APPROXIMATE DISTANCES

Table A.1 summarizes the hyperparameters and distributions used to compute the distances between reward functions. Most parameters are the same across all environments. We use a coverage distribution of uniform random transitions $\mathcal{D}_{\mathrm{unif}}$ in the simple `GridWorld` environment with known determinstic dynamics. In other environments, the coverage distribution is sampled from rollouts of a policy. We use a random policy $\pi_{\mathrm{uni}}$ for `PointMass`, `HalfCheetah` and `Hopper` in the hand-designed reward experiments (section 6.1). In `PointMaze`, we compare three coverage distributions (section 6.2) induced by rollouts of $\pi_{\mathrm{uni}}$, an expert policy $\pi^*$ and a **Mix**ture of the two policies, sampling actions from either $\pi_{\mathrm{uni}}$ or $\pi^*$ and switching between them with probability 0.05 per time step.

Table A.2: Hyperparameters for proximal policy optimisation (PPO) [19]. We used the implementation and default hyperparameters from Hill et al. [9]. PPO was used to train expert policies on ground-truth reward and to optimize learned reward functions for evaluation.

| Parameter | Value | In environment |
|---|---|---|
| Total Time Steps | $1 \times 10^6$ | All |
| Seeds | 9 | All |
| Batch Size | 4096 | All |
| Discount $\gamma$ | 0.99 | All |
| Entropy Coefficient | 0.01 | All |
| Learning Rate | $3 \times 10^{-4}$ | All |
| Value Function Coefficient | 0.5 | All |
| Gradient Clipping Threshold | 0.5 | All |
| Ratio Clipping Thrsehold | 0.2 | All |
| Lambda (GAE) | 0.95 | All |
| Minibatches | 4 | All |
| Optimization Epochs | 4 | All |
| Parallel Environments | 8 | All |

Table A.3: Hyperparameters for adversarial inverse reinforcement learning (AIRL) used in Wang et al. [24].

| Parameter | Value |
|---|---|
| RL Algorithm | PPO [19] |
| Total Time Steps | 1 000 000 |
| Discount $\gamma$ | 0.99 |
| Demonstration Time Steps | 100 000 |
| Generator Batch Size | 2048 |
| Discriminator Batch Size | 50 |
| Entropy Weight | 1.0 |
| Reward Function Architecture | MLP, two 32-unit hidden layers |
| Potential Function Architecture | MLP, two 32-unit hidden layers |

Table A.4: Hyperparameters for preference comparison used in our implementation of Christiano et al. [6].

| Parameter | Value | Range Tested |
|---|---|---|
| Total Time Steps | $5 \times 10^6$ | $[1, 10 \times 10^6]$ |
| Batch Size | 10 000 | $[500, 250\,000]$ |
| Trajectory Length | 5 | $[1, 100]$ |
| Learning Rate | $1 \times 10^{-2}$ | $[1 \times 10^{-4}, 1 \times 10^{-1}]$ |
| Discount $\gamma$ | 0.99 | |
| Reward Function Architecture | MLP, two 32-unit hidden layers | |
| Output L2 Regularization Weight | $1 \times 10^{-4}$ | |

Table A.5: Hyperparameters for regression used in our implementation of Christiano et al. [6, *target* method from section 3.3].

| Parameter | Value | Range Tested |
|---|---|---|
| Total Time Steps | $10 \times 10^6$ | $[1, 20 \times 10^6]$ |
| Batch Size | 4096 | $[256, 16\,384]$ |
| Learning Rate | $2 \times 10^{-2}$ | $[1 \times 10^{-3}, 1 \times 10^{-1}]$ |
| Discount $\gamma$ | 0.99 | |
| Reward Function Architecture | MLP, two 32-unit hidden layers | |

### A.2.2 Training Learned Reward Models

For the experiments on learned reward functions (sections 6.3 and 6.2), we trained reward models using adversarial inverse reinforcement learning (AIRL; 8), preference comparison [6] and by regression onto the ground-truth reward [*target* method from 6, section 3.3]. For AIRL, we use an existing open-source implementation [24]. We developed new implementations for preference comparison and regression, available at `https://github.com/HumanCompatibleAI/evaluating-rewards`. We also use the RL algorithm proximal policy optimization (PPO; 19) on the ground-truth reward to train expert policies to provide demonstrations for AIRL. We use 9 seeds, taking rollouts from the seed with the highest ground-truth return.

Our hyperparameters for PPO in Table A.2 were based on the defaults in Stable Baselines [9]. We only modified the batch size and learning rate, and disabled value function clipping to match the original PPO implementation.

Our AIRL hyperparameters in Table A.3 likewise match the defaults, except for increasing the total number of timesteps to $10^6$. Due to the high variance of AIRL, we trained 5 seeds, selecting the one with the highest ground-truth return. While this does introduce a positive bias for our AIRL results, in spite of this AIRL performed worse in our tests than other algorithms. Moreover, the goal in this paper is to evaluate distance metrics, not reward learning algorithms.

For preference comparison we performed a sweep over batch size, trajectory length and learning rate to decide on the hyperparameters in Table A.4. Total time steps was selected once diminishing returns were observed in loss curves. The exact value of the regularization weight was found to be unimportant, largely controlling the scale of the output at convergence.

Finally, for regression we performed a sweep over batch size, learning rate and total time steps to decide on the hyperparameters in Table A.5. We found batch size and learning rate to be relatively unimportant with many combinations performing well, but regression was found to converge slowly but steadily requiring a relatively large $10 \times 10^6$ time steps for good performance in our environments.

All algorithms are trained on synthetic data generated from the ground-truth reward function. AIRL is provided with a large demonstration dataset of $100\,000$ time steps from an expert policy trained on the ground-truth reward (see Table A.3). In preference comparison and regression, each batch is sampled afresh from the coverage distribution specified in Table A.1 and labeled according to the ground-truth reward.

### A.2.3 Computing infrastructure

Experiments were conducted on a workstation (Intel i9-7920X CPU with 64 GB of RAM), and a small number of `r5.24xlarge` AWS VM instances, with 48 CPU cores on an Intel Skylake processor and 768 GB of RAM. It takes less than three weeks of compute on a single `r5.24xlarge` instance to run all the experiments described in this paper.

### A.2.4 Comparing hand-designed reward functions

We compute distances between hand-designed reward functions in four environments: `GridWorld`, `PointMass`, `HalfCheetah` and `Hopper`. The reward functions for `GridWorld` are described in Figure A.1, and the distances are reported in Figure A.2. We report the approximate distances and confidence intervals between reward functions in the other environments in Figures A.3, A.4 and A.5.

We find the (approximate) EPIC distance closely matches our intuitions for similarity between the reward functions. NPEC often produces similar results to EPIC, but unfortunately is dogged by optimization error. This is particularly notable in higher-dimensional environments like `HalfCheetah` and `Hopper`, where the NPEC distance often exceeds the theoretical upper bound of 1.0 and the confidence interval width is frequently larger than 0.2.

By contrast, ERC distance generally has a tight confidence interval, but systematically fails in the presence of shaping. For example, it confidently assigns large distances between *equivalent* reward pairs in `PointMass` such as S🐎-D🐎. However, ERC produces reasonable results in `HalfCheetah` and `Hopper` where rewards are all similarly shaped. In fact, ERC picks up on a detail in `Hopper` that EPIC misses: whereas EPIC assigns a distance of around 0.71 between

| Distance Metric | Wall-Clock Time | Environment Time Steps | Reward Queries | # of Seeds | 95% CI Width | |
| --- | --- | --- | --- | --- | --- | --- |
| | | | | | Max | Mean |
| EPIC Quick | 17 s | 8192 | $1.67 \times 10^7$ | 3 | 0.023 04 | 0.008 60 |
| EPIC | 6738 s | 65 536 | $1.07 \times 10^9$ | 30 | 0.005 58 | 0.002 34 |
| NPEC | 29 769 s | $7.50 \times 10^7$ | $7.50 \times 10^7$ | 3 | 0.315 91 | 0.066 20 |
| ERC | 1376 s | $6.55 \times 10^6$ | $6.55 \times 10^6$ | — | 0.015 81 | 0.005 33 |
| RL (PPO) | 14 745 s | $7.50 \times 10^7$ | $7.50 \times 10^7$ | 3 | — | — |

Table A.6: Time and resources taken by different metrics to perform 25 distance comparisons on `PointMass`, and the confidence interval widths obtained (smaller is better). Methods *EPIC*, *NPEC* and *ERC* correspond to Figures 2(a), (b) and (c) respectively. *EPIC Quick* is an abbreviated version with fewer samples. *RL (PPO)* is estimated from the time taken using PPO to train a single policy (16m:23s) until convergence ($10^6$ time steps). EPIC samples $N_M + N_V$ time steps from the environment and performs $N_M N_V$ reward queries. In *EPIC Quick*, $N_M = N_V = 4096$; in *EPIC*, $N_M = N_V = 302768$. Other methods query the reward once per environment time step.

all rewards of different types (running vs backflipping), ERC assigns lower distances when the rewards are in the same direction (forward or backward). Given this, ERC may be attractive in some circumstances, especially given the ease of implementation. However, we would caution against using it in isolation due to the likelihood of misleading results in the presence of shaping.

### A.2.5 COMPARING LEARNED REWARD FUNCTIONS

Previously, we reported the mean approximate distance from a ground-truth reward of four learned reward models in `PointMaze` (Table 2). Since these distances are approximate, we report 95% lower and upper bounds computed via bootstrapping in Table A.7. We also include the relative difference of the upper and lower bounds from the mean, finding the relative difference to be fairly consistent across reward models for a given algorithm and coverage distribution pair. The relative difference is less than 1% for all EPIC and ERC distances. However, NPEC confidence intervals can be as wide as 50%: this is due to the method's high variance, and the small number of seeds we were able to run because of the method's computational expense.

### A.2.6 RUNTIME OF DISTANCE METRICS

We report the empirical runtime for EPIC and baselines in Table A.6, performing 25 pairwise comparisons across 5 reward functions in `PointMass`. These comparisons were run on an unloaded machine running Ubuntu 20.04 (kernel 5.4.0-52) with an Intel i9-7920X CPU and 64 GB of RAM. We report sequential runtimes: runtimes for all methods could be decreased further by parallelizing across seeds. The algorithms were configured to use 8 parallel environments for sampling. Inference and training took place on CPU. All methods used the same TensorFlow configuration, parallelizing operations across threads both within and between operations. We found GPUs offered no performance benefit in this setting, and in some cases even increased runtime. This is due to the fixed cost of CPU-GPU communication, and the relatively small size of the observations.

We find that in just 17 seconds EPIC can provide results with a 95% confidence interval $< 0.023$, an order of magnitude tighter than NPEC running for over 8 hours. Training policies for all learned rewards in this environment using PPO is comparatively slow, taking over 4 hours even with only 3 seeds. While ERC is relatively fast, it takes a large number of samples to achieve tight confidence intervals. Moreover, since `PointMass` has stochastic initial states, ERC can take on arbitrary values under shaping, as discussed in sections 5.1 and 6.3.

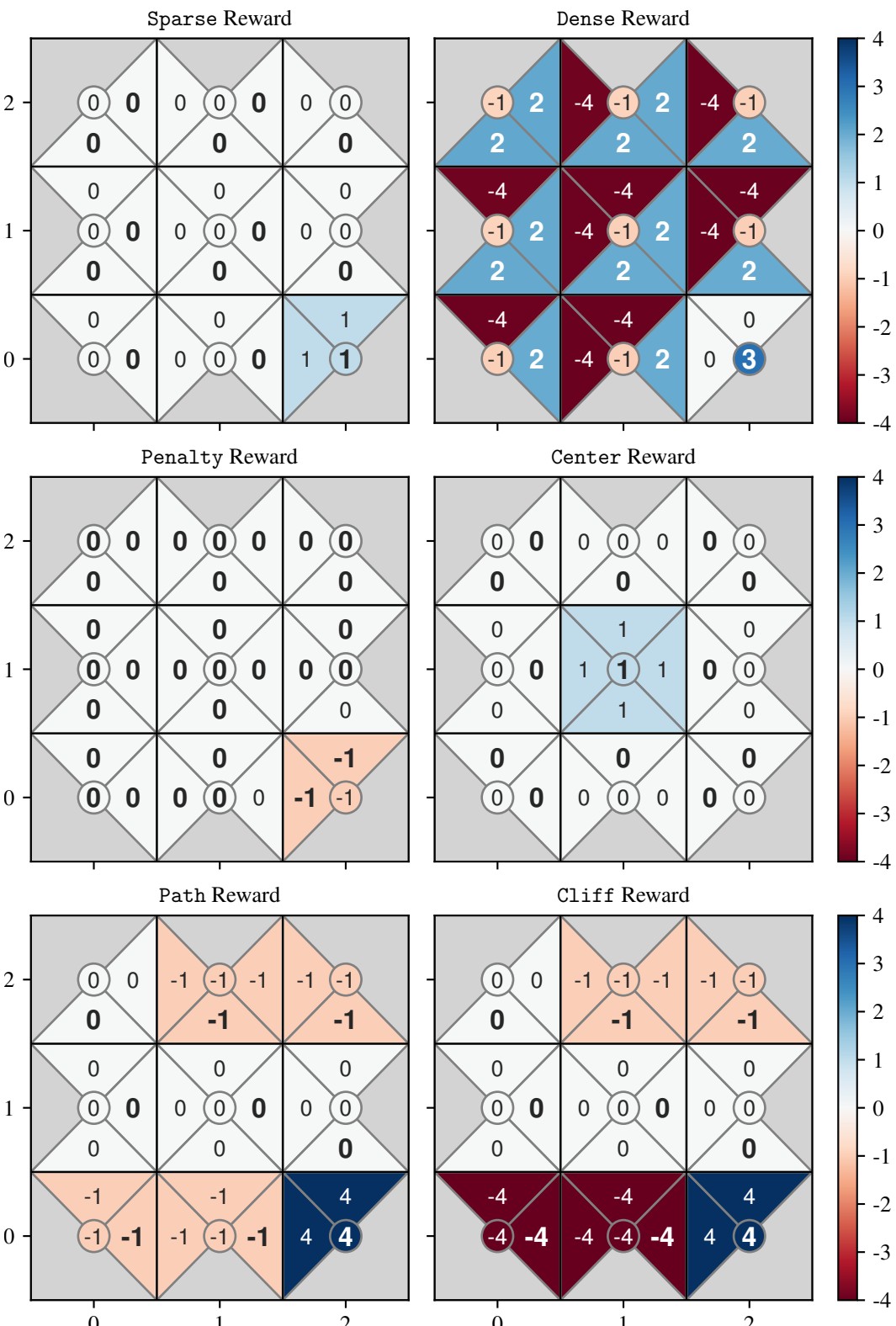

Figure A.1: Heatmaps of reward functions $R(s, a, s')$ for a $3 \times 3$ deterministic gridworld. $R(s, \text{stay}, s)$ is given by the central circle in cell $s$. $R(s, a, s')$ is given by the triangular wedge in cell $s$ adjacent to cell $s'$ in direction $a$. Optimal action(s) (for infinite horizon, discount $\gamma = 0.99$) have bold labels against a hatched background. See Figure A.2 for the distance between all reward pairs.

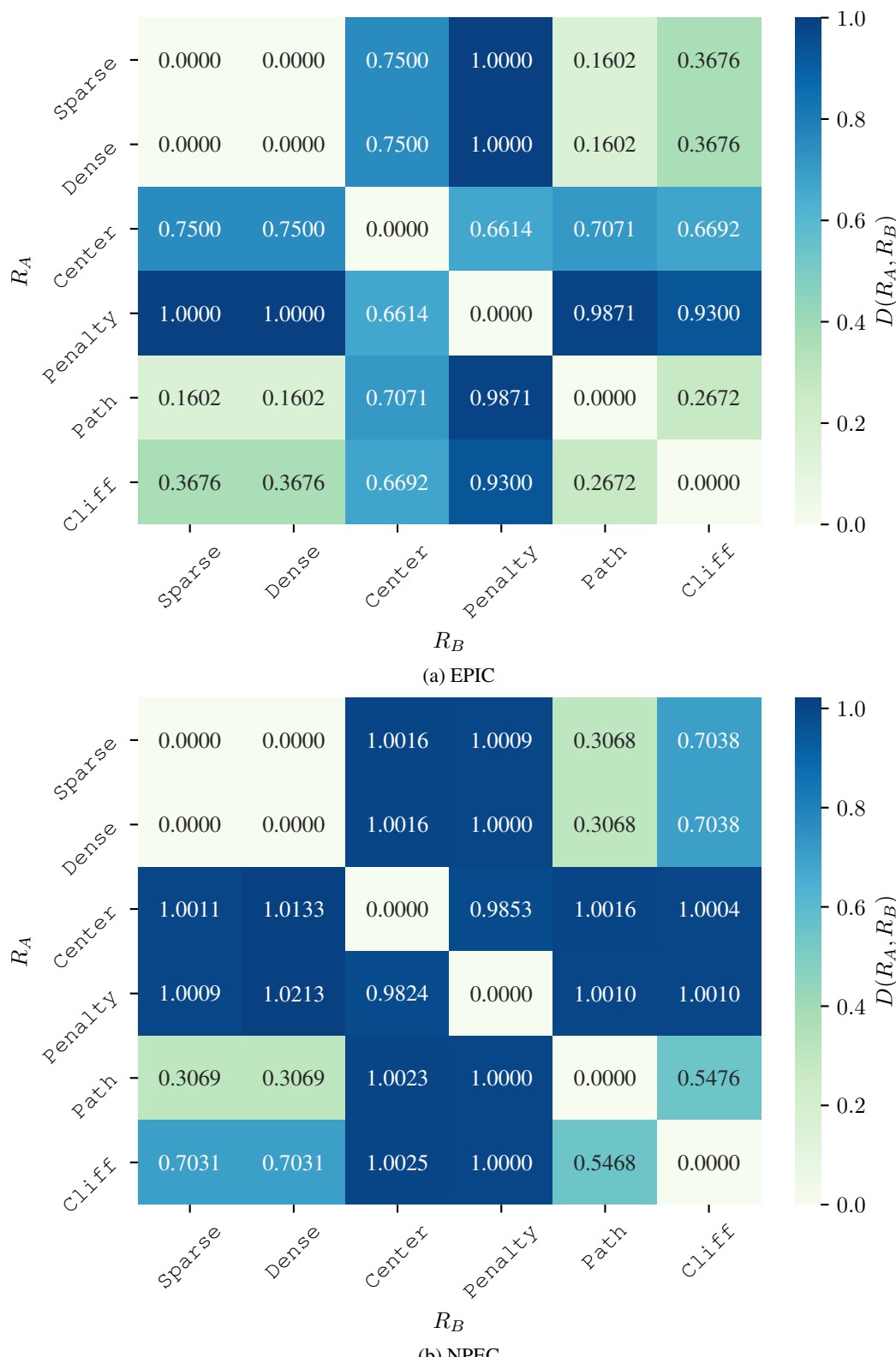

Figure A.2: Distances (EPIC, top; NPEC, bottom) between hand-designed reward functions for the $3 \times 3$ deterministic Gridworld environment. EPIC and NPEC produce similar results, but EPIC more clearly discriminates between rewards whereas NPEC distance tends to saturate. For example, the NPEC distance from Penalty to other rewards lies in the very narrow $[0.98, 1.0]$ range, whereas EPIC uses the wider $[0.66, 1.0]$ range. See Figure A.1 for definitions of each reward. Distances are computed using tabular algorithms. We do not report confidence intervals since these algorithms are deterministic and exact up to floating point error.

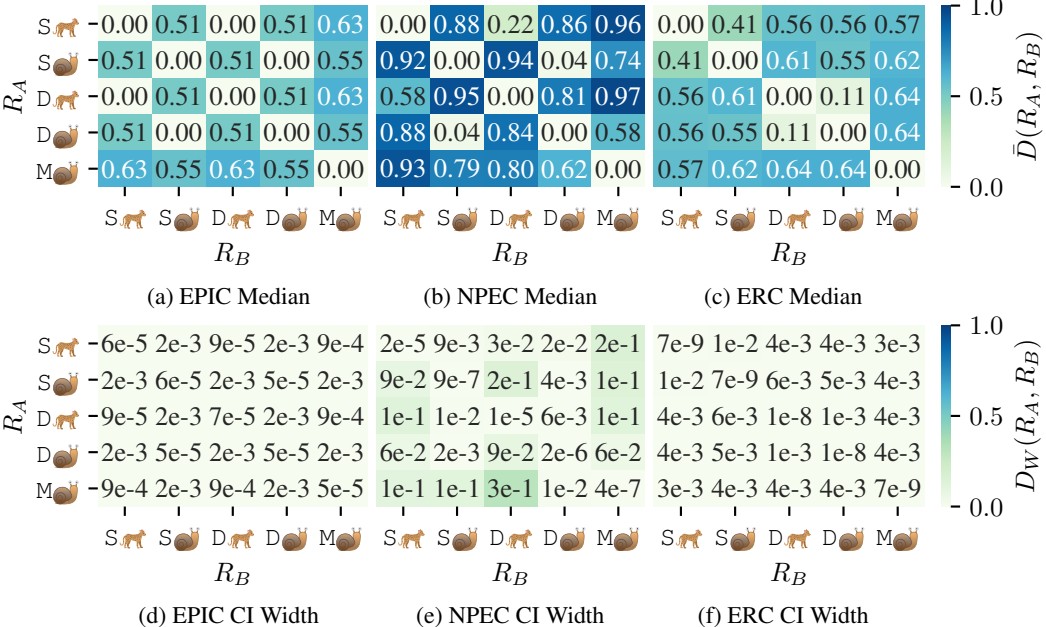

Figure A.3: Approximate distances between hand-designed reward functions in `PointMass`. The coverage distribution $\mathcal{D}$ is sampled from rollouts of a policy $\pi_{\mathrm{uni}}$ taking actions uniformly at random. **Key**: The agent has position $x \in \mathbb{R}$, velocity $\dot{x} \in \mathbb{R}$ and can accelerate $\ddot{x} \in \mathbb{R}$, producing future position $x' \in \mathbb{R}$. 🐕 quadratic penalty on control $\ddot{x}^2$, 🐎 no control penalty. S is $\mathrm{Sparse}(x) = \mathbb{1}[|x| < 0.05]$, D is shaped $\mathrm{Dense}(x, x') = \mathrm{Sparse}(x) + |x'| - |x|$, while M is $\mathrm{Magnitude}(x) = -|x|$. **Confidence Interval (CI)**: 95% CI computed by bootstraping over $10\,000$ samples.

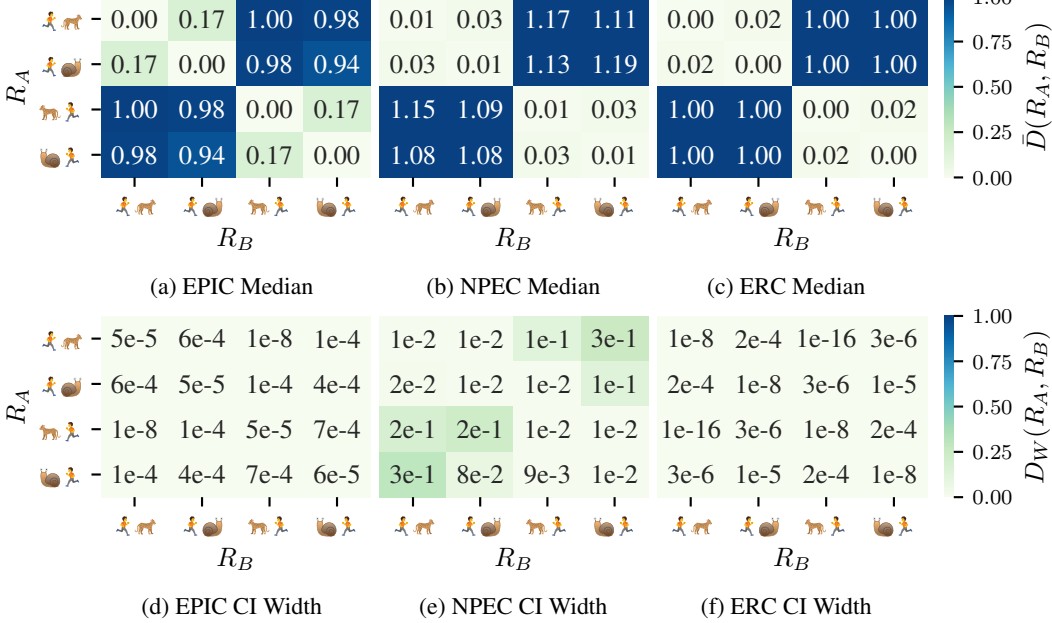

Figure A.4: Approximate distances between hand-designed reward functions in `HalfCheetah`. The coverage distribution $\mathcal{D}$ is sampled from rollouts of a policy $\pi_{\mathrm{uni}}$ taking actions uniformly at random. **Key**: 🏃 is a reward proportional to the change in center of mass and moving *forward* is rewarded when 🏃 to the right, and moving *backward* is rewarded when 🏃 to the left. 🐕 quadratic control penalty, 🐎 no control penalty. **Confidence Interval (CI)**: 95% CI computed by bootstraping over $10\,000$ samples.

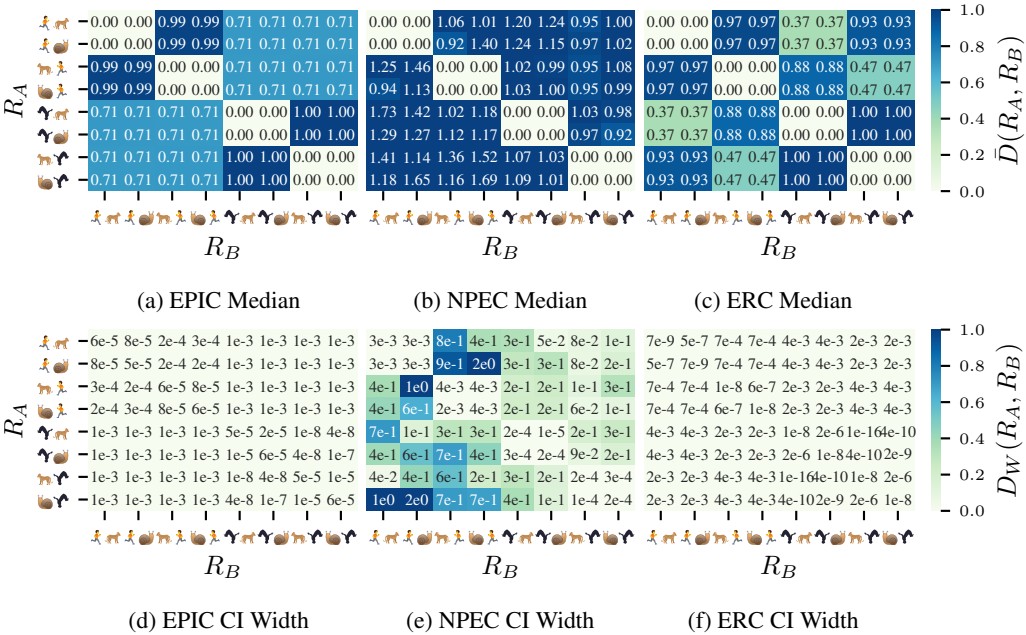

(a) EPIC Median   (b) NPEC Median   (c) ERC Median

(d) EPIC CI Width   (e) NPEC CI Width   (f) ERC CI Width

Figure A.5: Approximate distances between hand-designed reward functions in `Hopper`. The coverage distribution $\mathcal{D}$ is sampled from rollouts of a policy $\pi_{\mathrm{uni}}$ taking actions uniformly at random. **Key**: 🏃 is a reward proportional to the change in center of mass and 🤸 is the backflip reward defined in Amodei et al. [2, footnote]. Moving *forward* is rewarded when 🏃 or 🤸 is to the right, and moving *backward* is rewarded when 🏃 or 🤸 is to the left. 🐌 quadratic control penalty, 🐎 no control penalty. **Confidence Interval (CI)**: 95% CI computed by bootstraping over 10 000 samples.

Table A.7: Approximate distances of reward functions from the ground-truth (GT). We report the 95% bootstrapped lower and upper bounds, the mean, and a 95% bound on the relative error from the mean. Distances (1000× scale) use coverage distribution $\mathcal{D}$ from rollouts in the `PointMaze-Train` environment of: a uniform random policy $\pi_{\mathbf{uni}}$, an expert $\pi^*$ and a **Mix**ture of these policies. $\mathcal{D}_{\mathcal{S}}$ and $\mathcal{D}_{\mathcal{A}}$ are computed by marginalizing $\mathcal{D}$.

(a) 95% lower bound $D^{\text{LOW}}$ of approximate distance.

| Reward | $1000 \times D_{\text{EPIC}}^{\text{LOW}}$ | | | $1000 \times D_{\text{NPEC}}^{\text{LOW}}$ | | | $1000 \times D_{\text{ERC}}^{\text{LOW}}$ | | | Episode Return | |
| Function | $\pi_{\text{uni}}$ | $\pi^*$ | Mix | $\pi_{\text{uni}}$ | $\pi^*$ | Mix | $\pi_{\text{uni}}$ | $\pi^*$ | Mix | Train | Test |
|---|---|---|---|---|---|---|---|---|---|---|---|
| GT | 0.03 | 0.02 | <0.01 | 0.02 | 1.43 | <0.01 | 0.00 | 0.00 | 0.00 | −4.46 | −5.82 |
| Regress | 35.5 | 33.6 | 26.0 | 1.22 | 38.8 | 0.33 | 9.94 | 90.2 | 2.42 | −4.7 | −5.63 |
| Pref | 68.3 | 100 | 56.6 | 7.02 | 1239 | 9.25 | 24.7 | 358 | 19.5 | −5.26 | −4.88 |
| AIRL SO | 570 | 519 | 402 | 734 | 1645 | 424 | 547 | 521 | 238 | −27.3 | −22.7 |
| AIRL SA | 774 | 930 | 894 | 956 | 723 | 952 | 802 | 720 | 963 | −29.9 | −28 |
| Mirage | 3.49 | 0.02 | 381 | 0.17 | 4.03 | 481 | 25.8 | <0.01 | 162 | −28.4 | −26.2 |

(b) Mean approximate distance $\overline{D}$. Results are the same as Table 2.

| Reward | $1000 \times \overline{D}_{\text{EPIC}}$ | | | $1000 \times \overline{D}_{\text{NPEC}}$ | | | $1000 \times \overline{D}_{\text{ERC}}$ | | | Episode Return | |
| Function | $\pi_{\text{uni}}$ | $\pi^*$ | Mix | $\pi_{\text{uni}}$ | $\pi^*$ | Mix | $\pi_{\text{uni}}$ | $\pi^*$ | Mix | Train | Test |
|---|---|---|---|---|---|---|---|---|---|---|---|
| GT | 0.06 | 0.05 | 0.04 | 0.04 | 3.17 | 0.01 | 0.00 | 0.00 | 0.00 | −5.19 | −6.59 |
| Regress | 35.8 | 33.7 | 26.1 | 1.42 | 38.9 | 0.35 | 9.99 | 90.7 | 2.43 | −5.47 | −6.3 |
| Pref | 68.7 | 100 | 56.8 | 8.51 | 1333 | 9.74 | 24.9 | 360 | 19.6 | −5.57 | −5.04 |
| AIRL SO | 572 | 520 | 404 | 817 | 2706 | 488 | 549 | 523 | 240 | −27.3 | −22.7 |
| AIRL SA | 776 | 930 | 894 | 1067 | 2040 | 1039 | 803 | 722 | 964 | −30.7 | −29 |
| Mirage | 17.0 | 0.05 | 397 | 0.68 | 6.30 | 597 | 35.3 | <0.01 | 166 | −30.4 | −29.1 |

(c) 95% upper bound $D^{\text{UP}}$ of approximate distance.

| Reward | $1000 \times D_{\text{EPIC}}^{\text{UP}}$ | | | $1000 \times D_{\text{NPEC}}^{\text{UP}}$ | | | $1000 \times D_{\text{ERC}}^{\text{UP}}$ | | | Episode Return | |
| Function | $\pi_{\text{uni}}$ | $\pi^*$ | Mix | $\pi_{\text{uni}}$ | $\pi^*$ | Mix | $\pi_{\text{uni}}$ | $\pi^*$ | Mix | Train | Test |
|---|---|---|---|---|---|---|---|---|---|---|---|
| GT | 0.09 | 0.07 | 0.07 | 0.06 | 6.14 | 0.01 | <0.01 | <0.01 | <0.01 | −6.04 | −7.14 |
| Regress | 36.1 | 33.7 | 26.2 | 1.53 | 39.1 | 0.37 | 10.0 | 91.2 | 2.44 | −6.26 | −6.83 |
| Pref | 69.1 | 101 | 57.1 | 10.0 | 1432 | 10.1 | 25.2 | 361 | 19.7 | −5.9 | −5.22 |
| AIRL SO | 574 | 520 | 407 | 982 | 3984 | 532 | 551 | 526 | 242 | −27.3 | −22.8 |
| AIRL SA | 779 | 930 | 895 | 1241 | 4378 | 1124 | 805 | 724 | 964 | −31.7 | −29.8 |
| Mirage | 35.2 | 0.09 | 414 | 1.66 | 10.8 | 821 | 45.4 | <0.01 | 171 | −32 | −31.4 |

(d) Relative 95% confidence interval $D^{\text{REL}} = \left| \max\left( \frac{\text{Upper}}{\text{Mean}} - 1, 1 - \frac{\text{Lower}}{\text{Mean}} \right) \right|$ in percent. The population mean is contained within $\pm D^{\text{REL}}\%$ of the sample mean in Table A.7b with 95% probability.

| Reward | $D_{\text{EPIC}}^{\text{REL}}\%$ | | | $D_{\text{NPEC}}^{\text{REL}}\%$ | | | $D_{\text{ERC}}^{\text{REL}}\%$ | | | Episode Return | |
| Function | $\pi_{\text{uni}}$ | $\pi^*$ | Mix | $\pi_{\text{uni}}$ | $\pi^*$ | Mix | $\pi_{\text{uni}}$ | $\pi^*$ | Mix | Train | Test |
|---|---|---|---|---|---|---|---|---|---|---|---|
| GT | 50.0 | 62.5 | 80.0 | 61.8 | 94.0 | 29.7 | inf | inf | inf | 0.16 | 0.12 |
| Regress | 0.81 | 0.14 | 0.40 | 14.2 | 0.42 | 7.48 | 0.53 | 0.55 | 0.57 | 0.14 | 0.11 |
| Pref | 0.61 | 0.14 | 0.44 | 17.5 | 7.49 | 5.02 | 0.90 | 0.48 | 0.48 | 0.06 | 0.04 |
| AIRL SO | 0.38 | 0.08 | 0.67 | 20.2 | 47.2 | 13.2 | 0.34 | 0.40 | 0.69 | <0.01 | <0.01 |
| AIRL SA | 0.35 | 0.02 | 0.08 | 16.3 | 115 | 8.42 | 0.23 | 0.26 | 0.04 | 0.03 | 0.04 |
| Mirage | 108 | 65.5 | 4.17 | 142 | 70.9 | 37.5 | 28.5 | 0.55 | 2.66 | 0.07 | 0.10 |

Table A.8: Approximate distances of reward functions from the ground-truth (GT) under pathological coverage distributions. We report the 95% bootstrapped lower and upper bounds, the mean, and a 95% bound on the relative error from the mean. Distances ($1000\times$ scale) use four different coverage distributions $\mathcal{D}$. $\sigma$ independently samples states, actions and next states from the marginal distributions of rollouts from the uniform random policy $\pi_{\mathrm{uni}}$ in the PointMaze-Train environment. **Ind** independently samples the components of states and next states from $\mathcal{N}(0, 1)$, and actions from $U[-1, 1]$. **Jail** consists of rollouts of $\pi_{\mathrm{uni}}$ restricted to a small $0.09 \times 0.09$ "jail" square that excludes the goal state $0.5$ distance away. $\pi_{\mathbf{bad}}$ are rollouts in PointMaze-Train of a policy that goes to the corner opposite the goal state. $\sigma$ and **Ind** are not supported by ERC since they do not produce complete episodes.

(a) 95% lower bound $D^{\mathrm{LOW}}$ of approximate distance.

| Reward Function | $1000 \times D^{\mathrm{LOW}}_{\mathrm{EPIC}}$ | | | | $1000 \times D^{\mathrm{LOW}}_{\mathrm{NPEC}}$ | | | | $1000 \times D^{\mathrm{LOW}}_{\mathrm{ERC}}$ | |
|---|---|---|---|---|---|---|---|---|---|---|
| | $\sigma$ | Ind | Jail | $\pi_{\mathbf{bad}}$ | $\sigma$ | Ind | Jail | $\pi_{\mathbf{bad}}$ | Jail | $\pi_{\mathbf{bad}}$ |
| Regress | 127 | 398 | 705 | 205 | 87.6 | 590 | 2433 | 898 | 809 | 456 |
| Pref | 146 | 433 | 462 | 349 | 97.4 | 632 | 661 | 221 | 372 | 332 |
| AIRL SO | 570 | 541 | 712 | 710 | 697 | 821 | 957 | 621 | 751 | 543 |
| AIRL SA | 768 | 628 | 558 | 669 | 720 | 960 | 940 | 2355 | 428 | 753 |
| Mirage | 9.22 | 0.03 | <0.01 | 0.02 | 0.41 | 0.05 | 11.2 | 31.3 | <0.01 | 0.02 |

(b) Mean approximate distance $\overline{D}$. Results are the same as Table 2.

| Reward Function | $1000 \times \overline{D}_{\mathrm{EPIC}}$ | | | | $1000 \times \overline{D}_{\mathrm{NPEC}}$ | | | | $1000 \times \overline{D}_{\mathrm{ERC}}$ | |
|---|---|---|---|---|---|---|---|---|---|---|
| | $\sigma$ | Ind | Jail | $\pi_{\mathbf{bad}}$ | $\sigma$ | Ind | Jail | $\pi_{\mathbf{bad}}$ | Jail | $\pi_{\mathbf{bad}}$ |
| Regress | 128 | 398 | 705 | 206 | 97.2 | 591 | 2549 | 921 | 810 | 458 |
| Pref | 147 | 433 | 463 | 349 | 117 | 633 | 683 | 237 | 374 | 333 |
| AIRL SO | 573 | 541 | 713 | 710 | 826 | 823 | 988 | 852 | 753 | 545 |
| AIRL SA | 771 | 628 | 558 | 669 | 859 | 962 | 964 | 2694 | 430 | 754 |
| Mirage | 42.4 | 0.06 | 0.03 | 0.05 | 1.41 | 0.25 | 18.3 | 39 | <0.01 | 0.02 |

(c) 95% upper bound $D^{\mathrm{UP}}$ of approximate distance.

| Reward Function | $1000 \times D^{\mathrm{UP}}_{\mathrm{EPIC}}$ | | | | $1000 \times D^{\mathrm{UP}}_{\mathrm{NPEC}}$ | | | | $1000 \times D^{\mathrm{UP}}_{\mathrm{ERC}}$ | |
|---|---|---|---|---|---|---|---|---|---|---|
| | $\sigma$ | Ind | Jail | $\pi_{\mathbf{bad}}$ | $\sigma$ | Ind | Jail | $\pi_{\mathbf{bad}}$ | Jail | $\pi_{\mathbf{bad}}$ |
| Regress | 129 | 398 | 706 | 206 | 106 | 593 | 2654 | 948 | 812 | 460 |
| Pref | 148 | 433 | 464 | 349 | 132 | 635 | 705 | 265 | 376 | 335 |
| AIRL SO | 576 | 541 | 713 | 710 | 939 | 825 | 1047 | 1021 | 755 | 547 |
| AIRL SA | 774 | 628 | 559 | 669 | 1015 | 963 | 981 | 3012 | 432 | 756 |
| Mirage | 85.9 | 0.09 | 0.05 | 0.09 | 3.25 | 0.46 | 28.6 | 45.3 | <0.01 | 0.02 |

(d) Relative 95% confidence interval $D^{\mathrm{REL}} = \left| \max \left( \frac{\mathrm{Upper}}{\mathrm{Mean}} - 1, 1 - \frac{\mathrm{Lower}}{\mathrm{Mean}} \right) \right|$ in percent. The population mean is contained within $\pm D^{\mathrm{REL}}\%$ of the sample mean in Table A.8b with 95% probability.

| Reward Function | $D^{\mathrm{REL}}_{\mathrm{EPIC}}\%$ | | | | $D^{\mathrm{REL}}_{\mathrm{NPEC}}\%$ | | | | $D^{\mathrm{REL}}_{\mathrm{ERC}}\%$ | |
|---|---|---|---|---|---|---|---|---|---|---|
| | $\sigma$ | Ind | Jail | $\pi_{\mathbf{bad}}$ | $\sigma$ | Ind | Jail | $\pi_{\mathbf{bad}}$ | Jail | $\pi_{\mathbf{bad}}$ |
| Regress | 0.79 | 0.01 | 0.08 | 0.06 | 9.81 | 0.24 | 4.54 | 2.89 | 0.18 | 0.42 |
| Pref | 0.76 | <0.01 | 0.16 | 0.07 | 16.9 | 0.35 | 3.35 | 11.7 | 0.47 | 0.48 |
| AIRL SO | 0.48 | <0.01 | 0.07 | 0.02 | 15.6 | 0.28 | 6.01 | 27.1 | 0.23 | 0.38 |
| AIRL SA | 0.38 | <0.01 | 0.13 | 0.03 | 18.2 | 0.22 | 2.47 | 12.6 | 0.45 | 0.23 |
| Mirage | 103 | 50 | 80 | 83.3 | 131 | 85.4 | 56.4 | 19.7 | 0.54 | 0.55 |

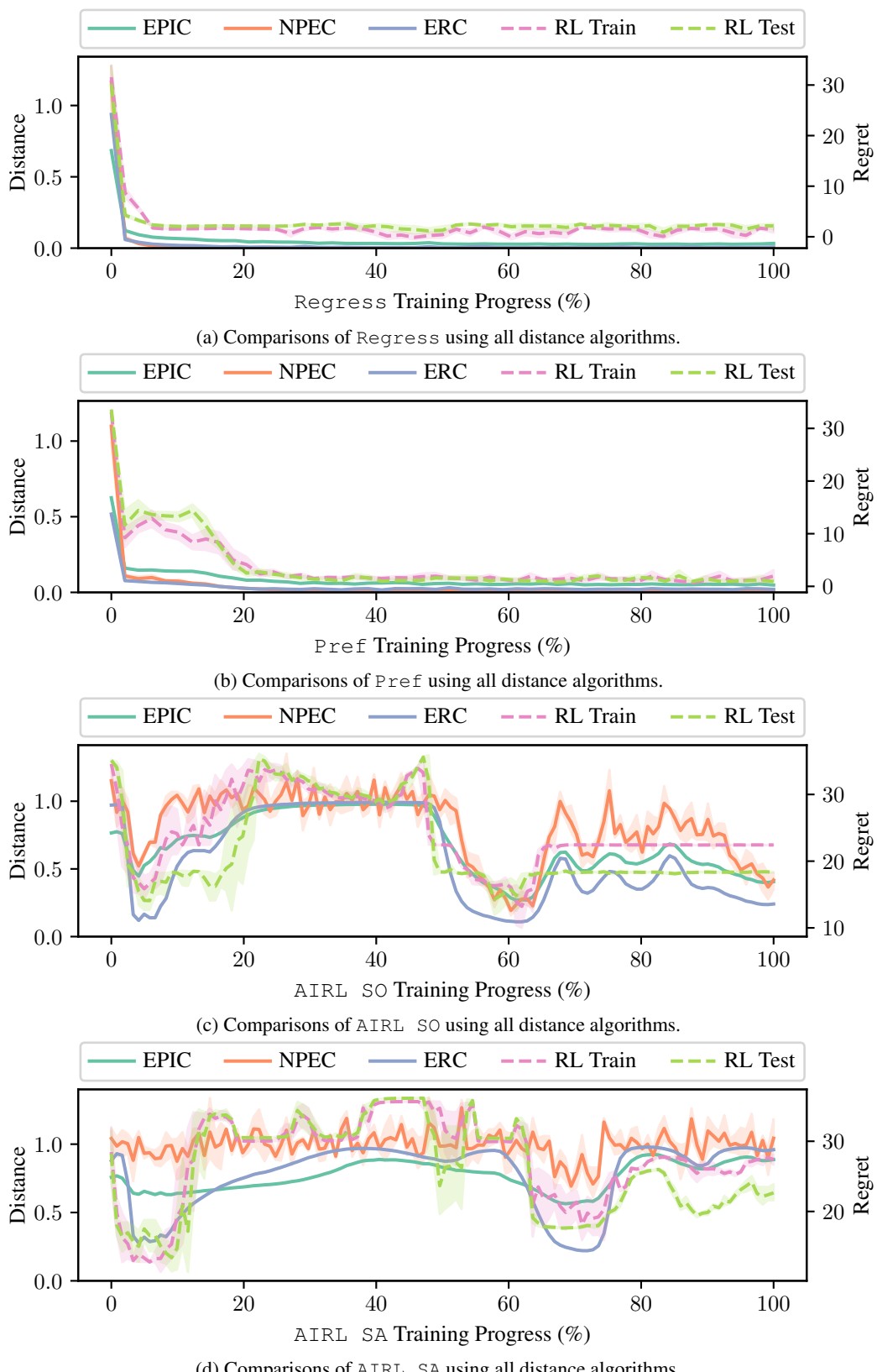

(a) Comparisons of `Regress` using all distance algorithms.

(b) Comparisons of `Pref` using all distance algorithms.

(c) Comparisons of `AIRL SO` using all distance algorithms.

(d) Comparisons of `AIRL SA` using all distance algorithms.

Figure A.6: Distance of reward checkpoints from the ground-truth in `PointMaze` and policy regret for reward checkpoints during reward model training. Each point evaluates a reward function checkpoint from a single seed. EPIC, NPEC and ERC distance use the `Mixture` distribution. Regret is computed by running RL on the checkpoint. The shaded region represents the bootstrapped 95% confidence interval for the distance or regret at that checkpoint, calculated following Section A.1.3.

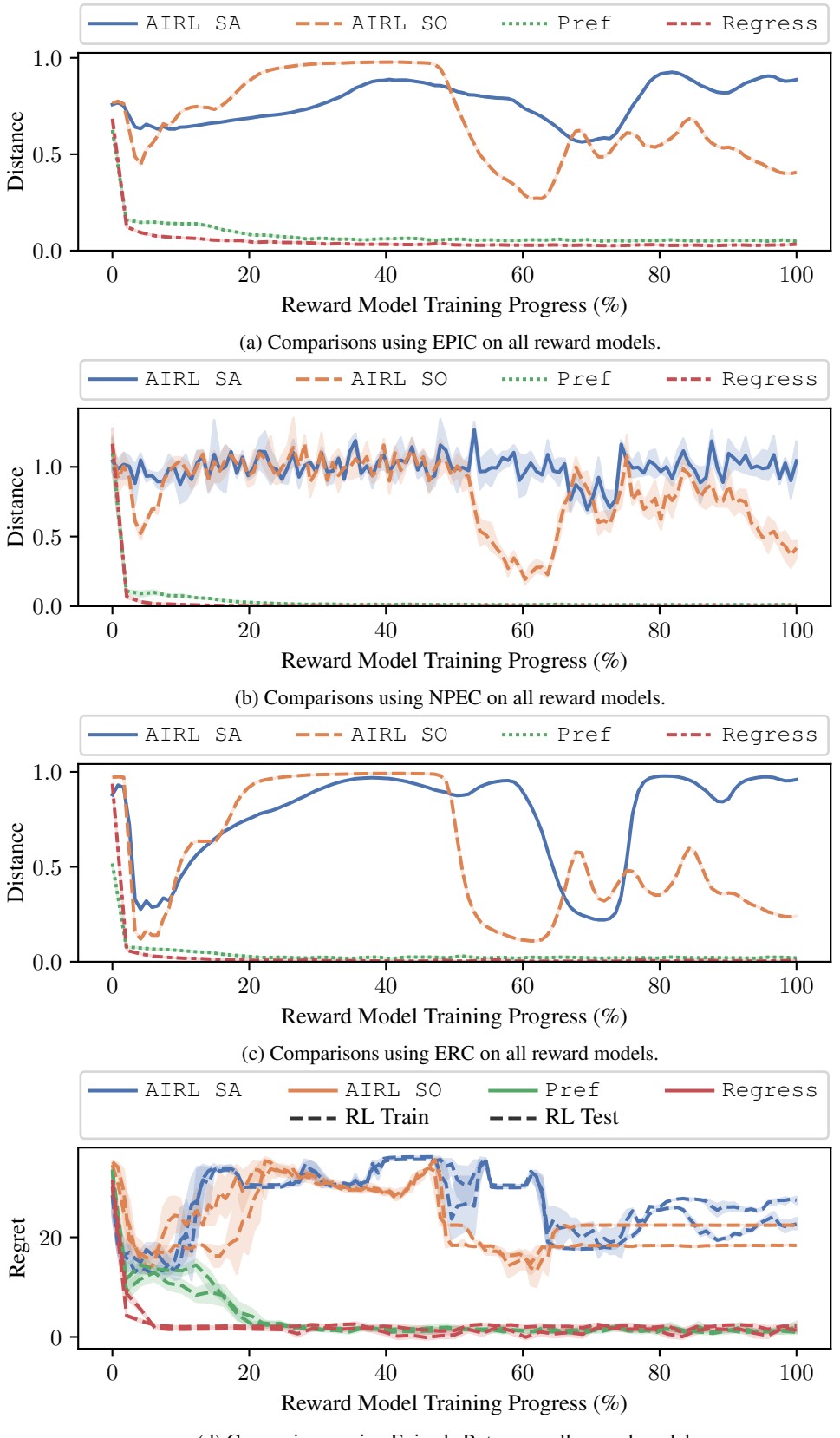

(a) Comparisons using EPIC on all reward models.

(b) Comparisons using NPEC on all reward models.

(c) Comparisons using ERC on all reward models.

(d) Comparisons using Episode Return on all reward models.

Figure A.7: Distance of reward checkpoints from the ground-truth in `PointMaze` and policy regret for reward checkpoints during reward model training. Each point evaluates a reward function checkpoint from a single seed. EPIC, NPEC and ERC distance use the `Mixture` distribution. Regret is computed by running RL on the checkpoint. The shaded region represents the bootstrapped 95% confidence interval for the distance or regret at that checkpoint, calculated following Section A.1.3.

### A.3 PROOFS

#### A.3.1 BACKGROUND

**Proposition 3.4.** *The binary relation $\equiv$ is an equivalence relation. Let $R_A, R_B, R_C : \mathcal{S} \times \mathcal{A} \times \mathcal{S} \to \mathbb{R}$ be bounded reward functions. Then $\equiv$ is reflexive, $R_A \equiv R_A$; symmetric, $R_A \equiv R_B$ implies $R_B \equiv R_A$; and transitive, $(R_A \equiv R_B) \wedge (R_B \equiv R_C)$ implies $R_A \equiv R_C$.*

*Proof.* $R_A \equiv R_A$ since choosing $\lambda = 1 > 0$ and $\Phi(s) = 0$, a bounded potential function, we have $R_A(s, a, s') = \lambda R_A(s, a, s') + \gamma \Phi(s') - \Phi(s)$ for all $s, s' \in \mathcal{S}$ and $a \in \mathcal{A}$.

Suppose $R_A \equiv R_B$. Then there exists some $\lambda > 0$ and a bounded potential function $\Phi : \mathcal{S} \to \mathbb{R}$ such that $R_B(s, a, s') = \lambda R_A(s, a, s') + \gamma \Phi(s') - \Phi(s)$ for all $s, s' \in \mathcal{S}$ and $a \in \mathcal{A}$. Rearranging:

$$R_A(s, a, s') = \frac{1}{\lambda} R_B(s, a, s') + \gamma \left( \frac{-1}{\lambda} \Phi(s') \right) - \left( \frac{-1}{\lambda} \Phi(s) \right). \tag{12}$$

Since $\frac{1}{\lambda} > 0$ and $\Phi'(s) = \frac{-1}{\lambda} \Phi(s)$ is a bounded potential function, it follows that $R_B \equiv R_A$.

Finally, suppose $R_A \equiv R_B$ and $R_B \equiv R_C$. Then there exists some $\lambda_1, \lambda_2 > 0$ and bounded potential functions $\Phi_1, \Phi_2 : \mathcal{S} \to \mathbb{R}$ such that for all $s, s' \in \mathcal{S}$ and $a \in \mathcal{A}$:

$$R_B(s, a, s') = \lambda_1 R_A(s, a, s') + \gamma \Phi_1(s') - \Phi_1(s), \tag{13}$$
$$R_C(s, a, s') = \lambda_2 R_B(s, a, s') + \gamma \Phi_2(s') - \Phi_2(s). \tag{14}$$

Substituting the expression for $R_B$ into the expression for $R_C$:

$$R_C(s, a, s') = \lambda_2 \left( \lambda_1 R_A(s, a, s') + \gamma \Phi_1(s') - \Phi_1(s) \right) + \gamma \Phi_2(s') - \Phi_2(s) \tag{15}$$
$$= \lambda_1 \lambda_2 R_A(s, a, s') + \gamma \left( \lambda_2 \Phi_1(s') + \Phi_2(s') \right) - \left( \lambda_2 \Phi_1(s) + \Phi_2(s) \right) \tag{16}$$
$$= \lambda R_A(s, a, s') + \gamma \Phi(s') - \Phi(s), \tag{17}$$

where $\lambda = \lambda_1 \lambda_2 > 0$ and $\Phi(s) = \lambda_2 \Phi_1(s) + \Phi_2(s)$ is bounded. Thus $R_A \equiv R_C$. $\square$

#### A.3.2 EQUIVALENT-POLICY INVARIANT COMPARISON (EPIC) PSEUDOMETRIC

**Proposition 4.2** (The Canonically Shaped Reward is Invariant to Shaping). *Let $R : \mathcal{S} \times \mathcal{A} \times \mathcal{S} \to \mathbb{R}$ be a reward function and $\Phi : \mathcal{S} \to \mathbb{R}$ a potential function. Let $\gamma \in [0, 1]$ be a discount rate, and $\mathcal{D}_{\mathcal{S}} \in \Delta(\mathcal{S})$ and $\mathcal{D}_{\mathcal{A}} \in \Delta(\mathcal{A})$ be distributions over states and actions. Let $R'$ denote $R$ shaped by $\Phi$: $R'(s, a, s') = R(s, a, s') + \gamma \Phi(s') - \Phi(s)$. Then the canonically shaped $R'$ and $R$ are equal: $C_{\mathcal{D}_{\mathcal{S}}, \mathcal{D}_{\mathcal{A}}}(R') = C_{\mathcal{D}_{\mathcal{S}}, \mathcal{D}_{\mathcal{A}}}(R)$.*

*Proof.* Let $s, a, s' \in \mathcal{S} \times \mathcal{A} \times \mathcal{S}$. Then by substituting in the definition of $R'$ and using linearity of expectations:

$$C_{\mathcal{D}_{\mathcal{S}}, \mathcal{D}_{\mathcal{A}}}(R')(s, a, s') \triangleq R'(s, a, s') + \mathbb{E}\left[ \gamma R'(s', A, S') - R'(s, A, S') - \gamma R'(S, A, S') \right] \tag{18}$$
$$= (R(s, a, s') + \gamma \Phi(s') - \Phi(s)) \tag{19}$$
$$+ \mathbb{E}\left[ \gamma R(s', A, S') + \gamma^2 \Phi(S') - \gamma \Phi(s') \right]$$
$$- \mathbb{E}\left[ R(s, A, S') + \gamma \Phi(S') - \Phi(s) \right]$$
$$- \mathbb{E}\left[ \gamma R(S, A, S') + \gamma^2 \Phi(S') - \gamma \Phi(S) \right]$$
$$= R(s, a, s') + \mathbb{E}\left[ \gamma R(s', A, S') - R(s, A, S') - \gamma R(S, A, S') \right] \tag{20}$$
$$+ (\gamma \Phi(s') - \Phi(s)) - \mathbb{E}\left[ \gamma \Phi(s') - \Phi(s) \right]$$
$$+ \mathbb{E}\left[ \gamma^2 \Phi(S') - \gamma \Phi(S') \right] - \mathbb{E}\left[ \gamma^2 \Phi(S') - \gamma \Phi(S) \right]$$
$$= R(s, a, s') + \mathbb{E}\left[ \gamma R(s', A, S') - R(s, A, S') - \gamma R(S, A, S') \right] \tag{21}$$
$$\triangleq C_{\mathcal{D}_{\mathcal{S}}, \mathcal{D}_{\mathcal{A}}}(R)(s, a, s'), \tag{22}$$

where the penultimate step uses $\mathbb{E}[\Phi(S')] = \mathbb{E}[\Phi(S)]$ since $S$ and $S'$ are identically distributed. $\square$

**Proposition 4.3.** *Let $\mathcal{S}$ and $\mathcal{A}$ be finite, with $|\mathcal{S}| \geq 2$. Let $\mathcal{D}_{\mathcal{S}} \in \Delta(\mathcal{S})$ and $\mathcal{D}_{\mathcal{A}} \in \Delta(\mathcal{A})$. Let $R, \nu : \mathcal{S} \times \mathcal{A} \times \mathcal{S} \to \mathbb{R}$ be reward functions, with $\nu(s, a, s') = \lambda \mathbb{I}[(s, a, s') = (x, u, x')]$, $\lambda \in \mathbb{R}$, $x, x' \in \mathcal{S}$ and $u \in \mathcal{A}$. Let $\Phi_{\mathcal{D}_{\mathcal{S}}, \mathcal{D}_{\mathcal{A}}}(R)(s, a, s') = C_{\mathcal{D}_{\mathcal{S}}, \mathcal{D}_{\mathcal{A}}}(R)(s, a, s') - R(s, a, s')$. Then:*

$$\left\| \Phi_{\mathcal{D}_{\mathcal{S}}, \mathcal{D}_{\mathcal{A}}}(R + \nu) - \Phi_{\mathcal{D}_{\mathcal{S}}, \mathcal{D}_{\mathcal{A}}}(R) \right\|_{\infty} = \lambda \left( 1 + \gamma \mathcal{D}_{\mathcal{S}}(x) \right) \mathcal{D}_{\mathcal{A}}(u) \mathcal{D}_{\mathcal{S}}(x'). \tag{3}$$

*Proof.* Observe that:

$$\Phi_{\mathcal{D}_{\mathcal{S}}, \mathcal{D}_{\mathcal{A}}}(R)(s, a, s') = \mathbb{E}\left[ \gamma R(s', A, S') - R(s, A, S') - \gamma R(S, A, S') \right], \tag{23}$$

where $S$ and $S'$ are random variables independently sampled from $\mathcal{D}_{\mathcal{S}}$, and $A$ independently sampled from $\mathcal{D}_{\mathcal{A}}$.

Then:

$$\Phi_{\mathcal{D}_{\mathcal{S}}, \mathcal{D}_{\mathcal{A}}}(R + \nu) - \Phi_{\mathcal{D}_{\mathcal{S}}, \mathcal{D}_{\mathcal{A}}}(R) = \Phi_{\mathcal{D}_{\mathcal{S}}, \mathcal{D}_{\mathcal{A}}}(\nu). \tag{24}$$

Now:

$$\text{LHS} \triangleq \left\| \Phi_{\mathcal{D}_{\mathcal{S}}, \mathcal{D}_{\mathcal{A}}}(R + \nu) - \Phi_{\mathcal{D}_{\mathcal{S}}, \mathcal{D}_{\mathcal{A}}}(R) \right\|_{\infty} \tag{25}$$

$$= \max_{s, s' \in \mathcal{S}} \left| \mathbb{E}\left[ \gamma \nu(s', A, S') - \nu(s, A, S') - \gamma \nu(S, A, S') \right] \right| \tag{26}$$

$$= \max_{s, s' \in \mathcal{S}} \left| \lambda \left( \gamma \mathbb{I}[x = s'] \mathcal{D}_{\mathcal{A}}(u) \mathcal{D}_{\mathcal{S}}(x') \right. \right. \tag{27}$$

$$\left. - \mathbb{I}[x = s] \mathcal{D}_{\mathcal{A}}(u) \mathcal{D}_{\mathcal{S}}(x') - \gamma \mathcal{D}_{\mathcal{S}}(x) \mathcal{D}_{\mathcal{A}}(u) \mathcal{D}_{\mathcal{S}}(x') \right) \right| \tag{28}$$

$$= \max_{s, s' \in \mathcal{S}} \left| \lambda \mathcal{D}_{\mathcal{A}}(u) \mathcal{D}_{\mathcal{S}}(x') \left( \gamma \mathbb{I}[x = s'] - \mathbb{I}[x = s] - \gamma \mathcal{D}_{\mathcal{S}}(x) \right) \right| \tag{29}$$

$$= \lambda \left( 1 + \gamma \mathcal{D}_{\mathcal{S}}(x) \right) \mathcal{D}_{\mathcal{A}}(u) \mathcal{D}_{\mathcal{S}}(x'), \tag{30}$$

where the final step follows by substituting $s = x$ and $s' \neq x$ (using $|\mathcal{S}| \geq 2$). $\qquad \square$

**Lemma 4.5.** *The Pearson distance $D_{\rho}$ is a pseudometric. Moreover, let $a, b \in (0, \infty)$, $c, d \in \mathbb{R}$ and $X, Y$ be random variables. Then it follows that $0 \leq D_{\rho}(aX + c, bY + d) = D_{\rho}(X, Y) \leq 1$.*

*Proof.* For a non-constant random variable $V$, define a standardized (zero mean and unit variance) version:

$$Z(V) = \frac{V - \mathbb{E}[V]}{\sqrt{\mathbb{E}\left[ (V - \mathbb{E}[V])^2 \right]}}. \tag{31}$$

The Pearson correlation coefficient on random variables $A$ and $B$ is equal to the expected product of these standardized random variables:

$$\rho(A, B) = \mathbb{E}\left[ Z(A) Z(B) \right]. \tag{32}$$

Let $W$, $X$, $Y$ be random variables.

**Identity**. Have $\rho(X, X) = 1$, so $D_{\rho}(X, X) = 0$.

**Symmetry**. Have $\rho(X, Y) = \rho(Y, X)$ by commutativity of multiplication, so $D_{\rho}(X, Y) = D_{\rho}(Y, X)$.

**Triangle Inequality**. For any random variables $A, B$:

$$\mathbb{E}\left[ (Z(A) - Z(B))^2 \right] = \mathbb{E}\left[ Z(A)^2 - 2Z(A)Z(B) + Z(B)^2 \right] \tag{33}$$

$$= \mathbb{E}\left[ Z(A)^2 \right] + \mathbb{E}\left[ Z(B)^2 \right] - 2\mathbb{E}\left[ Z(A)Z(B) \right] \tag{34}$$

$$= 2 - 2\mathbb{E}\left[ Z(A)Z(B) \right] \tag{35}$$

$$= 2 \left( 1 - \rho(A, B) \right) \tag{36}$$

$$= 4 D_{\rho}(A, B)^2. \tag{37}$$

So:

$$4D_\rho(W,Y)^2 = \mathbb{E}\left[(Z(W)-Z(Y))^2\right] \tag{38}$$

$$= \mathbb{E}\left[(Z(W)-Z(X)+Z(X)-Z(Y))^2\right] \tag{39}$$

$$= \mathbb{E}\left[(Z(W)-Z(X))^2\right] + \mathbb{E}\left[(Z(X)-Z(Y))^2\right] \tag{40}$$

$$+ 2\mathbb{E}\left[(Z(W)-Z(X))(Z(X)-Z(Y))\right]$$

$$= 4D_\rho(W,X)^2 + 4D_\rho(X,Y)^2 + 8\mathbb{E}\left[(Z(W)-Z(X))(Z(X)-Z(Y))\right]. \tag{41}$$

Since $\langle A, B \rangle = \mathbb{E}[AB]$ is an inner product over $\mathbb{R}$, it follows by the Cauchy-Schwarz inequality that $\mathbb{E}[AB] \leq \sqrt{\mathbb{E}[A^2]\mathbb{E}[B^2]}$. So:

$$D_\rho(W,Y)^2 \leq D_\rho(W,X)^2 + D_\rho(X,Y)^2 + 2D_\rho(W,X)D_\rho(X,Y) \tag{42}$$

$$= (D_\rho(W,X) + D_\rho(X,Y))^2. \tag{43}$$

Taking the square root of both sides:

$$D_\rho(W,Y) \leq D_\rho(W,X) + D_\rho(X,Y), \tag{44}$$

as required.

**Positive Affine Invariant and Bounded** $D_\rho(aX + c, bY + d) = D_\rho(X,Y)$ is immediate from $\rho(X,Y)$ invariant to positive affine transformations. Have $-1 \leq \rho(X,Y) \leq 1$, so $0 \leq 1-\rho(X,Y) \leq 2$ thus $0 \leq D_\rho(X,Y) \leq 1$. $\qquad\square$

**Theorem 4.7.** *The Equivalent-Policy Invariant Comparison distance is a pseudometric.*

*Proof.* The result follows from $D_\rho$ being a pseudometric. Let $R_A$, $R_B$ and $R_C$ be reward functions mapping from transitions $\mathcal{S} \times \mathcal{A} \times \mathcal{S}$ to real numbers $\mathbb{R}$.

**Identity**. Have:

$$D_{\text{EPIC}}(R_A, R_A) = D_\rho\left(C_{\mathcal{D}_\mathcal{S},\mathcal{D}_\mathcal{A}}\left(R_A\right)\left(S,A,S'\right), C_{\mathcal{D}_\mathcal{S},\mathcal{D}_\mathcal{A}}\left(R_A\right)\left(S,A,S'\right)\right) = 0, \tag{45}$$

since $D_\rho(X,X) = 0$.

**Symmetry**. Have:

$$D_{\text{EPIC}}(R_A, R_B) = D_\rho\left(C_{\mathcal{D}_\mathcal{S},\mathcal{D}_\mathcal{A}}\left(R_A\right)\left(S,A,S'\right), C_{\mathcal{D}_\mathcal{S},\mathcal{D}_\mathcal{A}}\left(R_B\right)\left(S,A,S'\right)\right) \tag{46}$$

$$= D_\rho\left(C_{\mathcal{D}_\mathcal{S},\mathcal{D}_\mathcal{A}}\left(R_B\right)\left(S,A,S'\right), C_{\mathcal{D}_\mathcal{S},\mathcal{D}_\mathcal{A}}\left(R_A\right)\left(S,A,S'\right)\right) \tag{47}$$

$$= D_{\text{EPIC}}(R_B, R_A), \tag{48}$$

since $D_\rho(X,Y) = D_\rho(Y,X)$.

**Triangle Inequality**. Have:

$$D_{\text{EPIC}}(R_A, R_C) = D_\rho\left(C_{\mathcal{D}_\mathcal{S},\mathcal{D}_\mathcal{A}}\left(R_A\right)\left(S,A,S'\right), C_{\mathcal{D}_\mathcal{S},\mathcal{D}_\mathcal{A}}\left(R_C\right)\left(S,A,S'\right)\right) \tag{49}$$

$$\leq D_\rho\left(C_{\mathcal{D}_\mathcal{S},\mathcal{D}_\mathcal{A}}\left(R_A\right)\left(S,A,S'\right), C_{\mathcal{D}_\mathcal{S},\mathcal{D}_\mathcal{A}}\left(R_B\right)\left(S,A,S'\right)\right) \tag{50}$$

$$+ D_\rho\left(C_{\mathcal{D}_\mathcal{S},\mathcal{D}_\mathcal{A}}\left(R_B\right)\left(S,A,S'\right), C_{\mathcal{D}_\mathcal{S},\mathcal{D}_\mathcal{A}}\left(R_C\right)\left(S,A,S'\right)\right) \tag{51}$$

$$= D_{\text{EPIC}}(R_A, R_B) + D_{\text{EPIC}}(R_B, R_C), \tag{52}$$

since $D_\rho(X,Z) \leq D_\rho(X,Y) + D_\rho(Y,Z)$. $\qquad\square$

**Theorem 4.8.** *Let $R_A$, $R'_A$, $R_B$, $R'_B : \mathcal{S} \times \mathcal{A} \times \mathcal{S} \to \mathbb{R}$ be reward functions such that $R'_A \equiv R_A$ and $R'_B \equiv R_B$. Then $0 \leq D_{\text{EPIC}}(R'_A, R'_B) = D_{\text{EPIC}}(R_A, R_B) \leq 1$.*

*Proof.* Since $D_{\text{EPIC}}$ is defined in terms of $D_\rho$, the bounds $0 \leq D_{\text{EPIC}}(R'_A, R'_B)$ and $D_{\text{EPIC}}(R_A, R_B) \leq 1$ are immediate from the bounds in lemma 4.5.

Since $R'_A \equiv R_A$ and $R'_B \equiv R_B$, we can write for $X \in \{A, B\}$:

$$R_X^\lambda(s,a,s') = \lambda_X R_X(s,a,s'), \tag{53}$$

$$R'_X(s,a,s') = R_X^\lambda(s,a,s') + \gamma\Phi_X(s') - \Phi_X(s), \tag{54}$$

for some scaling factor $\lambda_X > 0$ and potential function $\Phi_X : \mathcal{S} \to \mathbb{R}$.

By proposition 4.2:

$$C_{\mathcal{D}_\mathcal{S}, \mathcal{D}_\mathcal{A}} \left( R'_X \right) = C_{\mathcal{D}_\mathcal{S}, \mathcal{D}_\mathcal{A}} \left( R_X^\lambda \right). \tag{55}$$

Moreover, since $C_{\mathcal{D}_\mathcal{S}, \mathcal{D}_\mathcal{A}} \left( R \right)$ is defined as an expectation over $R$ and expectations are linear:

$$C_{\mathcal{D}_\mathcal{S}, \mathcal{D}_\mathcal{A}} \left( R_X^\lambda \right) = \lambda_X C_{\mathcal{D}_\mathcal{S}, \mathcal{D}_\mathcal{A}} \left( R_X \right). \tag{56}$$

Unrolling the definition of $D_{\mathrm{EPIC}}$ and applying this result gives:

$$
\begin{aligned}
D_{\mathrm{EPIC}}(R'_A, R'_B) &= D_\rho \left( C_{\mathcal{D}_\mathcal{S}, \mathcal{D}_\mathcal{A}} \left( R'_A \right) (S, A, S'), C_{\mathcal{D}_\mathcal{S}, \mathcal{D}_\mathcal{A}} \left( R'_B \right) (S, A, S') \right) && \text{(57)}\\
&= D_\rho \left( \lambda_A C_{\mathcal{D}_\mathcal{S}, \mathcal{D}_\mathcal{A}} \left( R_A \right) (S, A, S'), \lambda_B C_{\mathcal{D}_\mathcal{S}, \mathcal{D}_\mathcal{A}} \left( R_B \right) (S, A, S') \right) && \text{eqs. 55 and 56}\\
&= D_\rho \left( C_{\mathcal{D}_\mathcal{S}, \mathcal{D}_\mathcal{A}} \left( R_A \right) (S, A, S'), C_{\mathcal{D}_\mathcal{S}, \mathcal{D}_\mathcal{A}} \left( R_B \right) (S, A, S') \right) && \text{lemma 4.5}\\
&= D_{\mathrm{EPIC}}(R_A, R_B). && \square
\end{aligned}
$$

### A.3.3 NEAREST POINT IN EQUIVALENCE CLASS (NPEC) PREMETRIC

**Proposition A.1.** *(1) $D_{L^p, \mathcal{D}}$ is a metric in $L^p$ space, where functions $f$ and $g$ are identified if they agree almost everywhere on $\mathcal{D}$. (2) $D_{L^p, \mathcal{D}}$ is a pseudometric if functions are identified only if they agree at all points.*

*Proof.* (1) $D_{L^p, \mathcal{D}}$ is a metric in the $L^p$ space since $L^p$ is a norm in the $L^p$ space, and $d(x, y) = \|x - y\|$ is always a metric. (2) As $f = g$ at all points implies $f = g$ almost everywhere, certainly $D_{L^p, \mathcal{D}}(R, R) = 0$. Symmetry and triangle inequality do not depend on identity so still hold. $\square$

**Proposition A.2** (Properties of $D_{\mathrm{NPEC}}^U$). *Let $R_A, R_B, R_C : \mathcal{S} \times \mathcal{A} \times \mathcal{S} \to \mathbb{R}$ be bounded reward functions, and $\lambda \geq 0$. Then $D_{\mathrm{NPEC}}^U$:*

- **Is invariant under $\equiv$ in source:**
  $D_{\mathrm{NPEC}}^U(R_A, R_C) = D_{\mathrm{NPEC}}^U(R_B, R_C)$ if $R_A \equiv R_B$.

- **Invariant under scale-preserving $\equiv$ in target:**
  $D_{\mathrm{NPEC}}^U(R_A, R_B) = D_{\mathrm{NPEC}}^U(R_A, R_C)$ if $R_B - R_C \equiv \texttt{Zero}$.

- **Scalable in target:**
  $D_{\mathrm{NPEC}}^U(R_A, \lambda R_B) = \lambda D_{\mathrm{NPEC}}^U(R_A, R_B)$.

- **Bounded:**
  $D_{\mathrm{NPEC}}^U(R_A, R_B) \leq D_{\mathrm{NPEC}}^U(\texttt{Zero}, R_B)$.

*Proof.* We will show each case in turn.

**Invariance under $\equiv$ in source**

If $R_A \equiv R_B$, then:

$$
\begin{aligned}
D_{\mathrm{NPEC}}^U(R_A, R_C) &\triangleq \inf_{R \equiv R_A} D_{L^p, \mathcal{D}}(R, R_C) && \text{(58)}\\
&= \inf_{R \equiv R_B} D_{L^p, \mathcal{D}}(R, R_C) && \text{(59)}\\
&\triangleq D_{\mathrm{NPEC}}^U(R_B, R_C), && \text{(60)}\\
&&& \text{(61)}
\end{aligned}
$$

since $R \equiv R_A$ if and only if $R \equiv R_B$ as $\equiv$ is an equivalence relation.

**Invariance under scale-preserving $\equiv$ in target**

If $R_B - R_C \equiv \texttt{Zero}$, then we can write $R_B(s, a, s') - R_C(s, a, s') = \gamma \Phi(s') - \Phi(s)$ for some potential function $\Phi : \mathcal{S} \to \mathbb{R}$. Define $f(R)(s, a, s') = R(s, a, s') - \gamma \Phi(s') + \Phi(s)$. Then for any

reward function $R : \mathcal{S} \times \mathcal{A} \times \mathcal{S} \to \mathbb{R}$:

$$
\begin{aligned}
D_{L^p,\mathcal{D}}(R, R_B) &\triangleq \left( \underset{s,a,s' \sim \mathcal{D}}{\mathbb{E}} \left[ |R(s, a, s') - R_B(s, a, s')|^p \right] \right)^{1/p} \\
&= \left( \underset{s,a,s' \sim \mathcal{D}}{\mathbb{E}} \left[ |R(s, a, s') - (R_C(s, a, s') + \gamma\Phi(s') - \Phi(s))|^p \right] \right)^{1/p} \\
&= \left( \underset{s,a,s' \sim \mathcal{D}}{\mathbb{E}} \left[ |(R(s, a, s') - \gamma\Phi(s') + \Phi(s)) - R_C(s, a, s')|^p \right] \right)^{1/p} \\
&= \left( \underset{s,a,s' \sim \mathcal{D}}{\mathbb{E}} \left[ |f(R)(s, a, s') - R_C(s, a, s')|^p \right] \right)^{1/p} \\
&\triangleq D_{L^p,\mathcal{D}}(f(R), R_C),
\end{aligned}
\tag{62}
$$

Crucially, note $f(R)$ is a bijection on the equivalence class $[R]$. Now, substituting this into the expression for the NPEC premetric:

$$
\begin{aligned}
D_{\text{NPEC}}^U(R_A, R_B) &\triangleq \inf_{R \equiv R_A} D_{L^p,\mathcal{D}}(R, R_B) \\
&= \inf_{R \equiv R_A} D_{L^p,\mathcal{D}}(f(R), R_C) && \text{eq. 62} \\
&= \inf_{f(R) \equiv R_A} D_{L^p,\mathcal{D}}(f(R), R_C) && f \text{ bijection on } [R] \\
&= \inf_{R \equiv R_A} D_{L^p,\mathcal{D}}(R, R_C) && f \text{ bijection on } [R] \\
&\triangleq D_{\text{NPEC}}^U(R_A, R_C).
\end{aligned}
\tag{63}
$$

**Scalable in target**

First, note that $D_{L^p,\mathcal{D}}$ is absolutely scalable in both arguments:

$$
\begin{aligned}
D_{L^p,\mathcal{D}}(\lambda R_A, \lambda R_B) &\triangleq \left( \underset{s,a,s' \sim \mathcal{D}}{\mathbb{E}} \left[ |\lambda R_A(s, a, s') - \lambda R_B(s, a, s')|^p \right] \right)^{1/p} \\
&= \left( \underset{s,a,s' \sim \mathcal{D}}{\mathbb{E}} \left[ |\lambda|^p |R_A(s, a, s') - R_B(s, a, s')|^p \right] \right)^{1/p} && |\cdot| \text{ absolutely scalable} \\
&= \left( |\lambda|^p \underset{s,a,s' \sim \mathcal{D}}{\mathbb{E}} \left[ |R_A(s, a, s') - R_B(s, a, s')|^p \right] \right)^{1/p} && \mathbb{E} \text{ linear} \\
&= |\lambda| \left( \underset{s,a,s' \sim \mathcal{D}}{\mathbb{E}} \left[ |R_A(s, a, s') - R_B(s, a, s')|^p \right] \right)^{1/p} \\
&\triangleq |\lambda| D_{L^p,\mathcal{D}}(R_A, R_B).
\end{aligned}
\tag{64}
$$

Now, for $\lambda > 0$, applying this to $D_{\text{NPEC}}^U$:

$$
\begin{aligned}
D_{\text{NPEC}}^U(R_A, \lambda R_B) &\triangleq \inf_{R \equiv R_A} D_{L^p,\mathcal{D}}(R, \lambda R_B) && (65) \\
&= \inf_{R \equiv R_A} D_{L^p,\mathcal{D}}(\lambda R, \lambda R_B) && R \equiv \lambda R && (66) \\
&= \inf_{R \equiv R_A} \lambda D_{L^p,\mathcal{D}}(R, R_B) && \text{eq. 64} && (67) \\
&= \lambda \inf_{R \equiv R_A} D_{L^p,\mathcal{D}}(R, R_B) && && (68) \\
&\triangleq \lambda D_{\text{NPEC}}^U(R_A, R_B). && && (69)
\end{aligned}
$$

In the case $\lambda = 0$, then:

$$D^U_{\text{NPEC}}(R_A, \texttt{Zero}) \triangleq \inf_{R \equiv R_A} D_{L^p, \mathcal{D}}(R, \texttt{Zero}) \tag{70}$$

$$= \inf_{R \equiv R_A} D_{L^p, \mathcal{D}}\left(\frac{1}{2}R, \texttt{Zero}\right) \qquad R \equiv \frac{1}{2}R \tag{71}$$

$$= \inf_{R \equiv R_A} \frac{1}{2} D_{L^p, \mathcal{D}}(R, \texttt{Zero}) \tag{72}$$

$$= \frac{1}{2} \inf_{R \equiv R_A} D_{L^p, \mathcal{D}}(R, \texttt{Zero}) \tag{73}$$

$$= \frac{1}{2} D^U_{\text{NPEC}}(R_A, \texttt{Zero}). \tag{74}$$

Rearranging, we have:

$$D^U_{\text{NPEC}}(R_A, \texttt{Zero}) = 0. \tag{75}$$

### Bounded

Let $d \triangleq D_{\text{NPEC}}(\texttt{Zero}, R_B)$. Then for any $\epsilon > 0$, there exists some potential function $\Phi : \mathcal{S} \to \mathbb{R}$ such that the $L^p$ distance of the potential shaping $R(s, a, s') \triangleq \gamma\Phi(s) - \Phi(s)$ from $R_B$ satisfies:

$$D_{L^p, \mathcal{D}}(R, R_B) \le d + \epsilon. \tag{76}$$

Let $\lambda \in [0, 1]$. Define:

$$R'_\lambda(s, a, s') \triangleq \lambda R_A(s, a, s') + R(s, a, s'). \tag{77}$$

Now:

$$D_{L^p, \mathcal{D}}(R'_\lambda, R) \triangleq \left(\mathop{\mathbb{E}}_{s, a, s' \sim \mathcal{D}} \left[|R'_\lambda(s, a, s') - R(s, a, s')|^p\right]\right)^{1/p} \tag{78}$$

$$= \left(\mathop{\mathbb{E}}_{s, a, s' \sim \mathcal{D}} \left[|\lambda R_A(s, a, s')|^p\right]\right)^{1/p} \tag{79}$$

$$= \left(|\lambda|^p \mathop{\mathbb{E}}_{s, a, s' \sim \mathcal{D}} \left[|R_A(s, a, s')|^p\right]\right)^{1/p} \tag{80}$$

$$= |\lambda| \left(\mathop{\mathbb{E}}_{s, a, s' \sim \mathcal{D}} \left[|R_A(s, a, s')|^p\right]\right)^{1/p} \tag{81}$$

$$= |\lambda| D_{L^p, \mathcal{D}}(R_A, \texttt{Zero}). \tag{82}$$

Since $R_A$ is bounded, $D_{L^p, \mathcal{D}}(R_A, \texttt{Zero})$ must be finite, so:

$$\lim_{\lambda \to 0^+} D_{L^p, \mathcal{D}}(R'_\lambda, R) = 0. \tag{83}$$

It follows that for any $\epsilon > 0$ there exists some $\lambda > 0$ such that:

$$D_{L^p, \mathcal{D}}(R'_\lambda, R) \le \epsilon. \tag{84}$$

Note that $R_A \equiv R'_\lambda$ for all $\lambda > 0$. So:

$$D_{\text{NPEC}}(R_A, R_B) \le D_{L^p, \mathcal{D}}(R'_\lambda, R_B) \tag{85}$$

$$\le D_{L^p, \mathcal{D}}(R'_\lambda, R) + D_{L^p, \mathcal{D}}(R, R_B) \qquad \text{prop. A.1} \tag{86}$$

$$\le \epsilon + (d + \epsilon) \qquad \text{eq. 76 and eq. 84} \tag{87}$$

$$= d + 2\epsilon. \tag{88}$$

Since $\epsilon > 0$ can be made arbitrarily small, it follows:

$$D_{\text{NPEC}}(R_A, R_B) \le d \triangleq D_{\text{NPEC}}(\texttt{Zero}, R_B), \tag{89}$$

completing the proof. $\qquad\qquad\square$

**Theorem 5.4.** $D_{\mathrm{NPEC}}$ *is a premetric on the space of bounded reward functions. Moreover, let* $R_A, R_A{}', R_B, R_B{}' : \mathcal{S} \times \mathcal{A} \times \mathcal{S} \to \mathbb{R}$ *be bounded reward functions such that* $R_A \equiv R_A{}'$ *and* $R_B \equiv R_B{}'$. *Then* $0 \leq D_{\mathrm{NPEC}}(R_A{}', R_B{}') = D_{\mathrm{NPEC}}(R_A, R_B) \leq 1$.

*Proof.* We will first prove $D_{\mathrm{NPEC}}$ is a premetric, and then prove it is invariant and bounded.

**Premetric**

First, we will show that $D_{\mathrm{NPEC}}$ is a premetric.

*Respects identity:* $D_{\mathrm{NPEC}}(R_A, R_A) = 0$

If $D_{\mathrm{NPEC}}^U(\mathtt{Zero}, R_A) = 0$ then $D_{\mathrm{NPEC}}(R_A, R_A) = 0$ as required. Suppose from now on that $D_{\mathrm{NPEC}}^U(R_A, R_A) \neq 0$. It follows from prop A.1 that $D_{L^p, \mathcal{D}}(R_A, R_A) = 0$. Since $X \equiv X$, 0 is an upper bound for $D_{\mathrm{NPEC}}^U(R_A, R_A)$. By prop A.1 $D_{L^p, \mathcal{D}}$ is non-negative, so this is also a lower bound for $D_{\mathrm{NPEC}}^U(R_A, R_A)$. So $D_{\mathrm{NPEC}}^U(R_A, R_A) = 0$ and:

$$D_{\mathrm{NPEC}}(R_A, R_A) = \frac{D_{\mathrm{NPEC}}^U(R_A, R_A)}{D_{\mathrm{NPEC}}^U(\mathtt{Zero}, R_A)} = \frac{0}{D_{\mathrm{NPEC}}^U(\mathtt{Zero}, R_A)} = 0. \tag{90}$$

*Well-defined:* $D_{\mathrm{NPEC}}(R_A, R_B) \geq 0$

By prop A.1, it follows that $D_{L^p, \mathcal{D}}(R, R_B) \geq 0$ for all reward functions $R : \mathcal{S} \times \mathcal{A} \times \mathcal{S}$. Thus 0 is a lower bound for $\{D_{L^p, \mathcal{D}}(R, R_B) \mid R : \mathcal{S} \times \mathcal{A} \times \mathcal{S}\}$, and thus certainly a lower bound for $\{D_{L^p, \mathcal{D}}(R, Y) \mid R \equiv X\}$ for any reward function $X$. Since the infimum is the *greatest* lower bound, it follows that for any reward function $X$:

$$D_{\mathrm{NPEC}}^U(X, R_B) \triangleq \inf_{R \equiv X} D_{L^p, \mathcal{D}}(R, R_B) \geq 0. \tag{91}$$

In the case that $D_{\mathrm{NPEC}}^U(\mathtt{Zero}, R_B) = 0$, then $D_{\mathrm{NPEC}}(R_A, R_B) = 0$ which is non-negative. From now on, suppose that $D_{\mathrm{NPEC}}^U(\mathtt{Zero}, R_B) \neq 0$. The quotient of a non-negative value with a positive value is non-negative, so:

$$D_{\mathrm{NPEC}}(R_A, R_B) = \frac{D_{\mathrm{NPEC}}^U(R_A, R_B)}{D_{\mathrm{NPEC}}^U(\mathtt{Zero}, R_B)} \geq 0. \tag{92}$$

**Invariant and Bounded**

Since $R_B' \equiv R_B$, we have $R_B' - \lambda R_B \equiv \mathtt{Zero}$ for some $\lambda > 0$. By proposition A.2, $D_{\mathrm{NPEC}}^U$ is invariant under scale-preserving $\equiv$ in target and scalable in target. That is, for any reward $R$:

$$D_{\mathrm{NPEC}}^U(R, R_B') = D_{\mathrm{NPEC}}^U(R, \lambda R_B) = \lambda D_{\mathrm{NPEC}}^U(R, R_B). \tag{93}$$

In particular, $D_{\mathrm{NPEC}}^U(\mathtt{Zero}, R_B') = \lambda D_{\mathrm{NPEC}}^U(\mathtt{Zero}, R_B)$. As $\lambda > 0$, it follows that $D_{\mathrm{NPEC}}^U(\mathtt{Zero}, R_B') = 0 \iff D_{\mathrm{NPEC}}^U(\mathtt{Zero}, R_B) = 0$.

Suppose $D_{\mathrm{NPEC}}^U(\mathtt{Zero}, R_B) = 0$. Then $D_{\mathrm{NPEC}}(R, R_B) = 0 = D_{\mathrm{NPEC}}(R, R_B')$ for any reward $R$, so the result trivially holds. From now on, suppose $D_{\mathrm{NPEC}}^U(\mathtt{Zero}, R_B) \neq 0$.

By proposition A.2, $D_{\mathrm{NPEC}}^U$ is invariant to $\equiv$ in source. That is, $D_{\mathrm{NPEC}}^U(R_A, R_B) = D_{\mathrm{NPEC}}^U(R_A', R_B)$, so:

$$D_{\mathrm{NPEC}}(R_A', R_B) = \frac{D_{\mathrm{NPEC}}^U(R_A', R_B)}{D_{\mathrm{NPEC}}^U(\mathtt{Zero}, R_B)} = \frac{D_{\mathrm{NPEC}}^U(R_A, R_B)}{D_{\mathrm{NPEC}}^U(\mathtt{Zero}, R_B)} = D_{\mathrm{NPEC}}(R_A, R_B). \tag{94}$$

By eq. (93):

$$D_{\mathrm{NPEC}}(R_A, R_B') = \frac{\lambda D_{\mathrm{NPEC}}^U(R_A, R_B)}{\lambda D_{\mathrm{NPEC}}^U(\mathtt{Zero}, R_B)} = \frac{D_{\mathrm{NPEC}}^U(R_A, R_B)}{D_{\mathrm{NPEC}}^U(\mathtt{Zero}, R_B)} = D_{\mathrm{NPEC}}(R_A, R_B). \tag{95}$$

Since $D_{\mathrm{NPEC}}$ is a premetric it is non-negative. By the boundedness property of proposition A.2, $D_{\mathrm{NPEC}}^U(R, R_B) \leq D_{\mathrm{NPEC}}^U(\mathtt{Zero}, R_B)$, so:

$$D_{\mathrm{NPEC}}(R_A, R_B) = \frac{D_{\mathrm{NPEC}}^U(R_A, R_B)}{D_{\mathrm{NPEC}}^U(\mathtt{Zero}, R_B)} \leq 1, \tag{96}$$

which completes the proof. □

Note when $D_{L^p,\mathcal{D}}$ is a metric, then $D_{\text{NPEC}}(X,Y) = 0$ if and only if $X = Y$.

**Proposition A.3.** $D_{\text{NPEC}}$ *is* not *symmetric in the undiscounted case.*

*Proof.* We will provide a counterexample showing that $D_{\text{NPEC}}$ is not symmetric.

Choose the state space $\mathcal{S}$ to be binary $\{0,1\}$ and the actions $\mathcal{A}$ to be the singleton $\{0\}$. Choose the coverage distribution $\mathcal{D}$ to be uniform on $s \xrightarrow{0} s$ for $s \in \mathcal{S}$. Take $\gamma = 1$, i.e. undiscounted. Note that as the successor state is always the same as the start state, potential shaping has no effect on $D_{L^p,\mathcal{D}}$, so WLOG we will assume potential shaping is always zero.

Now, take $R_A(s) = 2s$ and $R_B(s) = 1$. Take $p = 1$ for the $L^p$ distance. Observe that $D_{L^p,\mathcal{D}}(\text{Zero}, R_A) = \frac{1}{2}(|0| + |2|) = 1$ and $D_{L^p,\mathcal{D}}(\text{Zero}, R_B) = \frac{1}{2}(|1| + |1|) = 1$. Since potential shaping has no effect, $D^U_{\text{NPEC}}(\text{Zero}, R) = D_{L^p,\mathcal{D}}(\text{Zero}, R)$ and so $D_{\text{NPEC}}(\text{Zero}, R_A) = 1$ and $D_{\text{NPEC}}(\text{Zero}, R_B) = 1$.

Now:

$$D^U_{\text{NPEC}}(R_A, R_B) = \inf_{\lambda > 0} D_{L^p,\mathcal{D}}(\lambda R_A, R_B) \tag{97}$$

$$= \inf_{\lambda > 0} \frac{1}{2}(|1| + |2\lambda - 1|) \tag{98}$$

$$= \frac{1}{2}, \tag{99}$$

with the infimum attained at $\lambda = \frac{1}{2}$. But:

$$D^U_{\text{NPEC}}(R_B, R_A) = \inf_{\lambda > 0} D_{L^p,\mathcal{D}}(\lambda R_B, R_A) \tag{100}$$

$$= \inf_{\lambda > 0} \frac{1}{2} f(\lambda) \tag{101}$$

$$= \frac{1}{2} \inf_{\lambda > 0} f(\lambda), \tag{102}$$

where:

$$f(\lambda) = |\lambda| + |2 - \lambda|, \qquad \lambda > 0. \tag{103}$$

Note that:

$$f(\lambda) = \begin{cases} 2 & \lambda \in (0,2], \\ 2\lambda - 2 & \lambda \in (2,\infty). \end{cases} \tag{104}$$

So $f(\lambda) \geq 2$ on all of its domain, thus:

$$D^U_{\text{NPEC}}(R_B, R_A) = 1. \tag{105}$$

Consequently:

$$D_{\text{NPEC}}(R_A, R_B) = \frac{1}{2} \neq 1 = D_{\text{NPEC}}(R_B, R_A), \tag{106}$$

so $D_{\text{NPEC}}$ is not symmetric. $\square$

### A.4 Full Normalization Variant of EPIC

Previously, we used the Pearson distance $D_\rho$ to compare the canonicalized rewards. Pearson distance is naturally invariant to scaling. An alternative is to explicitly normalize the canonicalized rewards, and then compare them using any metric over functions.

**Definition A.4** (Normalized Reward). *Let $R : \mathcal{S} \times \mathcal{A} \times \mathcal{S} \to \mathbb{R}$ be a bounded reward function. Let $\|\cdot\|$ be a norm on the vector space of reward functions over the real field. Then the normalized $R$ is:*

$$R^N(s, a, s') = \frac{R(s, a, s')}{\|R\|} \tag{107}$$

Note that $(\lambda R)^N = R^N$ for any $\lambda > 0$ as norms are absolutely homogeneous.

We say a reward is *standardized* if it has been canonicalized and then normalized.

**Definition A.5** (Standardized Reward). *Let $R : \mathcal{S} \times \mathcal{A} \times \mathcal{S} \to \mathbb{R}$ be a bounded reward function. Then the standardized $R$ is:*

$$R^S = \left(C_{\mathcal{D}_{\mathcal{S}}, \mathcal{D}_{\mathcal{A}}}(R)\right)^N. \tag{108}$$

Now, we can define a pseudometric based on the direct distance between the standardized rewards.

**Definition A.6** (Direct Distance Standardized Reward). *Let $\mathcal{D}$ be some coverage distribution over transitions $s \xrightarrow{a} s'$. Let $\mathcal{D}_{\mathcal{S}}$ and $\mathcal{D}_{\mathcal{A}}$ be some distributions over states $\mathcal{S}$ and $\mathcal{A}$ respectively. Let $S, A, S'$ be random variables jointly sampled from $\mathcal{D}$. The* Direct Distance Standardized Reward *pseudometric between two reward functions $R_A$ and $R_B$ is the $L^p$ distance between their standardized versions over $\mathcal{D}$:*

$$D_{\text{DDSR}}(R_A, R_B) = \frac{1}{2} D_{L^p, \mathcal{D}}\left(R_A^S(S, A, S'), R_B^S(S, A, S')\right), \tag{109}$$

*where the normalization step, $R^N$, uses the $L^p$ norm.*

For brevity, we omit the proof that $D_{\text{DDSR}}$ is a pseudometric, but this follows from $D_{L^p, \mathcal{D}}$ being a pseudometric in a similar fashion to theorem 4.7. Note it additionally is invariant to equivalence classes, similarly to EPIC.

**Theorem A.7.** *Let $R_A$, $R_A{}'$, $R_B$ and $R_B{}'$ be reward functions mapping from transitions $\mathcal{S} \times \mathcal{A} \times \mathcal{S}$ to real numbers $\mathbb{R}$ such that $R_A \equiv R_A{}'$ and $R_B \equiv R_B{}'$. Then:*

$$0 \leq D_{\text{DDSR}}(R_A', R_B') = D_{\text{DDSR}}(R_A, R_B) \leq 1. \tag{110}$$

*Proof.* The invariance under the equivalence class follows from $R^S$ being invariant to potential shaping and scale in $R$. The non-negativity follows from $D_{L^p, \mathcal{D}}$ being a pseudometric. The upper bound follows from the rewards being normalized to norm 1 and the triangle inequality:

$$D_{\text{DDSR}}(R_A, R_B) = \frac{1}{2} \|R_A^S - R_B^S\| \tag{111}$$

$$\leq \frac{1}{2} \left(\|R_A^S\| + \|R_B^S\|\right) \tag{112}$$

$$= \frac{1}{2} (1 + 1) \tag{113}$$

$$= 1. \qquad \square$$

Since both DDSR and EPIC are pseudometrics and invariant on equivalent rewards, it is interesting to consider the connection between them. In fact, under the $L^2$ norm, then DDSR recovers EPIC. First, we will show that canonical shaping centers the reward functions.

**Lemma A.8** (The Canonically Shaped Reward is Mean Zero). *Let $R$ be a reward function mapping from transitions $\mathcal{S} \times \mathcal{A} \times \mathcal{S}$ to real numbers $\mathbb{R}$. Then:*

$$\mathbb{E}\left[C_{\mathcal{D}_{\mathcal{S}}, \mathcal{D}_{\mathcal{A}}}(R)(S, A, S')\right] = 0. \tag{114}$$

*Proof.* Let $X$, $U$ and $X'$ be random variables that are independent of $S$, $A$ and $S'$ but identically distributed.

$$\text{LHS} \triangleq \mathbb{E}\left[C_{\mathcal{D}_{\mathcal{S}}, \mathcal{D}_{\mathcal{A}}}(R)(S, A, S')\right] \tag{115}$$

$$= \mathbb{E}\left[R(S, A, S') + \gamma R(S', U, X') - R(S, U, X') - \gamma R(X, U, X')\right] \tag{116}$$

$$= \mathbb{E}\left[R(S, A, S')\right] + \gamma \mathbb{E}\left[R(S', U, X')\right] - \mathbb{E}\left[R(S, U, X')\right] - \gamma \mathbb{E}\left[R(X, U, X')\right] \tag{117}$$

$$= \mathbb{E}\left[R(S, U, X')\right] + \gamma \mathbb{E}\left[R(X, U, X')\right] - \mathbb{E}\left[R(S, U, X')\right] - \gamma \mathbb{E}\left[R(X, U, X')\right] \tag{118}$$

$$= 0, \tag{119}$$

where the penultimate step follows since $A$ is identically distributed to $U$, and $S'$ is identically distributed to $X'$ and therefore to $X$. $\qquad \square$

**Theorem A.9.** *$D_{\text{DDSR}}$ with $p = 2$ is equivalent to $D_{\text{EPIC}}$. Let $R_A$ and $R_B$ be reward functions mapping from transitions $\mathcal{S} \times \mathcal{A} \times \mathcal{S}$ to real numbers $\mathbb{R}$. Then:*

$$D_{\text{DDSR}}(R_A, R_B) = D_{\text{EPIC}}(R_A, R_B). \tag{120}$$

*Proof.* Recall from the proof of lemma 4.5 that:

$$D_\rho(U, V) = \frac{1}{2}\sqrt{\mathbb{E}\left[(Z(U) - Z(V))^2\right]} \tag{121}$$

$$= \frac{1}{2}\|Z(U) - Z(V)\|_2, \tag{122}$$

where $\|\cdot\|_2$ is the $L^2$ norm (treating the random variables as functions on a measure space) and $Z(U)$ is a centered (zero-mean) and rescaled (unit variance) random variable. By lemma A.8, the canonically shaped reward functions are already centered under the joint distribution $\mathcal{D}_\mathcal{S} \times \mathcal{D}_\mathcal{A} \times \mathcal{D}_\mathcal{S}$, and normalization by the $L^2$ norm also ensures they have unit variance. Consequently:

$$D_{\text{EPIC}}(R_A, R_B) = D_\rho\left(C_{\mathcal{D}_\mathcal{S}, \mathcal{D}_\mathcal{A}}(R_A)(S, A, S'), C_{\mathcal{D}_\mathcal{S}, \mathcal{D}_\mathcal{A}}(R_B)(S, A, S')\right) \tag{123}$$

$$= \frac{1}{2}\left\|\left(C_{\mathcal{D}_\mathcal{S}, \mathcal{D}_\mathcal{A}}(R_A)(S, A, S')\right)^N - \left(C_{\mathcal{D}_\mathcal{S}, \mathcal{D}_\mathcal{A}}(R_B)(S, A, S')\right)^N\right\|_2 \tag{124}$$

$$= \frac{1}{2}\left\|R_A^S(S, A, S') - R_B^S(S, A, S')\right\|_2 \tag{125}$$

$$= \frac{1}{2}D_{L^p, \mathcal{D}}\left(R_A^S(S, A, S'), R_B^S(S, A, S')\right) \tag{126}$$

$$= D_{\text{DDSR}}(R_A, R_B), \tag{127}$$

completing the proof. $\square$

## A.5 REGRET BOUND

In this section, we derive an upper bound on the regret in terms of the EPIC distance. Specifically, given two reward functions $R_A$ and $R_B$ with optimal policies $\pi_A^*$ and $\pi_B^*$, we show that the regret (under reward $R_A$) of using policy $\pi_B^*$ instead of a policy $\pi_A^*$ is bounded by a function of $D_{\text{EPIC}}(R_A, R_B)$. First, in section A.5.1 we derive a bound for MDPs with finite state and action spaces. In section A.6 we then present an alternative bound for MDPs with arbitrary state and action spaces and Lipschitz reward functions. Finally, in section A.7 we show that in both cases the regret tends to 0 as $D_{\text{EPIC}}(R_A, R_B) \to 0$.

### A.5.1 DISCRETE MDPs

We start in lemma A.10 by showing that $L^2$ distance upper bounds $L^1$ distance. Next, in lemma A.11 we show regret is bounded by the $L^1$ distance between reward functions using an argument similar to [21]. Then in lemma A.12 we relate regret bounds for standardized rewards $R^S$ to the original reward $R$. Finally, in theorem 4.9 we use section A.4 to express $D_{\text{EPIC}}$ in terms of the $L^2$ distance on standardized rewards, deriving a bound on regret in terms of the EPIC distance.

**Lemma A.10.** *Let $(\Omega, \mathcal{F}, P)$ be a probability space and $f : \Omega \to \mathbb{R}$ a measurable function whose absolute value raised to the $n$-th power for $n \in \{1, 2\}$ has a finite expectation. Then the $L^1$ norm of $f$ is bounded above by the $L^2$ norm:*

$$\|f\|_1 \leq \|f\|_2. \tag{128}$$

*Proof.* Let $X$ be a random variable sampled from $P$, and consider the variance of $f(X)$:

$$\mathbb{E}\left[(|f(X)| - \mathbb{E}[|f(X)|])^2\right] = \mathbb{E}\left[|f(X)|^2 - 2|f(X)|\mathbb{E}[|f(X)|] + \mathbb{E}[|f(X)|]^2\right] \tag{129}$$

$$= \mathbb{E}\left[|f(X)|^2\right] - 2\mathbb{E}[|f(X)|]\mathbb{E}[|f(X)|] + \mathbb{E}[|f(X)|]^2 \tag{130}$$

$$= \mathbb{E}\left[|f(X)|^2\right] - \mathbb{E}[|f(X)|]^2 \tag{131}$$

$$\geq 0. \tag{132}$$

Rearranging terms, we have

$$\|f\|_2^2 = \mathbb{E}\left[|f(X)|^2\right] \geq \mathbb{E}[|f(X)|]^2 = \|f\|_1^2. \tag{133}$$

Taking the square root of both sides gives:

$$\|f\|_1 \leq \|f\|_2, \tag{134}$$

as required. $\square$

**Lemma A.11.** *Let $M$ be an MDP\R with finite state and action spaces $\mathcal{S}$ and $\mathcal{A}$. Let $R_A, R_B : \mathcal{S} \times \mathcal{A} \times \mathcal{S} \to \mathbb{R}$ be rewards. Let $\pi_A^*$ and $\pi_B^*$ be policies optimal for rewards $R_A$ and $R_B$ in $M$. Let $\mathcal{D}_\pi(t, s_t, a_t, s_{t+1})$ denote the distribution over trajectories that policy $\pi$ induces in $M$ at time step $t$. Let $\mathcal{D}(s, a, s')$ be the (stationary) coverage distribution over transitions $\mathcal{S} \times \mathcal{A} \times \mathcal{S}$ used to compute $D_{\text{EPIC}}$. Suppose that there exists some $K > 0$ such that $K\mathcal{D}(s_t, a_t, s'_{t+1}) \geq \mathcal{D}_\pi(t, s_t, a_t, s'_{t+1})$ for all time steps $t \in \mathbb{N}$, triples $s_t, a_t, s_{t+1} \in \mathcal{S} \times \mathcal{A} \times \mathcal{S}$ and policies $\pi \in \{\pi_A^*, \pi_B^*\}$. Then the regret under $R_A$ from executing $\pi_B^*$ optimal for $R_B$ instead of $\pi_A^*$ is at most:*

$$G_{R_A}(\pi_A^*) - G_{R_A}(\pi_B^*) \leq \frac{2K}{1 - \gamma} D_{L^1, \mathcal{D}}(R_A, R_B). \tag{135}$$

*Proof.* Noting $G_{R_A}(\pi)$ is maximized when $\pi = \pi_A^*$, it is immediate that

$$G_{R_A}(\pi_A^*) - G_{R_A}(\pi_B^*) = |G_{R_A}(\pi_A^*) - G_{R_A}(\pi_B^*)| \tag{136}$$
$$= |(G_{R_A}(\pi_A^*) - G_{R_B}(\pi_B^*)) + (G_{R_B}(\pi_B^*) - G_{R_A}(\pi_B^*))| \tag{137}$$
$$\leq |G_{R_A}(\pi_A^*) - G_{R_B}(\pi_B^*)| + |G_{R_B}(\pi_B^*) - G_{R_A}(\pi_B^*)|. \tag{138}$$

We will show that both these terms are bounded above by $\frac{K}{1-\gamma} D_{L^1, \mathcal{D}}(R_A, R_B)$, from which the result follows.

First, we will show that for policy $\pi \in \{\pi_A^*, \pi_B^*\}$:

$$|G_{R_A}(\pi) - G_{R_B}(\pi)| \leq \frac{K}{1 - \gamma} D_{L^1, \mathcal{D}}(R_A, R_B). \tag{139}$$

Let $T$ be the horizon of $M$. This may be infinite ($T = \infty$) when $\gamma < 1$; note since $\mathcal{S} \times \mathcal{A} \times \mathcal{S}$ is bounded, so are $R_A, R_B$ so the discounted infinite returns $G_{R_A}(\pi), G_{R_B}(\pi)$ converge (as do their differences). Writing $\tau = (s_0, a_0, s_1, a_1, \cdots)$, we have for any policy $\pi$:

$$\Delta \triangleq |G_{R_A}(\pi) - G_{R_B}(\pi)| \tag{140}$$

$$= \left| \mathbb{E}_{\tau \sim \mathcal{D}_\pi} \left[ \sum_{t=0}^{T} \gamma^t \left( R_A(s_t, a_t, s_{t+1}) - R_B(s_t, a_t, s_{t+1}) \right) \right] \right| \tag{141}$$

$$\leq \mathbb{E}_{\tau \sim \mathcal{D}_\pi} \left[ \sum_{t=0}^{T} \gamma^t |R_A(s_t, a_t, s_{t+1}) - R_B(s_t, a_t, s_{t+1})| \right] \tag{142}$$

$$= \sum_{t=0}^{T} \gamma^t \mathbb{E}_{s_t, a_t, s_{t+1} \sim \mathcal{D}_\pi} [|R_A(s_t, a_t, s_{t+1}) - R_B(s_t, a_t, s_{t+1})|] \tag{143}$$

$$= \sum_{t=0}^{T} \gamma^t \sum_{s_t, a_t, s_{t+1} \in \mathcal{S} \times \mathcal{A} \times \mathcal{S}} \mathcal{D}_\pi(t, s_t, a_t, s_{t+1}) |R_A(s_t, a_t, s_{t+1}) - R_B(s_t, a_t, s_{t+1})|. \tag{144}$$

Let $\pi \in \{\pi_A^*, \pi_B^*\}$. By assumption, $\mathcal{D}_\pi(t, s_t, a_t, s'_{t+1}) \leq K\mathcal{D}(s_t, a_t, s'_{t+1})$, so:

$$\Delta \leq K \sum_{t=0}^{T} \gamma^t \sum_{s_t, a_t, s_{t+1} \in \mathcal{S} \times \mathcal{A} \times \mathcal{S}} \mathcal{D}(s_t, a_t, s_{t+1}) |R_A(s_t, a_t, s_{t+1}) - R_B(s_t, a_t, s_{t+1})| \tag{145}$$

$$= K \sum_{t=0}^{T} \gamma^t D_{L^1, \mathcal{D}}(R_A, R_B) \tag{146}$$

$$= \frac{K}{1 - \gamma} D_{L^1, \mathcal{D}}(R_A, R_B), \tag{147}$$

as required.

In particular, substituting $\pi = \pi_B^*$ gives:

$$|G_{R_B}(\pi_B^*) - G_{R_A}(\pi_B^*)| = |G_{R_A}(\pi_B^*) - G_{R_B}(\pi_B^*)| \leq \frac{K}{1 - \gamma} D_{L^1, \mathcal{D}}(R_A, R_B). \tag{148}$$

Rearranging gives:

$$G_{R_A}(\pi_B^*) \geq G_{R_B}(\pi_B^*) - \frac{K}{1-\gamma} D_{L^1,\mathcal{D}}(R_A, R_B). \tag{149}$$

So certainly:

$$G_{R_A}(\pi_A^*) = \max_\pi G_{R_A}(\pi) \geq G_{R_B}(\pi_B^*) - \frac{K}{1-\gamma} D_{L^1,\mathcal{D}}(R_A, R_B). \tag{150}$$

By a symmetric argument, substituting $\pi = \pi_A^*$ gives:

$$G_{R_B}(\pi_B^*) = \max_\pi G_{R_B}(\pi) \geq G_{R_A}(\pi_A^*) - \frac{K}{1-\gamma} D_{L^1,\mathcal{D}}(R_A, R_B). \tag{151}$$

Eqs. 150 and 151 respectively give $G_{R_B}(\pi_B^*) - G_{R_A}(\pi_A^*) \leq \frac{K}{1-\gamma}$ and $G_{R_A}(\pi_A^*) - G_{R_B}(\pi_B^*) \leq \frac{K}{1-\gamma}$. Combining these gives:

$$|G_{R_A}(\pi_A^*) - G_{R_B}(\pi_B^*)| \leq \frac{K}{1-\gamma} D_{L^1,\mathcal{D}}(R_A, R_B). \tag{152}$$

Substituting inequalities 148 and 152 into eq. 138 yields the required result. □

Note that if $\mathcal{D} = \mathcal{D}_{\mathrm{unif}}$, uniform over $\mathcal{S} \times \mathcal{A} \times \mathcal{S}$, then $K \leq |\mathcal{S}|^2 |\mathcal{A}|$.

**Lemma A.12.** *Let $M$ be an MDP\R with state and action spaces $\mathcal{S}$ and $\mathcal{A}$. Let $R_A, R_B : \mathcal{S} \times \mathcal{A} \times \mathcal{S} \to \mathbb{R}$ be bounded rewards. Let $\pi_A^*$ and $\pi_B^*$ be policies optimal for rewards $R_A$ and $R_B$ in $M$. Suppose the regret under the standardized reward $R_A^S$ from executing $\pi_B^*$ instead of $\pi_A^*$ is upper bounded by some $U \in \mathbb{R}$:*

$$G_{R_A^S}(\pi_A^*) - G_{R_A^S}(\pi_B^*) \leq U. \tag{153}$$

*Then the regret under the original reward $R_A$ is bounded by:*

$$G_{R_A}(\pi_A^*) - G_{R_A}(\pi_B) \leq 4U \|R_A\|_2. \tag{154}$$

*Proof.* Recall that

$$R^S = \frac{C_{\mathcal{D}_\mathcal{S}, \mathcal{D}_\mathcal{A}}(R)}{\|C_{\mathcal{D}_\mathcal{S}, \mathcal{D}_\mathcal{A}}(R)\|_2}, \tag{155}$$

where $C_{\mathcal{D}_\mathcal{S}, \mathcal{D}_\mathcal{A}}(R)$ is simply $R$ shaped with some (bounded) potential $\Phi$. It follows that:

$$G_{R^S}(\pi) = \frac{1}{\|C_{\mathcal{D}_\mathcal{S}, \mathcal{D}_\mathcal{A}}(R)\|_2} G_{C_{\mathcal{D}_\mathcal{S}, \mathcal{D}_\mathcal{A}}(R)}(\pi) \tag{156}$$

$$= \frac{1}{\|C_{\mathcal{D}_\mathcal{S}, \mathcal{D}_\mathcal{A}}(R)\|_2} \left( G_R(\pi) - \mathbb{E}_{s_0 \sim d_0}[\Phi(s_0)] \right), \tag{157}$$

where $s_0$ depends only on the initial state distribution $d_0$. [†] Since $s_0$ does not depend on $\pi$, the terms cancel when taking the difference in returns:

$$G_{R_A^S}(\pi_A^*) - G_{R_A^S}(\pi_B^*) = \frac{1}{\|C_{\mathcal{D}_\mathcal{S}, \mathcal{D}_\mathcal{A}}(R_A)\|_2} \left( G_{R_A}(\pi_A^*) - G_{R_A}(\pi_B^*) \right). \tag{158}$$

Combining this with eq 153 gives

$$G_{R_A}(\pi_A^*) - G_{R_A}(\pi_B^*) \leq U \|C_{\mathcal{D}_\mathcal{S}, \mathcal{D}_\mathcal{A}}(R_A)\|_2. \tag{159}$$

Finally, we will bound $\|C_{\mathcal{D}_\mathcal{S}, \mathcal{D}_\mathcal{A}}(R_A)\|_2$ in terms of $\|R_A\|_2$, completing the proof. Recall:

$$C_{\mathcal{D}_\mathcal{S}, \mathcal{D}_\mathcal{A}}(R)(s, a, s') = R(s, a, s') + \mathbb{E}[\gamma R(s', A, S') - R(s, A, S') - \gamma R(S, A, S')], \tag{160}$$

where $S$ and $S'$ are random variables independently sampled from $\mathcal{D}_\mathcal{S}$ and $A$ sampled from $\mathcal{D}_\mathcal{A}$. By the triangle inequality on the $L^2$ norm and linearity of expectations, we have:

$$\|C_{\mathcal{D}_\mathcal{S}, \mathcal{D}_\mathcal{A}}(R)\|_2 \leq \|R\|_2 + \gamma \|f\|_2 + \|g\|_2 + \gamma|c|, \tag{161}$$

---

[†]In the finite-horizon case, there is also a term $\gamma^T \Phi(s_T)$, where $s_T$ is the fixed terminal state. Since $s_T$ is fixed, it also cancels in eq. 158. This term can be neglected in the discounted infinite-horizon case as $\gamma^T \Phi(s_T) \to 0$ as $T \to \infty$ for any bounded $\Phi$.

where $f(s, a, s') = \mathbb{E}\left[R(s', A, S')\right]$, $g(s, a, s') = \mathbb{E}\left[R(s, A, S')\right]$ and $c = \mathbb{E}\left[R(S, A, S')\right]$. Letting $X'$ be a random variable sampled from $\mathcal{D}_S$ independently from $S$ and $S'$, have

$$\|f\|_2^2 = \mathbb{E}_{X'}\left[\mathbb{E}\left[R(X', A, S')\right]^2\right] \tag{162}$$

$$\leq \mathbb{E}_{X'}\left[\mathbb{E}\left[R(X', A, S')^2\right]\right] \tag{163}$$

$$= \mathbb{E}\left[R(X', A, S')^2\right] \tag{164}$$

$$= \|R\|_2^2. \tag{165}$$

So $\|f\|_2 \leq \|R\|_2$ and, by an analogous argument, $\|g\|_2 \leq \|R\|_2$. Similarly

$$|c| = |\mathbb{E}\left[R(S, A, S')\right]| \tag{166}$$

$$\leq \mathbb{E}\left[|R(S, A, S')|\right] \tag{167}$$

$$= \|R\|_1 \tag{168}$$

$$\leq \|R\|_2 \qquad\qquad \text{lemma A.10.} \tag{169}$$

Combining these results, we have

$$\|C_{\mathcal{D}_S, \mathcal{D}_A}(R)\|_2 \leq 4\|R\|_2. \tag{170}$$

Substituting eq. 170 into eq. 159 yields:

$$G_{R_A}(\pi_A^*) - G_{R_A}(\pi_B^*) \leq 4U\|R_A\|_2, \tag{171}$$

as required. $\qquad\square$

**Theorem 4.9.** *Let $M$ be a $\gamma$-discounted MDP\R with finite state and action spaces $\mathcal{S}$ and $\mathcal{A}$. Let $R_A, R_B : \mathcal{S} \times \mathcal{A} \times \mathcal{S} \to \mathbb{R}$ be rewards, and $\pi_A^*, \pi_B^*$ be respective optimal policies. Let $\mathcal{D}_\pi(t, s_t, a_t, s_{t+1})$ denote the distribution over transitions $\mathcal{S} \times \mathcal{A} \times \mathcal{S}$ induced by policy $\pi$ at time $t$, and $\mathcal{D}(s, a, s')$ be the coverage distribution used to compute $D_{\text{EPIC}}$. Suppose there exists $K > 0$ such that $K\mathcal{D}(s_t, a_t, s_{t+1}) \geq \mathcal{D}_\pi(t, s_t, a_t, s_{t+1})$ for all times $t \in \mathbb{N}$, triples $(s_t, a_t, s_{t+1}) \in \mathcal{S} \times \mathcal{A} \times \mathcal{S}$ and policies $\pi \in \{\pi_A^*, \pi_B^*\}$. Then the regret under $R_A$ from executing $\pi_B^*$ instead of $\pi_A^*$ is at most*

$$G_{R_A}(\pi_A^*) - G_{R_A}(\pi_B^*) \leq 16K\|R_A\|_2 \left(1 - \gamma\right)^{-1} D_{\text{EPIC}}(R_A, R_B),$$

*where $G_R(\pi)$ is the return of policy $\pi$ under reward $R$.*

*Proof.* Recall from section A.4 that:

$$D_{\text{EPIC}}(R_A, R_B) = \frac{1}{2}\left\|R_A^S(S, A, S') - R_B^S(S, A, S')\right\|_2. \tag{172}$$

Applying lemma A.10 we obtain:

$$D_{L^1, \mathcal{D}}(R_A^S, R_B^S) = \left\|R_A^S(S, A, S') - R_B^S(S, A, S')\right\|_1 \leq 2D_{\text{EPIC}}(R_A, R_B). \tag{173}$$

Note that $\pi_A^*$ is optimal for $R_A^S$ and $\pi_B^*$ is optimal for $R_B^S$ since the set of optimal policies for $R^S$ is the same as for $R$. Applying lemma A.11 and eq. 173 gives

$$G_{R_A^S}(\pi_A^*) - G_{R_A^S}(\pi_B^*) \leq \frac{2K}{1 - \gamma}D_{L^1, \mathcal{D}}(R_A^S, R_B^S) \leq \frac{4K}{1 - \gamma}D_{\text{EPIC}}(R_A, R_B). \tag{174}$$

Since $\mathcal{S} \times \mathcal{A} \times \mathcal{S}$ is bounded, $R_A$ and $R_B$ must be bounded, so we can apply lemma A.12, giving:

$$G_{R_A}(\pi_A^*) - G_{R_A}(\pi_B^*) \leq \frac{16K\|R_A\|_2}{1 - \gamma}D_{\text{EPIC}}(R_A, R_B), \tag{175}$$

completing the proof. $\qquad\square$

### A.6 LIPSCHITZ REWARD FUNCTIONS

In this section, we generalize the previous results to MDPs with continuous state and action spaces. The challenge is that even though the spaces may be continuous, the distribution $\mathcal{D}_{\pi^*}$ induced by an optimal policy $\pi^*$ may only have support on some measure zero set of transitions $B$. However, the expectation over a continuous distribution $\mathcal{D}$ is unaffected by the reward at any measure zero subset of points. Accordingly, the reward can be varied *arbitrarily* on transitions $B$ – causing arbitrarily small or large regret – while leaving the EPIC distance fixed. To rule out this pathological case, we assume the rewards are Lipschitz smooth. This guarantees that if the expected difference between rewards is small on a given region, then all points in this region have bounded reward difference.

We start by defining a relaxation of the Wasserstein distance $W_\alpha$ in definition A.13. In lemma A.14 we then bound the expected value under distribution $\mu$ in terms of the expected value under alternative distribution $\nu$ plus $W_\alpha(\mu, \nu)$. Next, in lemma A.15 we bound the regret in terms of the $L^1$ distance between the rewards plus $W_\alpha$; this is analogous to lemma A.11 in the finite case. Finally, in theorem A.16 we use the previous results to bound the regret in terms of the EPIC distance plus $W_\alpha$.

**Definition A.13.** *Let $S$ be some set and let $\mu, \nu$ be probability measures on $S$ with finite first moment. We define the* relaxed Wasserstein distance *between $\mu$ and $\nu$ by:*

$$W_\alpha(\mu, \nu) \triangleq \inf_{p \in \Gamma_\alpha(\mu, \nu)} \int \|x - y\| \, dp(x, y), \tag{176}$$

*where $\Gamma_\alpha(\mu, \nu)$ is the set of probability measures on $S \times S$ satisfying for all $x, y \in S$:*

$$\int_S p(x, y) dy = \mu(x), \tag{177}$$

$$\int_S p(x, y) dx \leq \alpha \nu(y). \tag{178}$$

Note that $W_1$ is equal to the (unrelaxed) Wasserstein distance (in the $\ell_1$ norm).

**Lemma A.14.** *Let $S$ be some set and let $\mu, \nu$ be probability measures on $S$. Let $f : S \to \mathbb{R}$ be an $L$-Lipschitz function on the $\ell_1$ norm $\|\cdot\|_1$. Then, for any $\alpha \geq 1$:*

$$\mathbb{E}_{X \sim \mu} [|f(X)|] \leq \alpha \mathbb{E}_{Y \sim \nu} [|f(Y)|] + L W_\alpha(\mu, \nu). \tag{179}$$

*Proof.* Let $p \in \Gamma_\alpha(\mu, \nu)$. Then:

$$\mathbb{E}_{X \sim \mu} [|f(X)|] \triangleq \int |f(x)| d\mu(x) \qquad \text{definition of } \mathbb{E} \quad (180)$$

$$= \int |f(x)| dp(x, y) \qquad \mu \text{ is a marginal of } p \quad (181)$$

$$\leq \int |f(y)| + L \|x - y\| \, dp(x, y) \qquad f \text{ } L\text{-Lipschitz} \quad (182)$$

$$= \int |f(y)| dp(x, y) + L \int \|x - y\| \, dp(x, y) \qquad (183)$$

$$= \int |f(y)| \int p(x, y) dx dy + L \int \|x - y\| \, dp(x, y) \qquad (184)$$

$$\leq \int |f(y)| \alpha \nu(y) dy + L \int \|x - y\| \, dp(x, y) \qquad \text{eq. 178} \quad (185)$$

$$= \alpha \mathbb{E}_{Y \sim \nu} [|f(Y)|] + L \int \|x - y\| \, dp(x, y) \qquad \text{definition of } \mathbb{E}. \quad (186)$$

Since this holds for all choices of $p$, we can take the infimum of both sides, giving:

$$\mathbb{E}_{X \sim \mu} [|f(X)|] \leq \alpha \mathbb{E}_{Y \sim \nu} [|f(Y)|] + L \inf_{p \in \Gamma_\alpha(\mu, \nu)} \int \|x - y\| \, dp(x, y) \tag{187}$$

$$= \alpha \mathbb{E}_{Y \sim \nu} [|f(Y)|] + L W_\alpha(\mu, \nu). \qquad \square$$

**Lemma A.15.** *Let $M$ be an MDP\R with state and action spaces $\mathcal{S}$ and $\mathcal{A}$. Let $R_A, R_B : \mathcal{S} \times \mathcal{A} \times \mathcal{S} \to$ $\mathbb{R}$ be $L$-Lipschitz, bounded rewards on the $\ell_1$ norm $\|\cdot\|_1$. Let $\pi_A^*$ and $\pi_B^*$ be policies optimal for rewards $R_A$ and $R_B$ in $M$. Let $\mathcal{D}_{\pi,t}(s_t, a_t, s_{t+1})$ denote the distribution over trajectories that policy $\pi$ induces in $M$ at time step $t$. Let $\mathcal{D}(s, a, s')$ be the (stationary) coverage distribution over transitions $\mathcal{S} \times \mathcal{A} \times \mathcal{S}$ used to compute $D_{\mathrm{EPIC}}$. Let $\alpha \geq 1$, and let $B_\alpha(t) = \max_{\pi \in \{\pi_A^*, \pi_B^*\}} W_\alpha(\mathcal{D}_{\pi,t}, \mathcal{D})$. Then the regret under $R_A$ from executing $\pi_B^*$ optimal for $R_B$ instead of $\pi_A^*$ is at most:*

$$G_{R_A}(\pi_A^*) - G_{R_A}(\pi_B^*) \leq \frac{2\alpha}{1-\gamma} D_{L^1, \mathcal{D}}(R_A, R_B) + 4L \sum_{t=0}^{\infty} \gamma^t B_\alpha(t). \tag{188}$$

*Proof.* By the same argument as lemma A.11 up to eq. 144, we have for any policy $\pi$:

$$|G_{R_A}(\pi) - G_{R_B}(\pi)| \leq \sum_{t=0}^{\infty} \gamma^t D_{L^1, \mathcal{D}_{\pi,t}}(R_A, R_B). \tag{189}$$

Let $f(s, a, s') = R_A(s, a, s') - R_B(s, a, s')$, and note $f$ is $2L$-Lipschitz and bounded since $R_A$ and $R_B$ are both $L$-Lipschitz and bounded. Now, by lemma A.14, letting $\mu = \mathcal{D}_{\pi,t}$ and $\nu = \mathcal{D}$, we have:

$$D_{L^1, \mathcal{D}_{\pi,t}}(R_A, R_B) \leq \alpha D_{L^1, \mathcal{D}}(R_A, R_B) + 2L W_\alpha(\mathcal{D}_{\pi,t}, \mathcal{D}). \tag{190}$$

So, for $\pi \in \{\pi_A^*, \pi_B^*\}$, it follows that

$$|G_{R_A}(\pi) - G_{R_B}(\pi)| \leq \frac{\alpha}{1-\gamma} D_{L^1, \mathcal{D}}(R_A, R_B) + 2L \sum_{t=0}^{\infty} \gamma^t B_\alpha(t). \tag{191}$$

By the same argument as for eq. 148 to 152 in lemma A.11, it follows that

$$G_{R_A}(\pi_A^*) - G_{R_A}(\pi_B^*) \leq \frac{2\alpha}{1-\gamma} D_{L^1, \mathcal{D}}(R_A, R_B) + 4L \sum_{t=0}^{\infty} \gamma^t B_\alpha(t), \tag{192}$$

completing the proof. $\qquad\square$

**Theorem A.16.** *Let $M$ be an MDP\R with state and action spaces $\mathcal{S}$ and $\mathcal{A}$. Let $R_A, R_B : \mathcal{S} \times \mathcal{A} \times$ $\mathcal{S} \to \mathbb{R}$ be bounded, $L$-Lipschitz rewards on the $\ell_1$ norm $\|\cdot\|_1$. Let $\pi_A^*$ and $\pi_B^*$ be policies optimal for rewards $R_A$ and $R_B$ in $M$. Let $\mathcal{D}_\pi(t, s_t, a_t, s_{t+1})$ denote the distribution over trajectories that policy $\pi$ induces in $M$ at time step $t$. Let $\mathcal{D}(s, a, s')$ be the (stationary) coverage distribution over transitions $\mathcal{S} \times \mathcal{A} \times \mathcal{S}$ used to compute $D_{\mathrm{EPIC}}$. Let $\alpha \geq 1$, and let $B_\alpha(t) = \max_{\pi \in \pi_A^*, \pi_B^*} W_\alpha(\mathcal{D}_{\pi,t}, \mathcal{D})$. Then the regret under $R_A$ from executing $\pi_B^*$ optimal for $R_B$ instead of $\pi_A^*$ is at most:*

$$G_{R_A}(\pi_A^*) - G_{R_A}(\pi_B^*) \leq 16 \|R_A\|_2 \left( \frac{\alpha}{1-\gamma} D_{\mathrm{EPIC}}(R_A, R_B) + L \sum_{t=0}^{\infty} \gamma^t B_\alpha(t) \right). \tag{193}$$

*Proof.* The proof for theorem 4.9 holds in the general setting up to eq. 173. Applying lemma A.15 to eq. 173 gives

$$G_{R_A^S}(\pi_A^*) - G_{R_A^S}(\pi_B^*) \leq \frac{2\alpha}{1-\gamma} D_{L^1, \mathcal{D}}(R_A^S, R_B^S) + 4L \sum_{t=0}^{\infty} \gamma^t B_\alpha(t) \tag{194}$$

$$\leq \frac{4\alpha}{1-\gamma} D_{\mathrm{EPIC}}(R_A, R_B) + 4L \sum_{t=0}^{\infty} \gamma^t B_\alpha(t). \tag{195}$$

Applying lemma A.12 yields

$$G_{R_A}(\pi_A^*) - G_{R_A}(\pi_B^*) \leq 16 \|R_A\|_2 \left( \frac{\alpha}{1-\gamma} D_{\mathrm{EPIC}}(R_A, R_B) + L \sum_{t=0}^{\infty} \gamma^t B_\alpha(t) \right), \tag{196}$$

as required. $\qquad\square$

A.7    LIMITING BEHAVIOR OF REGRET

The regret bound for finite MDPs, Theorem 4.9, directly implies that, as EPIC distance tends to $0$, the regret also tends to $0$. By contrast, our regret bound in theorem A.16 for (possibly continuous) MDPs with Lipschitz reward functions includes the relaxed Wasserstein distance $W_\alpha$ as an additive term. At first glance, it might therefore appear possible for the regret to be positive even with a zero EPIC distance. However, in this section we will show that in fact the regret tends to $0$ as $D_{\text{EPIC}}(R_A, R_B) \to 0$ in the Lipschitz case as well as the finite case.

We show in lemma A.17 that if the expectation of a non-negative function over a totally bounded measurable metric space $M$ tends to zero under one distribution with adequate support, then it also tends to zero under all other distributions. For example, taking $M$ to be a hypercube in Euclidean space with the Lebesque measure satisfies these assumptions. We conclude in theorem A.18 by showing the regret tends to $0$ as the EPIC distance tends to $0$.

**Lemma A.17.** *Let $M = (S, d)$ be a totally bounded metric space, where $d(x, y) = \|x - y\|$. Let $(S, A, \mu)$ be a measure space on $S$ with the Borel $\sigma$-algebra $A$ and measure $\mu$. Let $p, q \in \Delta(S)$ be probability density functions on $S$. Let $\delta > 0$ such that $p(s) \geq \delta$ for all $s \in S$. Let $f_n : S \to \mathbb{R}$ be a sequence of $L$-Lipschitz functions on norm $\|\cdot\|$. Suppose $\lim_{n\to\infty} \mathbb{E}_{X \sim p}[|f_n(X)|] = 0$. Then $\lim_{n\to\infty} \mathbb{E}_{Y \sim q}[|f_n(Y)|] = 0$.*

*Proof.* Since $M$ is totally bounded, for each $r > 0$ there exists a finite collection of open balls in $S$ of radius $r$ whose union contains $M$. Let $B_r(c) = \{s \in S \mid \|s - c\| < r\}$, the open ball of radius $r$ centered at $c$. Let $C(r)$ denote some finite collection of $Q(r)$ open balls:

$$C(r) = \{B_r(c_{r,n}) \mid n \in \{1, \cdots, Q(r)\}\}, \tag{197}$$

such that $\bigcup_{B \in C(r)} B = S$.

It is possible for some balls $B_r(c_n)$ to have measure zero, $\mu(B_r(c_n)) = 0$, such as if $S$ contains an isolated point $c_n$. Define $P(r)$ to be the subset of $C(r)$ with positive measure:

$$P(r) = \{B \in C(r) \mid \mu(B) > 0\}, \tag{198}$$

and let $p_{r,1}, \cdots, p_{r,Q'(r)}$ denote the centers of the balls in $P$. Since $P(r)$ is a finite collection, it must have a minimum measure:

$$\alpha(r) = \min_{B \in P} \mu(B). \tag{199}$$

Moreover, by construction of $P$, $\alpha(r) > 0$.

Let $S'(r)$ be the union only over balls of positive measure:

$$S'(r) = \bigcup_{B \in P(r)} B. \tag{200}$$

Now, let $D(r) = S \setminus S'(r)$, comprising the (finite number of) measure zero balls in $C(r)$. Since measures are countably additive, it follows that $D(r)$ is itself measure zero: $\mu(D(r)) = 0$. Consequently:

$$\int_S g(s)d\mu = \int_{S'(r)} g(s)d\mu, \tag{201}$$

for any measurable function $g : S \to \mathbb{R}$.

Since $\lim_{n\to\infty} \mathbb{E}_{X \sim p}[|f_n(X)|] = 0$, for all $r > 0$ there exists some $N_r \in \mathbb{N}$ such that for all $n \geq N_r$:

$$\mathbb{E}_{X \sim p}[|f_n(X)|] < \delta L r \alpha(r). \tag{202}$$

By Lipschitz continuity, for any $s, s' \in S$:

$$|f_n(s')| \geq |f_n(s)| - L\|s' - s\|. \tag{203}$$

In particular, since any point $s \in S'(r)$ is at most $r$ distance from some ball center $p_{r,i}$, then $|f_n(p_{r,i})| \geq |f_n(s)| - Lr$. So if there exists $s \in S'(r)$ such that $|f_n(s)| \geq 3Lr$, then there must exist a ball center $p_{r,i}$ with $|f_n(p_{r,i})| \geq 2Lr$. Then for any point $s' \in B_r(p_{r,i})$:

$$|f_n(s')| \geq |f_n(p_{r,i})| - Lr \geq Lr. \tag{204}$$

Now, we have:

$$\mathbb{E}_{X \sim p}\left[|f_n(X)|\right] \triangleq \int_{s \in S} |f_n(s)| p(s) d\mu(s) \tag{205}$$

$$= \int_{s \in S'(r)} |f_n(s)| p(s) d\mu(s) \qquad \text{eq. 201} \tag{206}$$

$$\geq \int_{s \in B_r(p_{r,i})} |f_n(s)| p(s) d\mu(s) \qquad \text{non-negativity of } |f_n(s)| \tag{207}$$

$$\geq \delta \int_{s \in B_r(p_{r,i})} |f_n(s)| d\mu(s) \qquad p(s) \geq \delta \tag{208}$$

$$\geq \delta \cdot Lr \int_{s \in B_r(p_{r,i})} 1 d\mu(s) \qquad \text{eq. 204} \tag{209}$$

$$= \delta Lr \mu(B_r(p_{r,i})) \qquad \text{integrating w.r.t. } \mu \tag{210}$$

$$\geq \delta Lr \alpha(r) \qquad \alpha(r) \text{ minimum of } \mu(B_r(p_{r,i})). \tag{211}$$

But this contradicts eq. 202, and so can only hold if $n < N_r$. It follows that for all $n \geq N_r$ and $s \in S'(r)$, we have $|f_n(s)| < 3Lr$, and so in particular:

$$\mathbb{E}_{Y \sim q}\left[|f_n(Y)|\right] < 3Lr. \tag{212}$$

Let $\epsilon > 0$. Choose $r = \frac{\epsilon}{3L}$. Then for all $n \geq N_r$, $\mathbb{E}_{Y \sim q}\left[|f_n(Y)|\right] < \epsilon$. It follows that:

$$\lim_{n \to \infty} \mathbb{E}_{Y \sim q}\left[|f_n(Y)|\right] = 0, \tag{213}$$

completing the proof. $\qquad \square$

**Theorem A.18.** *Let $M$ be an MDP\R with state and action spaces $\mathcal{S}$ and $\mathcal{A}$. Let $R_A, R_B : \mathcal{S} \times \mathcal{A} \times \mathcal{S} \to \mathbb{R}$ be bounded rewards on some norm $\|\cdot\|$ on $\mathcal{S} \times \mathcal{A} \times \mathcal{S}$. Let $\pi_A^*$ and $\pi_B^*$ be policies optimal for rewards $R_A$ and $R_B$ in $M$. Let $\mathcal{D}_\pi(t, s_t, a_t, s_{t+1})$ denote the distribution over trajectories that policy $\pi$ induces in $M$ at time step $t$. Let $\mathcal{D}(s, a, s')$ be the (stationary) coverage distribution over transitions $\mathcal{S} \times \mathcal{A} \times \mathcal{S}$ used to compute $D_{\mathrm{EPIC}}$.*

*Suppose that either:*

1. *Discrete: $\mathcal{S}$ and $\mathcal{A}$ are discrete. Moreover, suppose that there exists some $K > 0$ such that $K\mathcal{D}(s_t, a_t, s'_{t+1}) \geq \mathcal{D}_\pi(t, s_t, a_t, s'_{t+1})$ for all time steps $t \in \mathbb{N}$, triples $s_t, a_t, s_{t+1} \in \mathcal{S} \times \mathcal{A} \times \mathcal{S}$ and policies $\pi \in \{\pi_A^*, \pi_B^*\}$.*

2. *Lipschitz: $(\mathcal{S} \times \mathcal{A} \times \mathcal{S}, d)$ is a totally bounded measurable metric space where $d(x, y) = \|x - y\|$. Moreover, $R_A$ and $R_B$ are $L$-Lipschitz on $\|\cdot\|$. Furthermore, suppose there exists some $\delta > 0$ such that $\mathcal{D}(s, a, s') \geq \delta$ for all $s, a, s' \in \mathcal{S} \times \mathcal{A} \times \mathcal{S}$, and that $\mathcal{D}_\pi(t, s_t, a_t, s_{t+1})$ is a non-degenerate probability density function (i.e. no single point has positive measure).*

*Then as $D_{\mathrm{EPIC}}(R_A, R_B) \to 0$, $G_{R_A}(\pi_A^*) - G_{R_A}(\pi_B^*) \to 0$.*

*Proof.* In case *(1) Discrete*, by theorem 4.9:

$$G_{R_A}(\pi_A^*) - G_{R_A}(\pi_B^*) \leq \frac{16K\|R_A\|_2}{1 - \gamma} D_{\mathrm{EPIC}}(R_A, R_B). \tag{214}$$

Moreover, by optimality of $\pi_A^*$ we have $0 \leq G_{R_A}(\pi_A^*) - G_{R_A}(\pi_B^*)$. So by the squeeze theorem, as $D_{\mathrm{EPIC}}(R_A, R_B) \to 0$, $G_{R_A}(\pi_A^*) - G_{R_A}(\pi_B^*) \to 0$.

From now on, suppose we are in case *(2) Lipschitz*. By the same argument as lemma A.11 up to eq. 144, we have for any policy $\pi$:

$$\left| G_{R_A^S}(\pi) - G_{R_B^S}(\pi) \right| \leq \sum_{t=0}^{\infty} \gamma^t D_{L^1, \mathcal{D}_{\pi,t}}(R_A^S, R_B^S). \tag{215}$$

Applying lemma A.12 we have:

$$|G_{R_A}(\pi) - G_{R_B}(\pi)| \leq 4\,\|R_A\|_2 \sum_{t=0}^{\infty} \gamma^t D_{L^1,\mathcal{D}_{\pi,t}}(R_A^S, R_B^S). \tag{216}$$

By equation 173, we know that $D_{L^1,\mathcal{D}}(R_A^S, R_B^S) \to 0$ as $D_{\text{EPIC}}(R_A, R_B) \to 0$. By lemma A.17, we know that $D_{L^1,\mathcal{D}_{\pi,t}}(R_A^S, R_B^S) \to 0$ as $D_{L^1,\mathcal{D}}(R_A^S, R_B^S) \to 0$. So we can conclude that as $D_{\text{EPIC}}(R_A, R_B) \to 0$:

$$|G_{R_A}(\pi) - G_{R_B}(\pi)| \to 0, \tag{217}$$

as required. $\qquad\square$

