# OpenReview forum: "Quantifying Differences in Reward Functions"
_ICLR.cc/2021/Conference — ICLR 2021 Spotlight_

### Official Review · AnonReviewer1 · 2020-10-25
**Promising work with some issues**

**Rating:** 8
**Confidence:** 4

**Review:**

**Summary of decision**

The problem setting is significant and the approach is original. The work provides both clear theory and experiments in favour of EPIC to compare reward functions. I currently am leaning towards a marginal rejection because of theoretical issues I've outlined below, but would happily raise my score if they were addressed. Adding additional experiments that I outline below would increase the score further, but fixing the theoretical issues are primary.

**Pros**

A gap in the literature of learning and evaluating reward functions is clearly identified and an explicit argument is made in favour of searching for alternatives to the rollout method.

I really appreciated the serious attempt to develop baselines in section 5, based on a couple of intuitive notions for a desirable pseudometric. A substantial portion of the appendix is devoted to proving properties of the baselines!

The theoretical results are well-presented and comprehensive. I found especially significant the regret bound in Theorem 4.9; in light of this theorem, we have the practical consequence that minimizing EPIC distance closes the gap between optimal policies of different reward functions. I have verified the proofs for correctness and have found issues which I note below. These issues, however, seem quite fixable, and I would certainly increase my score upon seeing a revised version with the fixes.

I enjoyed the discussion of the theory and it's motivation: I especially liked how the validity of the triangle inequality is tied to being able to measure how well one is doing wrt the original reward by using a proxy.

The experimental questions are precise and the experiments to answer them clearly designed. Rather than focus on chasing some benchmarks, I like that the work analyzes the proposed algorithm in terms of being able to compare equivalent reward functions, predicting policy performance, and the sensitivity to different visitation distributions.

**Cons**

Table 2 is a little hard to understand; it might be helpful to provide some summary measure of the discrepancy between the uniform and expert policy distances, which would help in interpreting table 2 in light of section 6.3.

I would have liked to see at least one experiment into examining the failure cases of EPIC, along the lines of the discussion in the last paragraph of section 6.3. The discussion is nice, but it would be even better if a concrete experiment was performed!

The trends in policy performance in table 2 are certainly suggestive and lend credence to the claim that reward distance predicts performance, but I feel the evaluation is still a little lacking. Some more analysis on the probability of achieving the rankings given by each method would be helpful. I also would have liked to see a variant of table 2 where each of the models were treated separately, but measuring the reward distances at various points of training. For example, for the Regression onto reward labels method, the reward distance and episode return could be calculated at selected checkpoints in the Regression training method (e.g., after every x steps in minimizing the mean-squared error). This variant of the experiment would help to give a better idea of the relationship between reward distance and return because it would be "apples to apples". A similar comment applies to the experiment for comparing the reward distances over different visitation distributions.

Being more explicit in the proof of Theorem 4.8 would be helpful. It seems to me to be a result of affine invariance rather than D_\rho being a pseudometric?

The first equality in the proof of invariance under scale on page 24 is incorrect. The expectation is inside of the power 1/p, and the operand is the absolute value of the reward difference to the power of p. I don't see how one can write the term as shown. As far as I can tell, D is also not defined. The same issue permeates the scalable in target proof, and the boundedness proof. It seems though that these proofs still follows through if one uses the proper definition of the L_p norm, so I am willing to discount this issue if it is fixed in a revision.

In the equation after the definition of the joint distribution in Lemme A.15, it seems that the integral of the joint over y should be \alpha \mu(x) \int_{||x - y|| < r*(y)} \nu(y) dy. Since we are integrating over y here, I'm not sure the expression as given is correct. The overall claim of Lemma A.15 seems plausible to me though given the definition of the relaxed Wasserstein distance, so I think a fix should be straightforward.

**Things which didn't affect my decision**

The D_S(x) notation in proposition 4.3 should be defined; from what I understand, it refers to the probability mass of x under the empirical distribution of D_S.

Typo in first sentence of proof of Lemma 4.5: should be "zero mean and unit variance".

Type in equation 21: should be p(x, y) instead of p(x, x).

** Edit after author response **
I've raised my score after the response from the authors regarding my concerns with the theory.

** Edit after further author experiments **
I think the paper is now even stronger given the inclusion of additional experiments into the failure cases of EPIC. I have upgraded my score accordingly.

---

> ### Author Response · Authors · 2020-11-20
> **Revision with corrected proofs, experiments to follow**
>
> Thank you for the detailed review and for catching the mistakes in the supplementary material. We have revised the paper to correct the proofs as follows:
>
>   - Theorem 4.8 (page 23): we inadvertently deleted the proof for this theorem during editing, so the proof presented was actually for Theorem 4.7 -- which would indeed make it difficult to understand! We would like to apologize for this. We have added the correct proof to the supplementary material. As you suggest, this relies on invariance properties (as well as boundedness) of EPIC.
>   - Proposition A.1 (page 24): the proof had been written for an older version of NPEC. It has now been fixed, and the structure of the proof remains largely the same. Thank you very much for spotting this.
>   - Lemma A.16 (page 36; previously Lemma A.15): thanks for catching the issue with the integral being over an area that depends on the variable being integrated over. We have replaced the erroneous proof with a direct proof (not relying on relaxed Wasserstein distance). The new formulation requires that S be totally bounded, not just bounded, but the two are equivalent when S is a subset of $\mathbb{R}^n$.
>
> We have also corrected the typos -- thanks for catching these.
>
> We appreciate the suggestions for two additional experiments: computing distances of reward function checkpoints during training, and examining failures when the visitation distribution $\mathcal{D}$ does not have adequate support. We agree these experiments would strengthen the paper. We will post an update in several days with results from one of these experiments -- let us know if you have a preference for which one to prioritize.

---

> > ### Comment · AnonReviewer1 · 2020-11-21
> > **response to authors**
> >
> > Thank you for the response! I just had an additional question about your new proofs: in the paragraph after Eq 210 for Lemma A.16, I'm not quite sure how the contradiction is generated, as isn't it possible that equality holds throughout? It seems that there should be some strict inequality somewhere for the contradiction to go through. I'd be willing to raise my score immediately to an accept once this last thing is sorted out.
> >
> >
> >
> > I also don't have a preference for the ordering of the additional experiments, but would be willing to raise my score further after seeing the results.

---

> > > ### Author Response · Authors · 2020-11-21
> > > **Fixed limit definition**
> > >
> > > Thank you for the quick response. Eq. (203) erroneously contained $\leq$ when it should have been a strict $<$ following the definition of a limit [1]. We've now fixed this mistake, thank you for catching it.
> > >
> > > [1] Stover, Christopher. "Limit." From MathWorld--A Wolfram Web Resource, created by Eric W. Weisstein. https://mathworld.wolfram.com/Limit.html

---

> > > > ### Comment · AnonReviewer1 · 2020-11-21
> > > > **response to limit**
> > > >
> > > > Thanks for the quick response! I'll update my score.

---

> ### Author Response · Authors · 2020-11-25
> **New experiment investigating pathological cases**
>
> We have revised the paper with a new experiment highlighting possible failure cases of EPIC in the PointMaze environment, in section 6.2 (formerly section 6.3), Table 2 and Table A.8. We have also provided additional guidance on how to choose coverage distributions $\mathcal{D}$ in section 4, immediately after Theorem 4.9.
>
> Thank you for the suggestion -- these experiments proved enlightening. To our surprise, EPIC was very robust to changing the coverage distribution $\mathcal{D}$, giving the same ranking over learned reward models even under extreme choices such as independently sampling states and next states (leading to physically impossible transitions) or restricting the agent to a small region around the initial state. However, more pathological coverage distributions caused the relative difference between reward functions to shrink, likely since all reward models make mistakes on sufficiently challenging distributions.
>
> We did, however, manage to find a more compelling failure mode of EPIC by introducing a hand-designed reward function we call *Mirage* that is identical to the ground-truth at the majority of points, yet leads to a substantially different optimal policy. This fools both EPIC and the baseline methods when the coverage distribution $\mathcal{D}$ does not have adequate support. However, under the **Mix** coverage distribution -- which has broad support over physically possible transitions -- both EPIC and the baselines detect that *Mirage* is substantially different from the ground-truth.
>
> Unfortunately we did not have time to run the other experiment you suggested, comparing checkpoints. However, we are excited about this direction: we will continue to work on this and will include the results from the checkpoint experiment in the final version of the paper.

---

### Official Review · AnonReviewer3 · 2020-10-29
**Recommendation to Accept**

**Rating:** 7
**Confidence:** 3

**Review:**


##########################################################################

Summary:

The paper introduced Equivalent-Policy Invariant Comparison (EPIC) pseudometric to compare different reward functions directly without training a policy function. The authors provide an interesting direction for inverse reinforcement learning. The EPIC distance gives a bound on the regret between policies optimizing for one of the two reward functions relative to the other. The authors also conduct a didactic example to demonstrate efficacy.

##########################################################################

Reasons for score:

Overall, I'd vote for acceptance for the paper. The paper is well written and well-motivated. I find the topic is interesting and worth researching. The paper gives a deep analysis of EPIC in a reasonable setting.  However, my concerns lie on the feasibility :
1) it is not clear to me what the computation cost of distance in high-dimension environments is.
2) the visitation distribution D should have adequate support for every state; however, in most IRL settings, the demonstrated policy could be bias. How does EPIC apply under such settings?

##########################################################################
Questions during the rebuttal period:

Please address and clarify the concerns above.

---

> ### Author Response · Authors · 2020-11-20
> **Answers to questions**
>
> Thank you for your review; we are pleased you found the paper well written and motivated.
>
> To answers your questions:
> > it is not clear to me what the computation cost of distance in high-dimension environments is.
>
> We have added an empirical comparison of runtime to the paper in section A.2.6; thanks for the suggestion. To summarize, we can compute a heatmap like Figure 2(a) (consisting of 25 distance comparisons) in less than 17 seconds with $N_M = N_V = 4096$ samples (see A.1.1 for details of sampling procedure), obtaining a 95% confidence interval $< 0.023$ for all comparisons. By contrast, training PPO in this environment takes around 15 minutes for each reward *per seed*, so it would take over 4 hours to compute a complete heatmap with only 3 seeds. Similarly, computing NPEC in Figure 2(b) with 3 seeds took over 8 hours of CPU time, and has much wider confidence intervals of up to $0.315$.
>
> In general, we would expect the runtime cost of EPIC to be much lower than training an RL policy, which is the currently dominant method. Specifically, the main costs of computing EPIC are:
>   - Sampling transitions from a visitation distribution. This is no more expensive than rolling out a policy for RL training, and could be cheaper, since a database collected offline could be reused to compare multiple reward functions.
>   - Computing the predicted reward. This is also required for RL policy training, and is usually inexpensive.
>
> We would be happy to test the scalability experimentally in higher-dimensional environments. Are there any particular environments you would be interested in seeing runtime experiments in, e.g. Humanoid? The main challenge may be learning good reward functions in a high-dimensional environment -- reward learning methods themselves do not scale too well. So we might have to compare synthetic rewards, such as randomly initialized reward functions, or hand-designed rewards.
>
> > the visitation distribution D should have adequate support for every state; however, in most IRL settings, the demonstrated policy could be bias. How does EPIC apply under such settings?
>
> The visitation distribution $\mathcal{D}$ can be chosen independently of the dataset of expert demonstrations $E$. To clarify, we believe you are considering the setting where EPIC is used to evaluate a reward learned by IRL on demonstrations $E$ that are systematically suboptimal, and $\mathcal{D}$ is sampled from $E$ -- please let us know if we've misinterpreted the question.
>
> You are correct that in this setting IRL would fail to learn the correct reward, and that EPIC (evaluated on the same biased dataset) might miss this. For example, suppose $E$ never visits the most valuable state. IRL would learn not to value this state, and since $\mathcal{D} = E$ never visits this EPIC would not account for this error.
>
> Fortunately, in discrete settings choosing $\mathcal{D}$ to be a uniform distribution over transitions will always satisfy the adequate support assumption of Theorem 4.9. In continuous settings, rollouts from a policy that broadly explores the transitions usually works well: e.g. in Table 2 we use a policy taking actions uniformly at random. EPIC should therefore be able to provide reliable results even in the presence of biased demonstrations.
>
> Of course, if $E$ were not biased, it might be a good choice of $\mathcal{D}$, since it puts more weight on the more important states. If the practitioner is unsure whether or not $E$ is biased, we’d therefore suggest just running EPIC with both $E$ and a more uninformative visitation distribution and comparing results, or mixing the two distributions, similar to our sensitivity analysis in Table 2.
>
> We briefly discuss the importance of choosing an appropriate $\mathcal{D}$ in section 4: “If $\mathcal{D}_S$ and $\mathcal{D}$ have no support for a particular state then the reward of that state has no effect on the distance.” Based on your and others feedback, we intend to expand section 6.3 with an experiment highlighting the problems of choosing the wrong $\mathcal{D}$. We would also be happy to expand on this point further elsewhere in the paper -- do you feel this would be beneficial?

---

> > ### Comment · AnonReviewer3 · 2020-11-23
> > **response to authors**
> >
> > Thanks for your timely response, and it addressed my concerns. I will retain the score I have presented above.
> > looking forward to seeing the further study, I believe that will be beneficial for practitioners leveraging EPIC in more complex environments.

---

### Official Review · AnonReviewer2 · 2020-10-29
**Interesting work on measuring the rewards without policy optimization**

**Rating:** 7
**Confidence:** 4

**Review:**

Summary:

The paper introduces a pseudometric on reward functions, EPIC (Equivalent-Policy Invariant Comparison), based on the potential-based reward shaping (Ng. 2020). It formally analyzes the EPIC distance in detail and demonstrates its usefulness in comparing learned reward functions without the necessity of optimizing reward-specific policies. The empirical results show that the EPIC is more predictive of the policy returns than some baseline variants and robust to visitation distributions, even in unseen test environments.


Pros:

-- This paper proposes and analyzes a theoretically grounded distance metric to evaluate learned rewards. It has several properties appealing to practical RL tasks. The EPIC pseudometric does not rely on policy optimization, which differentiates it from previous approaches. Moreover, EPIC is computable over arbitrary visitation distributions, making EPIC flexible in various practical scenarios. Finally, EPIC provides bounds on the policy performance difference when the visitation distributions meet some constraints.

-- EPIC can significantly impact the reward learning/design-related communities because it is the first rigorous approach to evaluating candidate reward functions against the ground truth reward function without direct or indirect policy optimization.


Cons:

-- The claimed property that EPIC bounds the difference in policy returns relies heavily on the visitation distribution. It seems that the capability of EPIC to provide relevant information mostly comes from the visitation distribution. It would be helpful if there are some additional discussions about the EPIC's predictive power when this kind of reliance could be problematic. Moreover, the claim focuses on the upper bound of the difference. In some cases, it could also be helpful to know the lower bound of the policy-return difference if the EPIC value is big enough.

-- The theoretical derivations and analyses of EPIC are the most substantial parts of the paper.  Some additional experimental studies on domains with high-dimension perceptions would strengthen the paper's contributions and provide more guidance in adapting EPIC.



In section 6.3, all three metrics degenerate significantly in the mixture case. Why does the predictive power of EPIC decrease in this mixture setting?

---

> ### Author Response · Authors · 2020-11-20
> **Clarification & response to questions**
>
> Thank you for your review, and we are pleased that you feel EPIC has the potential to significantly impact the reward-learning community.
>
> > The claimed property that EPIC bounds the difference in policy returns relies heavily on the visitation distribution.
>
> We would like to clarify the dependence of EPIC on the visitation distribution $\mathcal{D}$. Theorem 4.9, which bounds the difference in policy return for discrete MDPs, requires that $\mathcal{D}$ differs by no more than a constant factor than the distribution induced by an optimal policy. Fortunately, this is *always* attainable in a finite state space by choosing $\mathcal{D}$ to be uniform over all transitions.
>
> Of course, a tighter bound can be achieved by choosing $\mathcal{D}$ to be closer to that of desired behaviour. However, our empirical results in Table 2 show that even with uninformative choices of $\mathcal{D}$ -- such as rollouts from a policy taking actions at random -- EPIC is able to predict the return of policies trained on the reward both in the training and an unseen test environment.
>
> > It would be helpful if there are some additional discussions about the EPIC's predictive power when this kind of reliance could be problematic.
>
> We plan to add an additional experiment using a poorly chosen or pathological visitation distribution that violates the assumption of theorem 4.9 to illustrate possible failure modes. We will post a reply when we revise the paper to include this.
>
> > In some cases, it could also be helpful to know the lower bound of the policy-return difference if the EPIC value is big enough.
>
> This would be a nice property, but unfortunately without additional assumptions on the environment no method can provide such a lower bound. For example, suppose the deployment environment transitions to a randomly chosen state independent of the action taken. In this case, all policies will obtain the same expected return, so the policy-return difference is always zero -- no matter how different the reward functions may be. Thank you for posing this question: we have added this impossibility result to section 4 (after theorem 4.9).
>
> > In section 6.3, all three metrics degenerate significantly in the mixture case. Why does the predictive power of EPIC decrease in this mixture setting?
>
> We did not observe any significant decrease in the ability of EPIC to predict policy returns in the mixture distribution. In particular, EPIC has an almost identical ranking in the mixture setting (Regress > Preferences > AIRL {SO,SA}). It only disagrees about the ordering between AIRL SO and AIRL SA, however these are fairly close in the other two distributions as well.
>
> The only notable difference we observed between the mixture distribution and other distributions is that all three distances are smaller in absolute terms (even if the relative distances are similar.). The absolute value of the distance from the ground-truth will tend to increase when the distribution is more challenging to predict, and decrease when it is easier to predict.
>
> Our conjecture for these results is that the mixture distribution is easier for learned rewards to predict. Since it is a mixture of the expert and a random distribution, many of the transitions sampled are near to but not at the goal state, either moving towards or away from it. This is easy to classify (moving towards good, away bad). By contrast, since the expert policy spends most of its time at the goal state, a learned reward function that is slightly wrong about the location of the goal state would classify many of these states incorrectly (e.g. moving away from the goal when in fact the agent is moving towards it). Conversely, the random policy is likely to reach states that were rarely or never seen during training the learned rewards, so the output is more likely to deviate from the ground truth.
>
> We are not sure if we fully understood this question, so please do let us know if we can help by providing any further information.

---

> > ### Comment · AnonReviewer2 · 2020-11-21
> > **Updates after authors' response**
> >
> > Thanks for the response! It addressed my concerns, and I have updated the score.

---

> ### Author Response · Authors · 2020-11-25
> **New experiment investigating visitation distributions**
>
> We are glad our previous response addressed your questions, thank you for the quick reply. We have made a final revision of the paper to address your and reviewer 1’s concern regarding the dependence of our method on the coverage distribution:
>
> > The claimed property that EPIC bounds the difference in policy returns relies heavily on the visitation distribution. It seems that the capability of EPIC to provide relevant information mostly comes from the visitation distribution. It would be helpful if there are some additional discussions about the EPIC's predictive power when this kind of reliance could be problematic.
>
> Specifically, we have added a discussion on how to choose an appropriate distribution $\mathcal{D}$ in section 4 immediately following theorem 4.9, and connect our empirical results to this at the end of section 6.2.
>
> We have also expanded our earlier sensitivity analysis to include a range of pathological coverage distributions in Table A.8, discussed in section 6.2. To our surprise, EPIC produced a similar ranking over learned reward models even under some extreme choices of coverage distribution, including distributions containing mostly physically impossible transitions, or those limited to a very narrow region of state space.
>
> However, we found that EPIC and baseline methods are sensitive to the choice of coverage distribution on a hand-designed reward *Mirage*, that is identical to the ground-truth except at a small number of points. In this case, when the coverage distribution is too narrow, the methods wrongly think *Mirage* is highly similar to the ground truth -- when in fact it has a substantially different optimal policy. While an adversarially chosen reward like *Mirage* is unlikely to occur in the wild, we hope this example will illustrate potential pitfalls, and help practitioners choose a distribution $\mathcal{D}$ that is appropriate for their needs.

---

### Official Review · AnonReviewer4 · 2020-10-31
**Marginally above acceptance threshold**

**Rating:** 6
**Confidence:** 2

**Review:**

- This paper proposes one main method EPIC to measure the differences between the reward functions of MDPs, and two weaker baselines methods NPEC, ERC. The methods are useful in directly comparing two different reward functions on a common MDPs, without running RL algorithms and comparing the resulted performance.
- To me, the proposed approach is novel and interesting, and there is no similar previous work to my best knowledge, so I tend to give it an accept.
- There are extensive experiments, but most of them are still based on hand-designed reward, and to judge whether the distance measure is good or not, it is still largely based on human judgement.  In Table 2, there is some experiment showing some correspondence between the "distance to ground truth" and the "return". The result shows some effect of all three proposed approaches.
- You're now comparing the reward functions directly in the "reward space". But very different reward functions might still give similar optimal policies, and make the resulted performance similar.   Is it possible to compare the reward functions by inferring the similarity between the optimal policies (and without resorting to a RL algorithm)?  What's the pros and cons of this direction and your current direction?

============ after rebuttal =============

Thank you for the response and revision, and I am sorry for my late response.  One experiment that I can think of to get around the issue of failure in RL optimization is to consider simple examples where RL algorithms can surely find the global optima.  This is the case for the tabular setting (like your GridWorld). This can be a proof of concept, but I admit that it might also be limited. So I think it is okay to keep your current presentation.

---

> ### Author Response · Authors · 2020-11-20
> **Clarification and response to questions**
>
> Thank you for your review, and we are glad that you found the approach novel and interesting.
>
> > Most of them [the experiments] are still based on hand-designed reward, and to judge whether the distance measure is good or not, it is still largely based on human judgement.
>
> We chose to use hand-designed reward functions for some of the experiments (e.g. Figure 2) so that the reward functions under comparison could be understood in full by the reader, providing clearer intuitions about the properties of the different reward metrics. Would your concern be alleviated if we add experiments comparing distances of the environment at different checkpoints of learned rewards, as suggested by reviewer #1? Are any other experiments that you feel would alleviate this concern?
>
> We agree that the subjectivity of human judgement can be problematic. However, conversely we would be hesitant on relying only on return as a quantitative metric, since it can produce false negatives (due to failures of RL optimization) and false positives (when rewards value different features that are correlated at training but not at deployment) -- see paragraph 2 of the introduction for more details. Unfortunately we’re not aware of any good quantitative metrics for us to compare our results to, which is why we developed two baselines (NPEC and ERC) for this paper. Given this, we felt that a mixture of experiments using human judgement and quantitative results was most informative. We would love to report other quantitative metrics. Do you know of any?
>
> > Is it possible to compare the reward functions by inferring the similarity between the optimal policies (and without resorting to a RL algorithm)?
>
> While we provide an upper bound on the difference between optimal policies return based on the distance, it is unfortunately not possible to lower bound the difference between policies without making additional assumptions on the environment. For example, suppose the deployment environment transitions to a randomly chosen state independently from the action taken. In this case, any policy is optimal for any reward function -- so there is no difference between optimal policies. We have added this impossibility result to section 4 (after theorem 4.9).
>
> It may be possible to make comparisons about optimal policies with additional assumptions about the environment, and we would be interested in exploring this in future work. Unfortunately, in the worst case any such method will be similarly expensive as finding the optimal policy. Given a distance $D$ based on optimal policy similarity, we can derive the optimal policy to any reward function R by probing $D$ as to whether taking action $u$ in state $x$ is optimal by comparing the ground-truth reward $R$ to a reward augmented with an indicator function $R'(s,a,s’) = \frac{R_{\max}}{1 - \gamma}\mathbb{1}[s = x, a = u] + R(s,a,s')$. By repeating this, one could effectively read out the optimal policy by queries to $D$.

---

### Author Response · Authors · 2020-11-25
**Author Summary**

Thanks to all reviewers for your time and effort the past two weeks. We are pleased the feedback has been positive, and are grateful for your constructive suggestions for improvements.

We have responded to all reviewers individually. For convenience, we summarize the main changes to the paper from the original submission:
  - The proofs for Theorem 4.8, Proposition A.1, and Lemma A.16 have been corrected.
  - We have outlined an impossibility result showing that distance metrics cannot lower bound difference in policy returns without making assumptions about environment dynamics (page 5, 4th paragraph after theorem 4.9).
  - We have added theoretically inspired recommendations for how to choose a coverage distribution $\mathcal{D}$ (page 5, 2nd and 3rd paragraphs after theorem 4.9).
  - We have added a new experiment on four different "pathological" coverage distributions (Table A.8) and a new reward function, *Mirage*. EPIC maintained the same ranking over learned reward models even under the pathological distributions, whereas baselines were sensitive to this. However, with *Mirage* -- which was hand-designed to be difficult to evaluate -- EPIC is sensitive to the choice of coverage distribution. These empirical results confirm the theoretically motivated recommendation of the previous point.

---

### Decision · Program_Chairs · 2021-01-07
**Final Decision**

**Decision:**

Accept (Spotlight)

**Comment:**

The proposed approach for evaluating reward functions is theoretically grounded while having several properties appealing to practical RL tasks. This novel approach fills a gap in the literature. All reviewers agree that this paper has a place at ICLR.